# SA-*Solver*: Stochastic Adams Solver for Fast Sampling of Diffusion Models

**Shuchen Xue**[1,4]\*,**Mingyang Yi**[2]†,**Weijian Luo**[3]\*,**Shifeng Zhang**[2],**Jiacheng Sun**[2],

**Zhenguo Li**[2]**, Zhi-Ming Ma**[1,4]

[1]University of Chinese Academy of Sciences [2] Huawei Noah's Ark Lab [3] Peking University
[4]Academy of Mathematics and Systems Science, Chinese Academy of Sciences

## Abstract

Diffusion Probabilistic Models (DPMs) have achieved considerable success in generation tasks. As sampling from DPMs is equivalent to solving diffusion SDE or ODE which is time-consuming, numerous fast sampling methods built upon improved differential equation solvers are proposed. The majority of such techniques consider solving the diffusion ODE due to its superior efficiency. However, stochastic sampling could offer additional advantages in generating diverse and high-quality data. In this work, we engage in a comprehensive analysis of stochastic sampling from two aspects: variance-controlled diffusion SDE and linear multi-step SDE solver. Based on our analysis, we propose *SA-Solver*, which is an improved efficient stochastic Adams method for solving diffusion SDE to generate data with high quality. Our experiments show that *SA-Solver* achieves: 1) improved or comparable performance compared with the existing state-of-the-art (SOTA) sampling methods for few-step sampling; 2) SOTA FID on substantial benchmark datasets under a suitable number of function evaluations (NFEs).

## 1   Introduction

Diffusion Probabilistic Models (DPMs) [1–3] have demonstrated substantial success across a broad spectrum of generative tasks such as image synthesis [4–6], video generation [7, 8], text-to-image generation [9–11], speech synthesis [12, 13], *etc*. The primary mechanism of DPMs involves a forward diffusion process that incrementally introduces noise into data. Simultaneously, a reverse diffusion process is learned to generate data from this noise. Despite DPMs demonstrating enhanced generation performance in comparison to alternative methods such as Generative Adversarial Networks (GAN) [14] or Variational Autoencoders (VAE) [15], the sampling process of DPMs demand hundreds of evaluations of network function evaluations (NFE) [2]. The substantial computation requirement poses a significant limitation to their wider application in practice.

The existing literature on improving the sampling efficacy of DPMs can be categorized into two ways, depending on whether conducting extra training on the DPMs. The first category necessitates supplementary training [16–21], which often emerges as a bottleneck, thereby limiting their practical application. Due to this, we focus on exploring the second category, which consists training-free methods to improve the sampling efficiency of DPMs in this paper. Current training-free samplers employ efficient numerical schemes to solve the diffusion SDE/ODE[22–26]. Compared with solving diffusion SDE (stochastic sampler) [25–27], solving diffusion ODE (deterministic sampler) [22–24]

---

\*Work done during an internship at Huawei Noah's Ark Lab. Email: `xueshuchen17@mails.ucas.ac.cn`,
`luoweijian@stu.pku.edu.cn`

†Corresponding authors: Mingyang Yi (`yimingyang2@huawei.com`)

37th Conference on Neural Information Processing Systems (NeurIPS 2023).

empirically exhibits better sampling efficiency. Existing stochastic samplers typically exhibit slower convergence speed. However, empirical observations in [3, 27] indicate that the stochastic sampler has the potential to generate higher-quality data when increasing the sampling steps. This empirical observation motivates us to further explore the efficient stochastic sampler.

Owing to the observed superior performance of stochastic sampler [3, 27], we speculate that adding properly scaled noise in the diffusion SDE may facilitate the quality of generated data. Thus, instead of solving the vanilla diffusion SDE in [22], we propose to consider a family of diffusion SDEs which shares the same marginal distribution [28, 27] with different noise scales. Meanwhile, efficient stochastic solvers are not carefully studied, which could be the reason that diffusion ODE exhibits better sampling efficiency. To overcome this problem, we study the linear multi-step SDE solvers [29] and incorporate them in the sampling.

Based on these studies, we propose *SA-Solver* with theoretical convergence order to solve the proposed diffusion SDEs. Our *SA-Solver* is based on the stochastic Adams method in [29], by adapting it to the exponentially weighted integral and analytical variance. With the proposed diffusion SDEs and *SA-Solver*, we can efficiently generate data with controllable noise scales. We empirically evaluate our *SA-Solver* on plenty of benchmark datasets of image generation. The evaluation criterion is the Fréchet Inception Distance (FID) score [30] under different number of function evaluations (NFEs). The experimental results can be summarized as three folds: 1) Under small NFEs, our *SA-Solver* has improved or comparable FID scores, compared with baseline methods; 2) Under suitable NFEs our *SA-Solver* achieves the State-of-the-Art FID scores over all benchmark datasets; 3) *SA-Solver* achieves superior performance over deterministic samplers when the model is not fully trained.

## 2   Related Works

The DPMs originate from the milestone work [1], and are further developed by [2] and [3] to successfully generate high-quality data, under the framework of discrete and continuous diffusion SDEs respectively. In this paper, we mainly focus on the latter framework. As mentioned in Section 1, plenty of papers are working on accelerating the sampling of DPMs due to their low efficiency, distinguished by whether conducting a supplementary training stage. Training-based methods, e.g., knowledge distillation [16–18], learning-to-sample [19], and integration with GANs [20, 21], have the potential to sampling for one or very few steps to enhance the efficiency, but their applicability is limited by the lack of a plug-and-play nature, thereby constraining their broad applicability across diverse tasks. Thus we mainly focus on the training-free methods in this paper.

**Solving Diffusion ODE.**   Since the sampling process is equivalent to solving diffusion SDE (ODE), the training-free methods are mainly built on solving the differential equations via high-efficiency numerical methods. As ODEs are easier to solve compared with SDEs, the ODE sampler has attracted great attention. For example, Song et al. [22] provides an empirically efficient solver DDIM. Zhang and Chen [28] and Lu et al. [23] point out the semi-linear structure of diffusion ODEs, and develop higher-order ODE samplers based on it. Zhao et al. [24] further improve these samplers in terms of NFEs by integrating the mechanism of predictor-corrector method.

**Solving Diffusion SDE.**   Though less explored than the ODE sampler, the SDE sampler exhibits the potential of generating higher-quality data [27]. Thus developing an efficient SDE sampler as we did in this paper is a meaningful topic. In the existing literature, researchers [2, 26, 3] solve the diffusion SDE by first-order discretization numerical method. The higher-order stochastic sampler of diffusion SDE has also been discussed in [25]. Karras et al. [27] proposes another stochastic sampler (which is not a general SDE numerical solver) tailored for diffusion problems. However, in contrast to our proposed *SA-Solver*, the existing SDE samplers are limited due to their low efficiency [2, 26, 3] or sensitivity to hyperparameters [27].

We found a concurrent paper proposing an SDE sampler SDE-DPM-Solver++ [31] which is similar to our *SA-Solver*. Though both methods develop multi-step diffusion SDE samplers, our *SA-Solver* is different from SDE-DPM-Solver++ as follows: 1) *SA-Solver* incorporates the predictor-corrector method, which helps improve the quality of generated data [3, 32, 24]; 2) In contrast to SDE-DPM-Solver++, *SA-Solver* has theoretical guarantees with proved convergence order; 3) SDE-DPM-Solver++ is a special case of *SA-Solver* when the predictor step equals 2 with no corrector in our predictor-corrector method, while our solver supports arbitrary orders with analytical forms.

# 3 Preliminary

In the regime of the continuous stochastic differential equation (SDE), Diffusion Probabilistic Models (DPMs) [1–3, 33] construct noisy data through the following linear SDE:

$$\mathrm{d}\boldsymbol{x}_t = f(t)\boldsymbol{x}_t \mathrm{d}t + g(t)\mathrm{d}\boldsymbol{w}_t, \tag{1}$$

where $\boldsymbol{w}_t \in \mathbb{R}^d$ represents the standard Wiener process, $f(t)\boldsymbol{x}_t$ and $g(t)$ respectively denote the drift and diffusion coefficients. For each time $t \in [0, T]$, $\boldsymbol{x}_t|\boldsymbol{x}_0 \sim \mathcal{N}(\alpha_t\boldsymbol{x}_0, \sigma_t^2\boldsymbol{I})$.

Let $p_t(\boldsymbol{x})$ denotes the marginal distribution of $\boldsymbol{x}_t$, the coefficients $f(t)$ and $g(t)$ are meticulously selected to guarantee that the marginal distribution $p_T(\boldsymbol{x}_T)$ closely approximates a Gaussian distribution, i.e., $\mathcal{N}(\boldsymbol{0}, \boldsymbol{I})$, and the *signal-to-noise-ratio* (SNR) $\alpha_t^2/\sigma_t^2$ is strictly decreasing w.r.t. $t$. In the sequel, we follow the established notations in [33]:

$$f(t) = \frac{\mathrm{d}\log\alpha_t}{\mathrm{d}t}, \quad g^2(t) = \frac{\mathrm{d}\sigma_t^2}{\mathrm{d}t} - 2\frac{\mathrm{d}\log\alpha_t}{\mathrm{d}t}\sigma_t^2. \tag{2}$$

Anderson [34] demonstrates a pivotal theorem that the forward process (1) has an equivalent reverse-time diffusion process (from $T$ to 0) as the following equation, so that generating process can be equivalent to numerically solve the diffusion SDE [2, 3].

$$\mathrm{d}\boldsymbol{x}_t = \left[f(t)\boldsymbol{x}_t - g^2(t)\nabla_{\boldsymbol{x}}\log p_t(\boldsymbol{x}_t)\right]\mathrm{d}t + g(t)\mathrm{d}\bar{\boldsymbol{w}}_t, \qquad \boldsymbol{x}_T \sim p_T(\boldsymbol{x}_T) \tag{3}$$

where $\bar{\boldsymbol{w}}_t$ represents the Wiener process in reverse time, and $\nabla_{\boldsymbol{x}}\log p_t(\boldsymbol{x})$ is the score function.

Moreover, Song et al. [3] also prove that there exists a corresponding deterministic process whose trajectories share the same marginal probability densities $p_t(\boldsymbol{x})$ as (3), so that the ODE solver can be adopted for efficient sampling [23, 24]:

$$\mathrm{d}\boldsymbol{x}_t = \left[f(t)\boldsymbol{x}_t - \frac{1}{2}g^2(t)\nabla_{\boldsymbol{x}}\log p_t(\boldsymbol{x}_t)\right]\mathrm{d}t, \qquad \boldsymbol{x}_T \sim p_T(\boldsymbol{x}_T) \tag{4}$$

To get the *score function* $\nabla_{\boldsymbol{x}}\log p_t(\boldsymbol{x}_t)$ in (3), we usually take neural network $\boldsymbol{s}_{\boldsymbol{\theta}}(\boldsymbol{x}, t)$ parameterized by $\boldsymbol{\theta}$ to approximate it by optimizing the denoising score matching loss [3]:

$$\boldsymbol{\theta}^* = \arg\min_{\boldsymbol{\theta}} \mathbb{E}_t\left\{\lambda(t)\mathbb{E}_{\boldsymbol{x}_0}\mathbb{E}_{\boldsymbol{x}_t|\boldsymbol{x}_0}\left[\left\|\boldsymbol{s}_{\boldsymbol{\theta}}(\boldsymbol{x}, t) - \nabla_{\boldsymbol{x}_t}\log p_{0t}(\boldsymbol{x}_t|\boldsymbol{x}_0)\right\|_2^2\right]\right\}. \tag{5}$$

In practice, two methods are used to reparameterize the score-based model [35]. The first approach utilizes a *noise prediction model* such that $\boldsymbol{\epsilon}_{\boldsymbol{\theta}}(\boldsymbol{x}_t, t) = -\sigma_t\boldsymbol{s}_{\boldsymbol{\theta}}(\boldsymbol{x}_t, t)$, while the second employs a *data prediction model*, represented by $\boldsymbol{x}_{\boldsymbol{\theta}}(\boldsymbol{x}_t, t) = (\boldsymbol{x}_t - \sigma_t\boldsymbol{\epsilon}_{\boldsymbol{\theta}}(\boldsymbol{x}_t, t))/\alpha_t$. The reparameterized models are plugged into the sampling process (3) or (4) according to their relationship with $\boldsymbol{s}_{\boldsymbol{\theta}}(\boldsymbol{x}_t, t)$.

# 4 Variance Controlled Diffusion SDEs

As mentioned in Section 1, most of the existing training-free efficient samplers are based on solving diffusion ODE (4), e.g., [23, 22, 24], because of their improved efficiency compared with the solvers of diffusion SDE (3). However, the empirical observations in [27, 22] exhibit that the quality of data generated by solving diffusion SDE outperforms diffusion ODE given sufficient computational budgets. For example, in [22], the diffusion ODE sampler DDIM [22] significantly improve the FID score of diffusion SDE sampler DDPM [2] (from 133.37 to 6.84) on CIFAR10 dataset [36] under 20 NFEs. However, under 1000 NFEs, the DDPM beats the DDIM in terms of FID score (3.17 v.s. 4.04). There may be a trade-off between stochasticity and efficiency. Thus, we conjecture that adding proper scale noise during the generating process may improve the quality of generated data with few NFEs.

In this section, we explore a family of variance-controlled diffusion SDEs, so that we can use proper noise scales during the sampling stage. Inspired by Proposition 1 in [28] and Eq. (6) in [27], we propose the following proposition to construct the aforementioned diffusion SDEs.

**Proposition 4.1.** *For any bounded measurable function $\tau(t) : [0, T] \rightarrow \mathbb{R}$, the following Reverse SDEs*

$$\mathrm{d}\boldsymbol{x}_t = \left[f(t)\boldsymbol{x}_t - \left(\frac{1 + \tau^2(t)}{2}\right)g^2(t)\nabla_{\boldsymbol{x}}\log p_t(\boldsymbol{x}_t)\right]\mathrm{d}t + \tau(t)g(t)\mathrm{d}\bar{\boldsymbol{w}}_t, \quad \boldsymbol{x}_T \sim p_T(\boldsymbol{x}_T) \tag{6}$$

*share the same marginal probability distributions with (4) and (3) .*

The proof can be found in Appendix A.1. The proposition indicates that by solving any of the diffusion SDEs in (6), we can sample from the target distribution. It is worth noting that the magnitude of noise varies with $\tau(t)$, and $\tau(t) = 0$ or $\tau(t) = 1$ respectively correspond to the diffusion ODE and SDE in [3]. Thus we can control the magnitude of added noise during the sampling process by varying it.

In practice, we numerically solve the diffusion SDEs (6) by substituting score function $\nabla_{\boldsymbol{x}} \log p_t(\boldsymbol{x}_t)$ in it with the "data prediction reparameterization model" $\boldsymbol{x}_{\boldsymbol{\theta}}(\boldsymbol{x}_t, t)$ according to $\nabla_{\boldsymbol{x}} \log p_t(\boldsymbol{x}_t) \approx -(\boldsymbol{x}_t - \alpha_t \boldsymbol{x}_{\boldsymbol{\theta}}(\boldsymbol{x}_t, t))/\sigma_t^2$ as pointed out in Section 3. Then diffusion SDEs to be solved become

$$d\boldsymbol{x}_t = \left[ f(t)\boldsymbol{x}_t + \left( \frac{1 + \tau^2(t)}{2\sigma_t} \right) g^2(t) \left( \frac{\boldsymbol{x}_t - \alpha_t \boldsymbol{x}_{\boldsymbol{\theta}}(\boldsymbol{x}_t, t)}{\sigma_t} \right) \right] dt + \tau(t)g(t)d\bar{\boldsymbol{w}}_t. \tag{7}$$

**Remark 1.** *We reparameterize the score function in diffusion SDEs (6) with data prediction model $\boldsymbol{x}_{\boldsymbol{\theta}}(\boldsymbol{x}_t, t)$ to get Eq. (9). The equation can be also reparameterized by the "noise prediction model" $\boldsymbol{\epsilon}_{\boldsymbol{\theta}}(\boldsymbol{x}_t, t)$ as discussed in Section 3. Though the obtained diffusion ODEs e.g., Eq. (9) are equivalent, the numerical solver applied to them will result in different solutions. For our proposed SA-Solver, we find the diffusion SDEs reparameterized data prediction model significantly improves the quality of generated data. More details and theoretical explanations are in Sec. 6 and Appendix A.2.4. For the remaining part of the paper, we focus on data reparameterization.*

We then solve the diffusion SDEs (9) with change-of-variable applying to it, i.e., changing time variable $t$ to log-SNR $\lambda_t = \log(\alpha_t/\sigma_t)$. Noting the following relationship in Eq. (2)

$$f(t) = \frac{\mathrm{d}\log\alpha_t}{\mathrm{d}t}, \quad g^2(t) = \frac{\mathrm{d}\sigma_t^2}{\mathrm{d}t} - 2\frac{\mathrm{d}\log\alpha_t}{\mathrm{d}t}\sigma_t^2 = -2\sigma_t^2\frac{\mathrm{d}\lambda_t}{\mathrm{d}t}, \tag{8}$$

and plugging them into (7), it becomes

$$\mathrm{d}\boldsymbol{x}_t = \left[ \frac{\mathrm{d}\log\alpha_t}{\mathrm{d}t}\boldsymbol{x}_t - (1 + \tau^2(t))(\boldsymbol{x}_t - \alpha_t \boldsymbol{x}_{\boldsymbol{\theta}}(\boldsymbol{x}_t, t))\frac{\mathrm{d}\lambda_t}{\mathrm{d}t} \right] \mathrm{d}t + \tau(t)\sigma_t\sqrt{-2\frac{\mathrm{d}\lambda_t}{\mathrm{d}t}}\mathrm{d}\bar{\boldsymbol{w}}_t. \tag{9}$$

The above equation has an explicit solution owing to its semi-linear structure [37].

**Proposition 4.2.** *Given $\boldsymbol{x}_s$ for any time $s > 0$, the solution $\boldsymbol{x}_t$ at time $t \in [0, s]$ of (9) is*

$$\boldsymbol{x}_t = \frac{\sigma_t}{\sigma_s}e^{-\int_{\lambda_s}^{\lambda_t}\tau^2(\tilde{\lambda})\mathrm{d}\tilde{\lambda}}\boldsymbol{x}_s + \sigma_t\boldsymbol{F}_{\boldsymbol{\theta}}(s, t) + \sigma_t\boldsymbol{G}(s, t),$$

$$\boldsymbol{F}_{\boldsymbol{\theta}}(s, t) = \int_{\lambda_s}^{\lambda_t}e^{-\int_{\lambda}^{\lambda_t}\tau^2(\tilde{\lambda})\mathrm{d}\tilde{\lambda}}\left(1 + \tau^2(\lambda)\right)e^{\lambda}\boldsymbol{x}_{\boldsymbol{\theta}}(\boldsymbol{x}_{\lambda}, \lambda)\,\mathrm{d}\lambda \tag{10}$$

$$\boldsymbol{G}(s, t) = \int_s^t e^{-\int_{\lambda_u}^{\lambda_t}\tau^2(\tilde{\lambda})\mathrm{d}\tilde{\lambda}}\tau(u)\sqrt{-2\frac{\mathrm{d}\lambda_u}{\mathrm{d}u}}\mathrm{d}\bar{\boldsymbol{w}}_u,$$

*where $\boldsymbol{G}(s, t)$ is an Itô integral [38] with the special property*

$$\sigma_t\boldsymbol{G}(s, t) \sim \mathcal{N}\left(\boldsymbol{0}, \sigma_t^2\left(1 - e^{-2\int_{\lambda_s}^{\lambda_t}\tau^2(\tilde{\lambda})\mathrm{d}\tilde{\lambda}}\right)\right). \tag{11}$$

The proof can be seen in Appendix A.2.2. With this proposition, we can sample from the diffusion model via numerically solving Eq. (10) starting from $\boldsymbol{x}_T$ approximated by a Gaussian distribution.

## 5  *SA-Solver*: Stochastic Adams Method to Solve Diffusion SDEs

Stochastic training-free samplers for solving diffusion SDEs have not been studied as systematically as their deterministic ODE counterparts. This stems from the inherent challenges associated with designing numerical schemes for SDEs compared to ODEs [39]. Existing stochastic sampling methods either use only variant of one-step discretization of diffusion SDEs [2, 26, 3], or are specifically designed sampling procedures for diffusion processes [27] which are not general purpose SDE solvers. Jolicoeur-Martineau *et al.* [25] uses stochastic Improved Euler's method [40] with adaptive step sizes. However, it still necessitates hundreds of steps to yield a high-quality sample. As observed by [25], off-the-shelf SDE solvers are generally ill-suited for diffusion models, often exhibiting inferior qualities or even failing to converge. We postulate that the current dearth of fast stochastic samplers is principally due to factor that existing methodologies predominantly tend to rely

**Algorithm 1** *SA-Solver*

---

**Require:** data prediction model $\boldsymbol{x_\theta}$, timesteps $\{t_i\}_{i=0}^M$, initial value $\boldsymbol{x}_{t_0}$, predictor step $s_p$, corrector step $s_c$, buffer $B$ to store former evaluation of $\boldsymbol{x_\theta}$, $\tau(t)$ to control variance.

1: $B \xleftarrow{\text{buffer}} \boldsymbol{x_\theta}(\boldsymbol{x}_{t_0}, t_0)$
2: **for** $i = 1$ to $max(s_p, s_c)$ **do**                                                       ▷ Warm-up
3:      sample $\boldsymbol{\xi} \sim \mathcal{N}(\mathbf{0}, \boldsymbol{I})$
4:      calculate steps for warm-up $s_p^m = min(i, s_p)$, $s_c^m = min(i, s_c)$
5:      $\boldsymbol{x}_{t_i}^p \leftarrow s_p^m$-step *SA-Predictor*$(\boldsymbol{x}_{t_{i-1}}, B, \boldsymbol{\xi})$ (Eq. (14))                ▷ Prediction Step
6:      $B \xleftarrow{\text{buffer}} \boldsymbol{x_\theta}(x_{t_i}^p, t_i)$                                           ▷ Evaluation Step
7:      $\boldsymbol{x}_{t_i} \leftarrow s_c^m$-step *SA-Corrector*$(\boldsymbol{x}_{t_i}^p, \boldsymbol{x}_{t_{i-1}}, B, \boldsymbol{\xi})$ (Eq. (17))         ▷ Correction Step
8: **for** $i = max(s_p, s_c) + 1$ to $M$ **do**
9:      sample $\boldsymbol{\xi} \sim \mathcal{N}(\mathbf{0}, \boldsymbol{I})$
10:     $\boldsymbol{x}_{t_i}^p \leftarrow s_p$-step *SA-Predictor*$(\boldsymbol{x}_{t_{i-1}}, B, \boldsymbol{\xi})$ (Eq. (14))                 ▷ Prediction Step
11:     $B \xleftarrow{\text{buffer}} \boldsymbol{x_\theta}(x_{t_i}^p, t_i)$                                          ▷ Evaluation Step
12:     $\boldsymbol{x}_{t_i} \leftarrow s_c$-step *SA-Corrector*$(\boldsymbol{x}_{t_i}^p, \boldsymbol{x}_{t_{i-1}}, B, \boldsymbol{\xi})$ (Eq. (17))          ▷ Correction Step
      **return** $\boldsymbol{x}_{t_M}$

---

on one-step discretization or its variants, or alternatively, on heuristic designs of stochastic samplers. To address this factor, we leverage advanced contemporary tools in numerical solutions for SDEs, specifically, *stochastic Adams methods* [29]. It necessitates fewer evaluations compared to Stochastic Runge-Kutta schemes, making it a more suitable choice for problems which are computationally expensive - a characteristic that diffusion sampling certainly exemplifies.

Next, we formally present our Stochastic Adams Solver (*SA-Solver*). To solve Eq. (9), we first take $M + 1$ time steps $\{t_i\}_{i=0}^M$ which is strictly decreased from $t_0 = T$ to $t_M = 0$.[3] Then we can iteratively obtain the $\boldsymbol{x}_{t_i}$ (so that $\boldsymbol{x}_0$ approximates the required data) by the following relationship.

$$\boldsymbol{x}_{t_{i+1}} = \frac{\sigma_{t_{i+1}}}{\sigma_{t_i}} e^{-\int_{\lambda_{t_i}}^{\lambda_{t_{i+1}}} \tau^2(\lambda_u) \mathrm{d}\lambda_u} \boldsymbol{x}_{t_i} + \sigma_{t_{i+1}} \boldsymbol{F_\theta}(t_i, t_{i+1}) + \sigma_{t_{i+1}} \boldsymbol{G}(t_i, t_{i+1}) \tag{12}$$

As pointed out in Proposition 4.2, the Itô integral term $\boldsymbol{G}(t_i, t_{i+1})$ in above equation follows a Gaussian that can be directly sampled so we need to solve the deterministic integral term $F_\theta(t_i, t_{i+1})$.

We further combine Eq. (12) with the predictor-corrector method, which is a widely used numerical method. It works in two main steps. First, a predictor step is taken to make an initial approximation of the solution. Second, a corrector step will refine the predictor's approximation by taking the predicted value into account. It has been proven successful in the wide application of numerical analysis [37]. Especially, there are some attempts to use the predictor-corrector method to help sample diffusion models [3, 32, 24]. In the subsequent Section 5.1 and Section 5.2, we will separately derive our *SA-Predictor* and *SA-Corrector* using Eq. (12). Our algorithm is outlined in Algorithm 1.

## 5.1 SA-Predictor

The fundamental idea behind stochastic Adams methods is to leverage preceding model evaluations like $\boldsymbol{x_\theta}(\boldsymbol{x}_{t_i}, t_i), \boldsymbol{x_\theta}(\boldsymbol{x}_{t_{i-1}}, t_{i-1}), \cdots, \boldsymbol{x_\theta}(\boldsymbol{x}_{t_{i-(s-1)}}, t_{i-(s-1)})$. These evaluations can be retained with negligible cost implications. Given these preceding model evaluations, a natural strategy for estimating $F_\theta(t_i, t_{i+1})$ involves the application of Lagrange interpolations [37] of these evaluations. Lagrange interpolation of $s$ points $\boldsymbol{x_\theta}(\boldsymbol{x}_{t_i}, t_i), \boldsymbol{x_\theta}(\boldsymbol{x}_{t_{i-1}}, t_{i-1}), \cdots, \boldsymbol{x_\theta}(\boldsymbol{x}_{t_{i-(s-1)}}, t_{i-(s-1)})$ is a polynomial $L(t)$ with degrees $s - 1$:

$$L(t) = \sum_{j=0}^{s-1} l_{i-j}(t) \boldsymbol{x_\theta}(\boldsymbol{x}_{t_{i-j}}, t_{i-j}), \tag{13}$$

where $l_{i-j}(t) : \mathbb{R} \to \mathbb{R}$ is the Lagrange basis. Lagrange interpolation is an excellent approximation of $\boldsymbol{x_\theta}(\boldsymbol{x}_t, t)$ with the special property: $L(t_{i-j}) = \boldsymbol{x_\theta}(\boldsymbol{x}_{t_{i-j}}, t_{i-j})$, $\quad \forall \, 0 \le j \le s - 1$. Thus a natural

---

[3]The diffusion SDEs (7) are reverse-time SDEs, so that the $t_i$ here is increased.

way to estimate $F_{\boldsymbol{\theta}}(t_i, t_{i+1})$ is to replace $\boldsymbol{x_\theta}(\boldsymbol{x}_{\lambda_u}, \lambda_u)$ with $L(\lambda)$, which is just a change-of-variable of $L(t)$. The formula for $s$-step *SA-Predictor* is then derived.

**$s$-step *SA-Predictor***    Given the initial value $\boldsymbol{x}_{t_i}$ at time $t_i$, a total of $s$ former model evaluations $\boldsymbol{x_\theta}(\boldsymbol{x}_{t_i}, t_i), \boldsymbol{x_\theta}(\boldsymbol{x}_{t_{i-1}}, t_{i-1}), \cdots, \boldsymbol{x_\theta}(\boldsymbol{x}_{t_{i-(s-1)}}, t_{i-(s-1)})$, our $s$-step *SA-Predictor* is defined as:

$$\boldsymbol{x}_{t_{i+1}} = \frac{\sigma_{t_{i+1}}}{\sigma_{t_i}} e^{-\int_{\lambda_{t_i}}^{\lambda_{t_{i+1}}} \tau^2(\tilde{\lambda})\mathrm{d}\tilde{\lambda}} \boldsymbol{x}_{t_i} + \sum_{j=0}^{s-1} b_{i-j} \boldsymbol{x_\theta}(\boldsymbol{x}_{t_{i-j}}, t_{i-j}) + \tilde{\sigma}_i \boldsymbol{\xi}, \quad \boldsymbol{\xi} \sim \mathcal{N}(\boldsymbol{0}, \boldsymbol{I}), \quad (14)$$

where $\tilde{\sigma}_i = \sigma_{t_{i+1}} \sqrt{1 - e^{-2\int_{\lambda_{t_i}}^{\lambda_{t_{i+1}}} \tau^2(\tilde{\lambda})\mathrm{d}\tilde{\lambda}}}$ according to Proposition 4.2 and $b_{i-j}$ is given by:

$$b_{i-j} = \sigma_{t_{i+1}} \int_{\lambda_{t_i}}^{\lambda_{t_{i+1}}} e^{-\int_\lambda^{\lambda_{t_{i+1}}} \tau^2(\tilde{\lambda})\mathrm{d}\tilde{\lambda}} \left(1 + \tau^2(\lambda)\right) e^\lambda l_{i-j}(\lambda)\mathrm{d}\lambda, \quad \forall\, 0 \le j \le s-1 \quad (15)$$

We show the convergence result in the following theorem. The proof can be found in Appendix B.

**Theorem 5.1** (Strong Convergence of $s$-step *SA-Predictor*). *Under mild regularity conditions, our $s$-step SA-Predictor (Eq. (14)) has a global error in strong convergence sense of $\mathcal{O}(\sup_{0 \le t \le T} \tau(t)h + h^s)$, where $h = \max_{1 \le i \le M} (t_i - t_{i-1})$.*

## 5.2   SA-Corrector

Eq. (14) offers a "prediction" $\boldsymbol{x}_{t_{i+1}}^p$ that relies on information preceding or coinciding with the time step $t_i$ since we only use $\boldsymbol{x_\theta}(\boldsymbol{x}_{t_i}, t_i)$ along with other model evaluations antecedent to it, while the integration is over time $[t_i, t_{i+1}]$. Then predictor-corrector method can be incorporated to better estimate $F_{\boldsymbol{\theta}}(t_i, t_{i+1})$ in Eq. (12). We perform a model evaluation $\boldsymbol{x_\theta}(\boldsymbol{x}_{t_{i+1}}^p, t_{i+1})$ and construct the Lagrange interpolations of $\boldsymbol{x_\theta}(\boldsymbol{x}_{t_{i+1}}^p, t_{i+1}), \boldsymbol{x_\theta}(\boldsymbol{x}_{t_i}, t_i), \cdots, \boldsymbol{x_\theta}(\boldsymbol{x}_{t_{i-(s'-1)}}, t_{i-(\hat{s}-1)})$:

$$\hat{L}(t) = \hat{l}_{i+1}(t) \boldsymbol{x_\theta}(\boldsymbol{x}_{t_{i+1}}^p, t_{i+1}) + \sum_{j=0}^{\hat{s}-1} \hat{l}_{i-j}(t) \boldsymbol{x_\theta}(\boldsymbol{x}_{t_{i-j}}, t_{i-j}), \quad (16)$$

where $\hat{l}_{i-j}(t) : \mathbb{R} \to \mathbb{R}$ is the Lagrange basis and $\hat{s}$ can be different with $s$ in Eq. (13). The $\hat{s}$-step *SA-Corrector* is derived by replacing $\boldsymbol{x_\theta}(\boldsymbol{x}_{\lambda_u}, \lambda_u)$ with $\hat{L}(\lambda)$ which is a change-of-variable of $\hat{L}(t)$.

**$\hat{s}$-step *SA-Corrector***    Given the initial value $\boldsymbol{x}_{t_i}$ at time $t_i$, a total of $\hat{s}$ former model evaluations $\boldsymbol{x_\theta}(\boldsymbol{x}_{t_i}, t_i), \boldsymbol{x_\theta}(\boldsymbol{x}_{t_{i-1}}, t_{i-1}), \cdots, \boldsymbol{x_\theta}(\boldsymbol{x}_{t_{i-(\hat{s}-1)}}, t_{i-(\hat{s}-1)})$, model evaluation of "prediction" $\boldsymbol{x_\theta}(\boldsymbol{x}_{t_{i+1}}^p, t_{i+1})$, our $\hat{s}$-step *SA-Corrector* is defined as:

$$\boldsymbol{x}_{t_{i+1}} = \frac{\sigma_{t_{i+1}}}{\sigma_{t_i}} e^{-\int_{\lambda_{t_i}}^{\lambda_{t_{i+1}}} \tau^2(\tilde{\lambda})\mathrm{d}\tilde{\lambda}} \boldsymbol{x}_{t_i} + \hat{b}_{i+1} \boldsymbol{x_\theta}(\boldsymbol{x}_{t_{i+1}}^p, t_{i+1}) + \sum_{j=0}^{\hat{s}-1} \hat{b}_{i-j} \boldsymbol{x_\theta}(\boldsymbol{x}_{t_{i-j}}, t_{i-j}) + \tilde{\sigma}_i \boldsymbol{\xi}, \quad (17)$$

where $\boldsymbol{\xi} \sim \mathcal{N}(\boldsymbol{0}, \boldsymbol{I})$, $\tilde{\sigma}_i = \sigma_{t_{i+1}} \sqrt{1 - e^{-2\int_{\lambda_{t_i}}^{\lambda_{t_{i+1}}} \tau^2(\tilde{\lambda})\mathrm{d}\tilde{\lambda}}}$ according to Proposition 4.2 and the coefficients $\hat{b}_{i+1}, \hat{b}_{i-j}$ is given by:

$$\hat{b}_{i-j} = \sigma_{t_{i+1}} \int_{\lambda_{t_i}}^{\lambda_{t_{i+1}}} e^{-\int_\lambda^{\lambda_{t_{i+1}}} \tau^2(\tilde{\lambda})\mathrm{d}\tilde{\lambda}} \left(1 + \tau^2(\lambda)\right) e^\lambda \hat{l}_{i-j}(\lambda)\mathrm{d}\lambda, \quad \forall\, 0 \le j \le s-1$$
$$\hat{b}_{i+1} = \sigma_{t_{i+1}} \int_{\lambda_{t_i}}^{\lambda_{t_{i+1}}} e^{-\int_\lambda^{\lambda_{t_{i+1}}} \tau^2(\tilde{\lambda})\mathrm{d}\tilde{\lambda}} \left(1 + \tau^2(\lambda)\right) e^\lambda \hat{l}_{i+1}(\lambda)\mathrm{d}\lambda$$
$$(18)$$

We show the convergence result in the following theorem. The proof can be found in Appendix B.

**Theorem 5.2** (Strong Convergence of $\hat{s}$-step *SA-Corrector*). *Under mild regularity conditions, our $\hat{s}$-step SA-Corrector (Eq. (17)) has a global error in strong convergence sense of $\mathcal{O}(\sup_{0 \le t \le T} \tau(t)h + h^{\hat{s}+1})$, where $h = \max_{1 \le i \le M} (t_i - t_{i-1})$.*

### 5.3 Connection with other samplers

We briefly discuss the relationship between our *SA-Solver* and other existing solvers for sampling diffusion ODEs or diffusion SDEs.

**Relationship with DDIM [22]**    DDIM generate samples through the following process:

$$\boldsymbol{x}_{t_{i+1}} = \alpha_{t_{i+1}} \left( \frac{\boldsymbol{x}_{t_i} - \sigma_{t_i} \boldsymbol{\epsilon_\theta}(\boldsymbol{x}_{t_i}, t_i)}{\alpha_{t_i}} \right) + \sqrt{1 - \alpha_{t_{i+1}}^2 - \hat{\sigma}_{t_i}^2} \boldsymbol{\epsilon_\theta}(\boldsymbol{x}_{t_i}, t_i) + \hat{\sigma}_{t_i} \boldsymbol{\xi}, \qquad (19)$$

where $\boldsymbol{\xi} \sim \mathcal{N}(\boldsymbol{0}, \boldsymbol{I})$, $\hat{\sigma}_{t_i}$ is a variable parameter. In practice, DDIM introduces a parameter $\eta$ such that when $\eta = 0$, the sampling process becomes deterministic and when $\eta = 1$, the sampling process coincides with original DDPM [2]. Specifically, $\hat{\sigma}_{t_i} = \eta \sqrt{\frac{1 - \alpha_{t_{i+1}}^2}{1 - \alpha_{t_i}^2} \left( 1 - \frac{\alpha_{t_i}^2}{\alpha_{t_{i+1}}^2} \right)}$.

**Corollary 5.3** (Relationship with DDIM). *For any $\eta$ in DDIM, there exists a $\tau_\eta(t) : \mathbb{R} \rightarrow \mathbb{R}$ which is a piecewise constant function such that DDIM-$\eta$ coincides with our 1-step SA-Predictor when $\tau(t) = \tau_\eta(t)$ with data parameterization of our variance-controlled diffusion SDE.*

The proof can be found in Appendix B.5.1.

**Relationship with DPM-Solver++(2M) [31]**    DPM-Solver++ is a high-order solver which solves diffusion ODEs for guided sampling. DPM-Solver++(2M) is a special case of our 2-step *SA-Predictor* when $\tau(t) \equiv 0$.

**Relationship with UniPC [24]**    UniPC is a unified predictor-corrector framework for solving diffusion ODEs. UniPC-p is a special case of our *SA-Solver* when $\tau(t) \equiv 0$ with predictor step $p$, corrector step $p$ in Algorithm 1.

## 6 Experiments

In this section, we demonstrate the effectiveness of *SA-Solver* over the existing sampling methods on both a small number of function evaluations (NFEs) settings and a considerable number of NFEs settings, with extensive experiments. We use Fenchel Inception Distance (FID) [30] as the evaluation metric to show the effectiveness of *SA-Solver*. Unless otherwise specified, 50K images are sampled for evaluation. The experiments are conducted on various datasets, with image sizes ranging from 32x32 to 256x256. We also evaluate the performance of various models, including ADM [4], EDM [27], Latent Diffusion [5], and DiT [41].

For ease of computation, we take $\tau(t) \equiv \tau$ as a constant function or a piecewise constant function. We leave the detailed settings for $\tau(t)$, predictor step, and corrector step in Appendix E. For the following experiments, we first discuss the effectiveness of the data-prediction model. Then we evaluate the performance of *SA-Solver* under different random noise scales $\tau$ to demonstrate the principles for selecting $\tau$ under few-steps and a considerable number of steps. Finally, we compare *SA-Solver* with the existing solver to demonstrate its effectiveness.

### 6.1 Comparison between Data-Prediction Model and Noise-Prediction Model

We first discuss the necessity of using a data-prediction model for *SA-Solver*. We test on ImageNet 256x256 (latent diffusion model) with $\tau(t) \equiv 1$. Results of the data-prediction and noise-prediction model are shown in Table 1. It can be seen that the data-prediction model can achieve better sampling quality values under different NFEs, thus we use the data-prediction model in the rest of the experiments. More detailed discussions and theoretical analysis can be seen in Appendix A.2.4.

### 6.2 Ablation Study on Predictor/Corrector Steps and Predictor-Corrector Method

To verify the effectiveness of our proposed Stochastic Linear Multi-step Methods and Predictor-Corrector Method, we conduct an ablation study on the CIFAR10 dataset as follows. We use EDM [27] baseline-VE pretrained checkpoint. Concretely, we vary the number of predictor steps

Table 1: Compared results by FID ↓ under data-prediction and noise-prediction models, measured by different NFEs. The latent diffusion model in ImageNet 256x256 is used for evaluation.

| NFEs | Noise-prediction | Data-prediction |
|------|------------------|-----------------|
| 20 | 310.5 | **3.88** |
| 40 | 5.85 | **3.47** |
| 60 | 3.54 | **3.41** |
| 80 | 3.41 | **3.38** |

Table 2: Compared results by FID ↓ under different predictor steps and corrector steps, measured by different NFEs. The VE-baseline model [27] in CIFAR10 32x32 is used for evaluation.

| method \ setting (NFE, $\tau$) | 15,0.4 | 23,0.8 | 31,1.0 | 47,1.4 |
|--------------------------------|--------|--------|--------|--------|
| Predictor 1-steps only | 13.76 | 12.44 | 11.72 | 14.67 |
| Predictor 1-steps, Corrector 1-step | 8.49 | 6.87 | 6.13 | 6.75 |
| Predictor 3-steps only | 5.30 | 3.93 | 3.52 | 2.98 |
| Predictor 3-steps, Corrector 3-steps | **4.91** | **3.77** | **3.40** | **2.92** |

and meanwhile conduct them with/without corrector to separately explore the effect of the two components. As can be seen in Table 2, both Stochastic Linear Multi-step Methods (Predictor 1-steps only v.s. Predictor 3-steps only) and Predictor-Corrector Method (Predictor 1-steps only v.s. Predictor 1-steps, Corrector 1-step, and Predictor 3-steps only v.s. Predictor 3-steps, Corrector 3-steps) improve the performance of our sampler.

## 6.3 Effect on Magnitude of Stochasticity

The proposed *SA-Solver* is evaluated on various types of datasets and models, including ImageNet 256x256 [43] (latent diffusion model [5]), LSUN Bedroom 256x256 [44] (pixel diffusion model [4]), ImageNet 64x64 (pixel diffusion model [4]), and CIFAR10 32x32 (pixel diffusion model [27]). The models corresponding to these datasets cover pixel-space and latent-space diffusion models, with unconditional, conditional, and classifier-free guidance settings ($s = 1.5$ in ImageNet 256x256).

We used different constant $\tau$ values for *SA-Solver*, namely $\{0.0, 0.2, 0.4, ..., 1.6\}$, where larger value of $\tau$ correspond to larger magnitude of stochasticity. The FID results under different NFE and $\tau$ values are shown in Fig. 1. Note that for LSUN Bedroom, 10K images are sampled for evaluation. The experiments indicate that (1) under relatively small NFEs, smaller nonzero $\tau$ values can achieve better FID results; (2) under a considerable number of steps (20-100), large $\tau$ can achieve better FID. This phenomenon is consistent with the theoretical analysis we conducted in Appendix B and Appendix C, in which the sampling error with stochasticity is dominated under small NFE, while larger $\tau$ can significantly improve the quality of generated samples as the number of steps increases. In subsequent experiments, unless otherwise specified, we will report the results of a proper $\tau(t)$ value. Details can be found in Appendix E.

## 6.4 Comparison with State-of-the-Art

We compare *SA-Solver* with existing state-of-the-art sampling methods, including DDIM [22], DPM-Solver [23], UniPC [24], Heun sampler and stochastic sampler in EDM [27]. Unless otherwise specified, the methods are tested using the default hyper-parameters in the original papers or code.

**Results on CIFAR10 32x32 and ImageNet 64x64.** We use the EDM [27] baseline-VE model for the CIFAR10 32x32 experiments and the ADM [4] model for the ImageNet 64x64 experiments. We use EDM's timesteps selection for all samplers for fair comparisons. EDM introduces a certain type of SDE and a corresponding stochastic sampler, which is used for comparison. The experimental results are shown in Fig. 2(a-b). It can be seen that the proposed *SA-Solver* consistently outperforms other samplers and achieves state-of-the-art FID results. It should be noticed for EDM samplers, we report its optimal result which is searched over four hyper-parameters. In fact, at 95 NFEs, *SA-Solver* can achieve the best FID value of 2.63 in CIFAR10 and 1.81 in ImageNet 64x64 which outperforms all other samplers.

Table 3: Sampling quality measured by FID of different sampling methods on DiT, Min-SNR ImageNet [41, 42] models. DiT-XL/2-G and ViT-XL-patch2-32 with $s = 1.5$ are used.

| Model | FID ($\downarrow$) | |
|---|---|---|
| DiT ImageNet 256x256 | DDPM (NFE=250) | **SA-Solver (Ours)** (NFE=60) |
| | 2.27 | **2.02** |
| Min-SNR ImageNet 256x256 | Heun (NFE=50) | **SA-Solver (Ours)** (NFE=20) |
| | 2.06 | **1.93** |
| DiT ImageNet 512x512 | DDPM (NFE=250) | **SA-Solver (Ours)** (NFE=60) |
| | 3.04 | **2.80** |

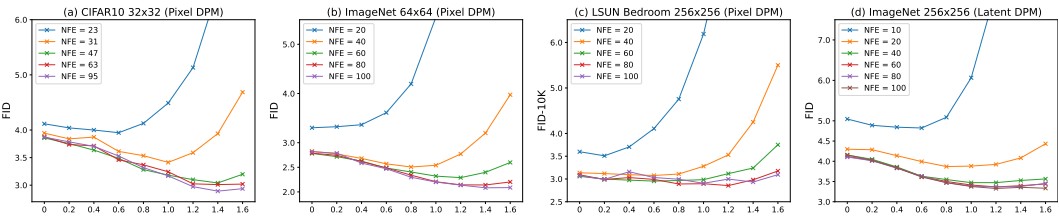

Figure 1: Sampling quality measured by FID $\downarrow$ of *SA-Solver* under a different number of function evaluations (NFE), varying the stochastic noise scale $\tau$. For LSUN Bedroom, 10K samples are used to evaluate FID.

**Results on ImageNet 256x256 and 512x512.** We evaluate with two classifier-free guidance models, one is the UNet-based latent diffusion model [5] in which the VQ-4 encoder-decoder model is adopted, and the other is the DiT [41] model using Vision Transformer based model with KL-8 encoder-decoder. The corresponding classifier-free guidance scale, namely $s = 1.5$, is adopted for evaluation. For ImageNet 256x256 dataset with UNet based latent diffusion model, the results of different samplers are shown in Fig. 2(c). Under a considerable number of steps, *SA-Solver* achieves the best sample quality, in which the FID value is 3.87 with only 20 NFEs and below 3.5 with 40 NFEs or more. While for ODE solvers, the FID values cannot reach below 4, which shows the superiority of the proposed SDE solver.

Table 3 consists results of current SOTA models in ImageNet 256x256 and 512x512. Note that the (Min-SNR) DiT-XL/2-G models are adopted [41, 42]. It can be seen clearly that better FID results are achieved compared with baseline solvers used by corresponding methods. We achieve 1.93 FID value in Min-SNR DiT model at ImageNet 256x256, and 2.80 in DiT model at ImageNet 512x512, both of which are state-of-the-art results under existing DPMs.

**Results of text-to-image generation** Fig. 3 shows the qualitative results on text-to-image generation. It can be seen that both UniPC and *SA-Solver* can generate images with more details. Our *SA-Solver* is able to generate more reasonable images with better details.

### 6.5 Effect of Stochasticity for Inaccurate Score Estimation

When the training data is not enough or the computational budget is limited, the estimated score is inaccurate. We empirically observed that the stochasticity significantly improve the sample quality under the circumstance. To further investigate this effect, we reproduce the early training stage of EDM [27] baseline-VE model for the CIFAR10 32x32 dataset and DiT-XL/2 [41] model for the ImageNet 256x256 dataset. We compare *SA-Solver* with different stochastic level $\tau$ and existing state-of-the-art deterministic sampling methods. We use the same hyper-parameters as the corresponding experiment in section 6.4.

Figure 4 shows that *SA-Solver* outperforms deterministic sampling methods, especially in the early stage of the training process. Moreover, larger $\tau$ value results in better performance. We also conduct a theoretical analysis that stochasticity can mitigate the error of estimation (see Appendix C).

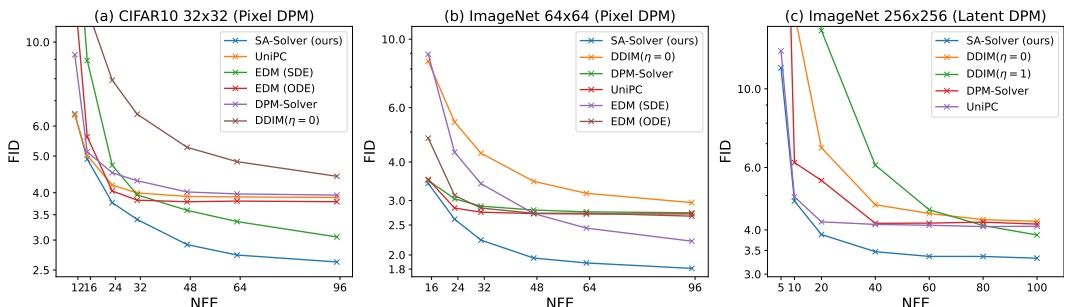

Figure 2: Sampling quality measured by FID ↓ of different sampling methods of DPMs under different NFEs.

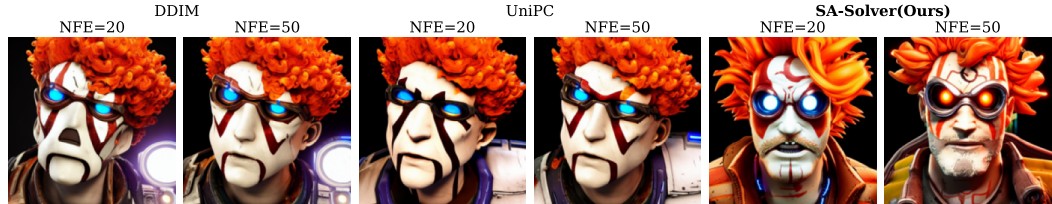

Figure 3: Qualitative comparisons between our *SA-Solver* and previous state-of-the-art methods. All images are generated by Stable Diffusion v1.5 with the same random seed. The main part of the prompt is "portrait of curly orange haired mad scientist man". We set the guidance scale as 7.5. The proposed *SA-Solver* is able to generate images with more details.

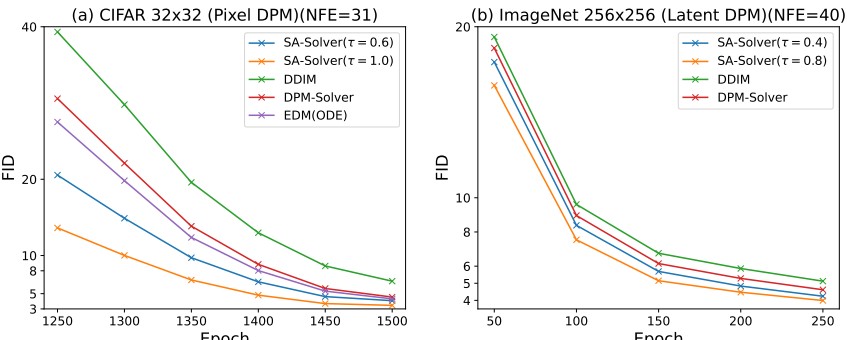

Figure 4: Sampling quality measured by FID ↓ of different sampling methods of DPMs under different training epochs.

# 7 Conclusions

In this paper, we propose an efficient solver named *SA-Solver* for solving Diffusion SDEs, achieving high sampling performance in both minimal steps and a suitable number of steps. To better control the scale of injected noise, we propose Variance Controlled Diffusion SDEs based on noise scale function $\tau(t)$ and propose the analytic form of the SDEs. Based on Variance Controlled Diffusion SDE, we propose *SA-Solver*, which is derived from the stochastic Adams method and uses exponentially weighted integral and analytical variance to achieve efficient SDE sampling. Meanwhile, *SA-Solver* has the optimal theoretical convergence bound. Experiments show that *SA-Solver* achieves state-of-the-art sampling performance in various pre-trained DPMs models. Moreover, *SA-Solver* achieves superior performance when the score estimation is inaccurate.

Although *SA-Solver* achieves optimal sampling performance, the noise scale $\tau(t)$ selection under different NFEs needs further research. The paper proposes empirical criteria for selecting $\tau(t)$, more in-depth theoretical analysis is still needed.

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

## A Derivations of Variance Controlled Diffusion SDEs

### A.1 Proof of Proposition 4.1

**Proposition 4.1** For any bounded measurable function $\tau(t) : [0, T] \to \mathbb{R}$, the following Reverse SDEs

$$\mathrm{d}\boldsymbol{x}_t = \left[ f(t)\boldsymbol{x}_t - \left( \frac{1 + \tau^2(t)}{2} \right) g^2(t)\nabla_{\boldsymbol{x}} \log p_t(\boldsymbol{x}_t) \right] \mathrm{d}t + \tau(t)g(t)\mathrm{d}\bar{\boldsymbol{w}}_t, \quad \boldsymbol{x}_T \sim p_T(\boldsymbol{x}_T) \quad (20)$$

has the same marginal probability distributions with (4) and (3) .

*Proof.* Denote $p(\boldsymbol{x}, t) : \mathbb{R}^d \times [0, T] \to \mathbb{R}^+$ as the probability density function of $\boldsymbol{x}_t$ at time $t$, thus $p(\boldsymbol{x}, T) = p_T(\boldsymbol{x})$. Fokker-Planck equation [38] determines a Partial Differential Equation (PDE) that $p_t(\boldsymbol{x})$ satisfies:

$$
\begin{aligned}
-\frac{\partial p_t(\boldsymbol{x})}{\partial t} &= -\sum_{i=1}^{d} \frac{\partial}{\partial x_i} \left[ -\left[ f(t)p_t(\boldsymbol{x})x_i - \left( \frac{1 + \tau^2(t)}{2} \right) g^2(t)p_t(\boldsymbol{x})\frac{\partial \log p_t(\boldsymbol{x})}{\partial x_i} \right] \right] \\
&\quad + \frac{1}{2} \sum_{i=1}^{d} \sum_{j=1}^{d} \frac{\partial^2}{\partial x_i \partial x_j} \left[ \tau^2(t)g^2(t)\delta_{ij}p_t(\boldsymbol{x}) \right] \\
&= \sum_{i=1}^{d} \frac{\partial}{\partial x_i} \left[ f(t)p_t(\boldsymbol{x})x_i - \left( \frac{1 + \tau^2(t)}{2} \right) g^2(t)p_t(\boldsymbol{x})\frac{\partial \log p_t(\boldsymbol{x})}{\partial x_i} \right] \\
&\quad + \frac{1}{2} \sum_{i=1}^{d} \frac{\partial}{\partial x_i} \left[ \frac{\partial}{\partial x_i} \left[ \tau^2(t)g^2(t)p_t(\boldsymbol{x}) \right] \right].
\end{aligned}
\quad (21)
$$

Eq. (20) is a reverse-time SDE running from $T$ to $0$, thus there are two additional minus signs in Eq. (21) before term $\frac{\partial p_t(\boldsymbol{x})}{\partial t}$ and term $\left[ f(t)p_t(\boldsymbol{x})x_i - \left( \frac{1+\tau^2(t)}{2} \right) g^2(t)p_t(\boldsymbol{x})\frac{\partial \log p_t(\boldsymbol{x})}{\partial x_i} \right]$ compared with vanilla Fokker-Planck equation in general cases. Here $\delta_{ij}$ is the Dirac symbol satisfies $\delta_{ij} = 1$ when $i = j$, otherwise, $\delta_{ij} = 0$. Notice that

$$\frac{\partial}{\partial x_i} \left[ \tau^2(t)g^2(t)p_t(\boldsymbol{x}) \right] = \tau^2(t)g^2(t)\frac{\partial}{\partial x_i}p_t(\boldsymbol{x}) = \tau^2(t)g^2(t)p_t(\boldsymbol{x})\frac{\partial \log p_t(\boldsymbol{x})}{\partial x_i}. \quad (22)$$

Substituting Eq. (22) into Eq. (21), we obtain that

$$
\begin{aligned}
-\frac{\partial p_t(\boldsymbol{x})}{\partial t} &= \sum_{i=1}^{d} \frac{\partial}{\partial x_i} \left[ f(t)p_t(\boldsymbol{x})x_i - \left( \frac{1 + \tau^2(t)}{2} \right) g^2(t)p_t(\boldsymbol{x})\frac{\partial \log p_t(\boldsymbol{x})}{\partial x_i} \right] \\
&\quad + \frac{1}{2} \sum_{i=1}^{d} \frac{\partial}{\partial x_i} \left[ \tau^2(t)g^2(t)p_t(\boldsymbol{x})\frac{\partial \log p_t(\boldsymbol{x})}{\partial x_i} \right] \\
&= \sum_{i=1}^{d} \frac{\partial}{\partial x_i} \left[ f(t)p_t(\boldsymbol{x})x_i - \frac{1}{2}g^2(t)p_t(\boldsymbol{x})\frac{\partial \log p_t(\boldsymbol{x})}{\partial x_i} \right],
\end{aligned}
\quad (23)
$$

which is independent of $\tau(t)$. With the same initial condition $p(\boldsymbol{x}, T) = p_T(\boldsymbol{x})$, the family of Reverse SDEs in Eq. (20) have exactly the same evolutions of probability density function because they share the same Fokker-Planck equation. Especially, when $\tau(t) = 0$, Eq. (20) degenerates to diffusion ODEs and when $\tau(t) = 1$, Eq. (20) degenerates to diffusion SDEs. □

### A.2 Two Reparameterizations and Exact Solution under Exponential Integrator

In this subsection, we will show the exact solution of SDE in both *data prediction* reparameterization and *noise prediction* reparameterization. The noise term in *data prediction* has smaller variance than *noise prediction* ones, implying the necessity of adopting *data prediction* reparameterization for the SDE sampler.

### A.2.1 Data Prediction Reparameterization

After approximating $\nabla_{\boldsymbol{x}} \log p_t(\boldsymbol{x}_t)$ with $\boldsymbol{s}_{\boldsymbol{\theta}}(\boldsymbol{x}_t, t)$ and reparameterizing $\boldsymbol{s}_{\boldsymbol{\theta}}(\boldsymbol{x}_t, t)$ with $-(\boldsymbol{x}_t - \alpha_t \boldsymbol{x}_{\boldsymbol{\theta}}(\boldsymbol{x}_t, t))/\sigma_t^2$, Eq. (20) becomes

$$\mathrm{d}\boldsymbol{x}_t = \left[ f(t)\boldsymbol{x}_t + \left( \frac{1 + \tau^2(t)}{2\sigma_t} \right) g^2(t) \left( \frac{\boldsymbol{x}_t - \alpha_t \boldsymbol{x}_{\boldsymbol{\theta}}(\boldsymbol{x}_t, t)}{\sigma_t} \right) \right] \mathrm{d}t + \tau(t) g(t) \mathrm{d}\bar{\boldsymbol{w}}_t. \qquad (24)$$

Applying change-of-variable with log-SNR $\lambda_t = \log(\alpha_t/\sigma_t)$ and substituting the following relationship

$$f(t) = \frac{\mathrm{d}\log\alpha_t}{\mathrm{d}t}, \quad g^2(t) = \frac{\mathrm{d}\sigma_t^2}{\mathrm{d}t} - 2\frac{\mathrm{d}\log\alpha_t}{\mathrm{d}t}\sigma_t^2 = -2\sigma_t^2\frac{\mathrm{d}\lambda_t}{\mathrm{d}t}, \qquad (25)$$

Eq. (24) becomes

$$
\begin{aligned}
\mathrm{d}\boldsymbol{x}_t &= \left[ \frac{\mathrm{d}\log\alpha_t}{\mathrm{d}t}\boldsymbol{x}_t - \left(1 + \tau^2(t)\right)\left(\boldsymbol{x}_t - \alpha_t \boldsymbol{x}_{\boldsymbol{\theta}}(\boldsymbol{x}_t, t)\right)\frac{\mathrm{d}\lambda_t}{\mathrm{d}t} \right] \mathrm{d}t + \tau(t)\sigma_t\sqrt{-2\frac{\mathrm{d}\lambda_t}{\mathrm{d}t}}\mathrm{d}\bar{\boldsymbol{w}}_t \\
&= \left[ \left( \frac{\mathrm{d}\log\alpha_t}{\mathrm{d}t} - \left(1 + \tau^2(t)\right)\frac{\mathrm{d}\lambda_t}{\mathrm{d}t} \right)\boldsymbol{x}_t + \left(1 + \tau^2(t)\right)\alpha_t\boldsymbol{x}_{\boldsymbol{\theta}}(\boldsymbol{x}_t, t)\frac{\mathrm{d}\lambda_t}{\mathrm{d}t} \right] \mathrm{d}t \\
&\quad + \tau(t)\sigma_t\sqrt{-2\frac{\mathrm{d}\lambda_t}{\mathrm{d}t}}\mathrm{d}\bar{\boldsymbol{w}}_t.
\end{aligned}
\qquad (26)
$$

### A.2.2 Proof of Proposition 4.2

**Proposition 4.2** Given $\boldsymbol{x}_s$ for any time $s > 0$, the solution $\boldsymbol{x}_t$ at time $t \in [0, s]$ of Eq. (9) is

$$
\begin{aligned}
\boldsymbol{x}_t &= \frac{\sigma_t}{\sigma_s}e^{-\int_{\lambda_s}^{\lambda_t}\tau^2(\tilde{\lambda})\mathrm{d}\tilde{\lambda}}\boldsymbol{x}_s + \sigma_t\boldsymbol{F}_{\boldsymbol{\theta}}(s, t) + \sigma_t\boldsymbol{G}(s, t), \\
\boldsymbol{F}_{\boldsymbol{\theta}}(s, t) &= \int_{\lambda_s}^{\lambda_t} e^{-\int_{\lambda}^{\lambda_t}\tau^2(\tilde{\lambda})\mathrm{d}\tilde{\lambda}}\left(1 + \tau^2(\lambda)\right)e^{\lambda}\boldsymbol{x}_{\boldsymbol{\theta}}(\boldsymbol{x}_{\lambda}, \lambda)\mathrm{d}\lambda \\
\boldsymbol{G}(s, t) &= \int_s^t e^{-\int_{\lambda_u}^{\lambda_t}\tau^2(\tilde{\lambda})\mathrm{d}\tilde{\lambda}}\tau(u)\sqrt{-2\frac{\mathrm{d}\lambda_u}{\mathrm{d}u}}\mathrm{d}\bar{\boldsymbol{w}}_u,
\end{aligned}
\qquad (27)
$$

where $\boldsymbol{G}(s, t)$ is an *Itô* integral [38] with the special property

$$\sigma_t\boldsymbol{G}(s, t) \sim \mathcal{N}\left(\boldsymbol{0}, \sigma_t^2\left(1 - e^{-2\int_{\lambda_s}^{\lambda_t}\tau^2(\tilde{\lambda})\mathrm{d}\tilde{\lambda}}\right)\right). \qquad (28)$$

*Proof.* Define $\boldsymbol{y}_t = e^{-\int_{t_0}^t \left(\frac{\mathrm{d}\log\alpha_v}{\mathrm{d}v} - (1+\tau^2(v))\frac{\mathrm{d}\lambda_v}{\mathrm{d}v}\right)\mathrm{d}v}\boldsymbol{x}_t$, where $t_0 \in [0, T]$ is a constant. Differentiate $\boldsymbol{y}_t$ with respect to $t$, we get

$$
\begin{aligned}
\mathrm{d}\boldsymbol{y}_t &= -\left( \frac{\mathrm{d}\log\alpha_t}{\mathrm{d}t} - \left(1 + \tau^2(t)\right)\frac{\mathrm{d}\lambda_t}{\mathrm{d}t} \right)e^{-\int_{t_0}^t\left(\frac{\mathrm{d}\log\alpha_v}{\mathrm{d}v} - (1+\tau^2(v))\frac{\mathrm{d}\lambda_v}{\mathrm{d}v}\right)\mathrm{d}v}\boldsymbol{x}_t\mathrm{d}t \\
&\quad + e^{-\int_{t_0}^t\left(\frac{\mathrm{d}\log\alpha_v}{\mathrm{d}v} - (1+\tau^2(v))\frac{\mathrm{d}\lambda_v}{\mathrm{d}v}\right)\mathrm{d}v}\mathrm{d}\boldsymbol{x}_t \\
&= -\left( \frac{\mathrm{d}\log\alpha_t}{\mathrm{d}t} - \left(1 + \tau^2(t)\right)\frac{\mathrm{d}\lambda_t}{\mathrm{d}t} \right)e^{-\int_{t_0}^t\left(\frac{\mathrm{d}\log\alpha_v}{\mathrm{d}v} - (1+\tau^2(v))\frac{\mathrm{d}\lambda_v}{\mathrm{d}v}\right)\mathrm{d}v}\boldsymbol{x}_t\mathrm{d}t \\
&\quad + e^{-\int_{t_0}^t\left(\frac{\mathrm{d}\log\alpha_v}{\mathrm{d}v} - (1+\tau^2(v))\frac{\mathrm{d}\lambda_v}{\mathrm{d}v}\right)\mathrm{d}v}\left[ \left( \frac{\mathrm{d}\log\alpha_t}{\mathrm{d}t} - \left(1 + \tau^2(t)\right)\frac{\mathrm{d}\lambda_t}{\mathrm{d}t} \right)\boldsymbol{x}_t \right]\mathrm{d}t \\
&\quad + e^{-\int_{t_0}^t\left(\frac{\mathrm{d}\log\alpha_v}{\mathrm{d}v} - (1+\tau^2(v))\frac{\mathrm{d}\lambda_v}{\mathrm{d}v}\right)\mathrm{d}v}\left[ \left(1 + \tau^2(t)\right)\alpha_t\boldsymbol{x}_{\boldsymbol{\theta}}(\boldsymbol{x}_t, t)\frac{\mathrm{d}\lambda_t}{\mathrm{d}t} \right]\mathrm{d}t \\
&\quad + e^{-\int_{t_0}^t\left(\frac{\mathrm{d}\log\alpha_v}{\mathrm{d}v} - (1+\tau^2(v))\frac{\mathrm{d}\lambda_v}{\mathrm{d}v}\right)\mathrm{d}v}\tau(t)\sigma_t\sqrt{-2\frac{\mathrm{d}\lambda_t}{\mathrm{d}t}}\mathrm{d}\bar{\boldsymbol{w}}_t \\
&= e^{-\int_{t_0}^t\left(\frac{\mathrm{d}\log\alpha_v}{\mathrm{d}v} - (1+\tau^2(v))\frac{\mathrm{d}\lambda_v}{\mathrm{d}v}\right)\mathrm{d}v}\left[ \left(1 + \tau^2(t)\right)\alpha_t\boldsymbol{x}_{\boldsymbol{\theta}}(\boldsymbol{x}_t, t)\frac{\mathrm{d}\lambda_t}{\mathrm{d}t} \right]\mathrm{d}t \\
&\quad + e^{-\int_{t_0}^t\left(\frac{\mathrm{d}\log\alpha_v}{\mathrm{d}v} - (1+\tau^2(v))\frac{\mathrm{d}\lambda_v}{\mathrm{d}v}\right)\mathrm{d}v}\tau(t)\sigma_t\sqrt{-2\frac{\mathrm{d}\lambda_t}{\mathrm{d}t}}\mathrm{d}\bar{\boldsymbol{w}}_t.
\end{aligned}
\qquad (29)
$$

Integrating both sides from $s$ to $t$

$$
\begin{aligned}
\boldsymbol{y}_t = \boldsymbol{y}_s &+ \int_s^t e^{-\int_{t_0}^u \left( \frac{\mathrm{d}\log \alpha_v}{\mathrm{d}v} - \left(1+\tau^2(v)\right)\frac{\mathrm{d}\lambda_v}{\mathrm{d}v} \right)\mathrm{d}v} \left[ \left(1+\tau^2(u)\right)\alpha_u \boldsymbol{x_\theta}(\boldsymbol{x}_u, u)\frac{\mathrm{d}\lambda_u}{\mathrm{d}u} \right]\mathrm{d}u \\
&+ \int_s^t e^{-\int_{t_0}^u \left( \frac{\mathrm{d}\log \alpha_v}{\mathrm{d}v} - \left(1+\tau^2(v)\right)\frac{\mathrm{d}\lambda_v}{\mathrm{d}v} \right)\mathrm{d}v}\tau(u)\sigma_u \sqrt{-2\frac{\mathrm{d}\lambda_u}{\mathrm{d}u}}\mathrm{d}\bar{\boldsymbol{w}}_u.
\end{aligned}
\tag{30}
$$

Substituting the definition of $\boldsymbol{y}_t$, $\boldsymbol{y}_s$ into Eq. (30), we obtain Eq. (27)

$$
\begin{aligned}
\boldsymbol{x}_t = e^{\int_s^t \left( \frac{\mathrm{d}\log \alpha_u}{\mathrm{d}u} - \left(1+\tau^2(u)\right)\frac{\mathrm{d}\lambda_u}{\mathrm{d}u} \right)\mathrm{d}u}&\boldsymbol{x}_s \\
&+ \int_s^t e^{-\int_t^u \left( \frac{\mathrm{d}\log \alpha_v}{\mathrm{d}v} - \left(1+\tau^2(v)\right)\frac{\mathrm{d}\lambda_v}{\mathrm{d}v} \right)\mathrm{d}v} \left[ \left(1+\tau^2(u)\right)\alpha_u \boldsymbol{x_\theta}(\boldsymbol{x}_u, u)\frac{\mathrm{d}\lambda_u}{\mathrm{d}u} \right]\mathrm{d}u \\
&+ \int_s^t e^{-\int_t^u \left( \frac{\mathrm{d}\log \alpha_v}{\mathrm{d}v} - \left(1+\tau^2(v)\right)\frac{\mathrm{d}\lambda_v}{\mathrm{d}v} \right)\mathrm{d}v}\tau(u)\sigma_u \sqrt{-2\frac{\mathrm{d}\lambda_u}{\mathrm{d}u}}\mathrm{d}\bar{\boldsymbol{w}}_u. \\
= e^{\int_s^t \left( \frac{\mathrm{d}\log \sigma_u}{\mathrm{d}u} - \tau^2(u)\frac{\mathrm{d}\lambda_u}{\mathrm{d}u} \right)\mathrm{d}u}&\boldsymbol{x}_s \\
&+ \int_s^t e^{-\int_t^u \left( \frac{\mathrm{d}\log \sigma_v}{\mathrm{d}v} - \tau^2(v)\frac{\mathrm{d}\lambda_v}{\mathrm{d}v} \right)\mathrm{d}v} \left[ \left(1+\tau^2(u)\right)\alpha_u \boldsymbol{x_\theta}(\boldsymbol{x}_u, u)\frac{\mathrm{d}\lambda_u}{\mathrm{d}u} \right]\mathrm{d}u \\
&+ \int_s^t e^{-\int_t^u \left( \frac{\mathrm{d}\log \sigma_v}{\mathrm{d}v} - \tau^2(v)\frac{\mathrm{d}\lambda_v}{\mathrm{d}v} \right)\mathrm{d}v}\tau(u)\sigma_u \sqrt{-2\frac{\mathrm{d}\lambda_u}{\mathrm{d}u}}\mathrm{d}\bar{\boldsymbol{w}}_u \\
= \frac{\sigma_t}{\sigma_s}e^{-\int_s^t \tau^2(u)\frac{\mathrm{d}\lambda_u}{\mathrm{d}u}\mathrm{d}u}&\boldsymbol{x}_s \\
&+ \int_s^t \frac{\sigma_t}{\sigma_u}e^{-\int_u^t \tau^2(v)\frac{\mathrm{d}\lambda_v}{\mathrm{d}v}\mathrm{d}v} \left[ \left(1+\tau^2(u)\right)\alpha_u \boldsymbol{x_\theta}(\boldsymbol{x}_u, u)\frac{\mathrm{d}\lambda_u}{\mathrm{d}u} \right]\mathrm{d}u \\
&+ \int_s^t \frac{\sigma_t}{\sigma_u}e^{-\int_u^t \tau^2(v)\frac{\mathrm{d}\lambda_v}{\mathrm{d}v}\mathrm{d}v}\tau(u)\sigma_u \sqrt{-2\frac{\mathrm{d}\lambda_u}{\mathrm{d}u}}\mathrm{d}\bar{\boldsymbol{w}}_u. \\
= \frac{\sigma_t}{\sigma_s}e^{-\int_{\lambda_s}^{\lambda_t} \tau^2(\tilde{\lambda})\mathrm{d}\tilde{\lambda}}&\boldsymbol{x}_s \\
&+ \sigma_t \int_{\lambda_s}^{\lambda_t} e^{-\int_\lambda^{\lambda_t} \tau^2(\tilde{\lambda})\mathrm{d}\tilde{\lambda}} \left(1+\tau^2(\lambda)\right)e^\lambda \boldsymbol{x_\theta}(\boldsymbol{x}_\lambda, \lambda)\mathrm{d}\lambda \\
&+ \sigma_t \int_s^t e^{-\int_u^t \tau^2(v)\frac{\mathrm{d}\lambda_v}{\mathrm{d}v}\mathrm{d}v}\tau(u)\sqrt{-2\frac{\mathrm{d}\lambda_u}{\mathrm{d}u}}\mathrm{d}\bar{\boldsymbol{w}}_u.
\end{aligned}
\tag{31}
$$

The last term $\sigma_t \int_s^t e^{-\int_u^t \tau^2(v)\frac{\mathrm{d}\lambda_v}{\mathrm{d}v}\mathrm{d}v}\tau(u)\sqrt{-2\frac{\mathrm{d}\lambda_u}{\mathrm{d}u}}\mathrm{d}\bar{\boldsymbol{w}}_u$ of Eq. (31) is the *Itô* integral term. It follows a Gaussian distribution, which can be directly derived from two basic facts [38]: first, the definition of *Itô* integral is the limitation in $L^2$ space; second, the limit of Gaussian Process in $L^2$ space is still Gaussian. Then we can compute the mean as:

$$
\mathbb{E}\left[ \sigma_t \int_s^t e^{-\int_u^t \tau^2(v)\frac{\mathrm{d}\lambda_v}{\mathrm{d}v}\mathrm{d}v}\tau(u)\sqrt{-2\frac{\mathrm{d}\lambda_u}{\mathrm{d}u}}\mathrm{d}\bar{\boldsymbol{w}}_u \right] = 0
\tag{32}
$$

and the variance is

$$
\mathrm{Var}\left[\sigma_t \int_s^t e^{-\int_u^t \tau^2(v)\frac{\mathrm{d}\lambda_v}{\mathrm{d}v}\mathrm{d}v}\tau(u)\sqrt{-2\frac{\mathrm{d}\lambda_u}{\mathrm{d}u}}\mathrm{d}\bar{\boldsymbol{w}}_u\right]
$$

$$
=\mathbb{E}\left[\left(\sigma_t \int_s^t e^{-\int_u^t \tau^2(v)\frac{\mathrm{d}\lambda_v}{\mathrm{d}v}\mathrm{d}v}\tau(u)\sqrt{-2\frac{\mathrm{d}\lambda_u}{\mathrm{d}u}}\mathrm{d}\bar{\boldsymbol{w}}_u\right)^2\right] - \left(\mathbb{E}\left[\sigma_t \int_s^t e^{-\int_u^t \tau^2(v)\frac{\mathrm{d}\lambda_v}{\mathrm{d}v}\mathrm{d}v}\tau(u)\sqrt{-2\frac{\mathrm{d}\lambda_u}{\mathrm{d}u}}\mathrm{d}\bar{\boldsymbol{w}}_u\right]\right)^2
$$

$$
=\mathbb{E}\left[\left(\sigma_t \int_t^s e^{-\int_u^t \tau^2(v)\frac{\mathrm{d}\lambda_v}{\mathrm{d}v}\mathrm{d}v}\tau(u)\sqrt{-2\frac{\mathrm{d}\lambda_u}{\mathrm{d}u}}\mathrm{d}\boldsymbol{w}_u\right)^2\right] - 0
$$

$$
=\sigma_t^2\mathbb{E}\left[\int_t^s \left(e^{-\int_u^t \tau^2(v)\frac{\mathrm{d}\lambda_v}{\mathrm{d}v}\mathrm{d}v}\tau(u)\sqrt{-2\frac{\mathrm{d}\lambda_u}{\mathrm{d}u}}\right)^2 \mathrm{d}u\right]
$$

$$
=\sigma_t^2 \int_t^s e^{-\int_u^t 2\tau^2(v)\frac{\mathrm{d}\lambda_v}{\mathrm{d}v}\mathrm{d}v}\tau^2(u)\left(-2\frac{\mathrm{d}\lambda_u}{\mathrm{d}u}\right)\mathrm{d}u
$$

$$
=\sigma_t^2 \int_{\lambda_s}^{\lambda_t} 2e^{-\int_\lambda^{\lambda_t} 2\tau^2(\tilde{\lambda})\mathrm{d}\tilde{\lambda}}\tau^2(\lambda)\mathrm{d}\lambda
$$

(33)

The expectation equals zero because the Itô integral is a martingale [38]. The computation of variance uses the *Itô Isometry*, which is a crucial fact of Itô integral. We can further simplify the result by using the change of variable $P(\lambda) = e^{\int_{\lambda_t}^\lambda 2\tau^2(\tilde{\lambda})\mathrm{d}\tilde{\lambda}}$.

$$
\mathrm{Var}\left[\sigma_t \int_s^t e^{-\int_u^t \tau^2(v)\frac{\mathrm{d}\lambda_v}{\mathrm{d}v}\mathrm{d}v}\tau(u)\sqrt{-2\frac{\mathrm{d}\lambda_u}{\mathrm{d}u}}\mathrm{d}\bar{\boldsymbol{w}}_u\right]
$$

$$
=\sigma_t^2 \int_{\lambda_s}^{\lambda_t} 2e^{-\int_\lambda^{\lambda_t} 2\tau^2(\tilde{\lambda})\mathrm{d}\tilde{\lambda}}\tau^2(\lambda)\mathrm{d}\lambda
$$

$$
=\sigma_t^2 \int_{P(\lambda_s)}^{P(\lambda_t)} P(\lambda)\frac{\mathrm{d}P(\lambda)}{P(\lambda)}
$$

$$
=\sigma_t^2\left(P(\lambda_t) - P(\lambda_s)\right)
$$

$$
=\sigma_t^2(1 - e^{-2\int_{\lambda_s}^{\lambda_t} \tau^2(\lambda)\mathrm{d}\lambda})
$$

(34)

$\square$

### A.2.3 Noise Prediction Reparameterization

After approximating $\nabla_{\boldsymbol{x}}\log p_t(\boldsymbol{x}_t)$ with $\boldsymbol{s_\theta}(\boldsymbol{x}_t, t)$ and reparameterizing $\boldsymbol{s_\theta}(\boldsymbol{x}_t, t)$ with $-\boldsymbol{\epsilon_\theta}(\boldsymbol{x}_t, t))/\sigma_t$, Eq. (20) becomes

$$
\mathrm{d}\boldsymbol{x}_t = \left[f(t)\boldsymbol{x}_t + \left(\frac{1+\tau^2(t)}{2\sigma_t}\right)g^2(t)\boldsymbol{\epsilon_\theta}(\boldsymbol{x}_t, t)\right]\mathrm{d}t + \tau(t)g(t)\mathrm{d}\bar{\boldsymbol{w}}_t.
$$

(35)

Applying change-of-variable with log-SNR $\lambda_t = \log(\alpha_t/\sigma_t)$ and substituting the following relationship

$$
f(t) = \frac{\mathrm{d}\log\alpha_t}{\mathrm{d}t}, \quad g^2(t) = \frac{\mathrm{d}\sigma_t^2}{\mathrm{d}t} - 2\frac{\mathrm{d}\log\alpha_t}{\mathrm{d}t}\sigma_t^2 = -2\sigma_t^2\frac{\mathrm{d}\lambda_t}{\mathrm{d}t},
$$

(36)

Eq. (35) becomes

$$
\mathrm{d}\boldsymbol{x}_t = \left[\frac{\mathrm{d}\log\alpha_t}{\mathrm{d}t}\boldsymbol{x}_t - \left(1+\tau^2(t)\right)\sigma_t\boldsymbol{\epsilon_\theta}(\boldsymbol{x}_t, t)\frac{\mathrm{d}\lambda_t}{\mathrm{d}t}\right]\mathrm{d}t + \tau(t)\sigma_t\sqrt{-2\frac{\mathrm{d}\lambda_t}{\mathrm{d}t}}\mathrm{d}\bar{\boldsymbol{w}}_t.
$$

(37)

Eq (35) is the formulation of *noist prediction model*. Similar with Proposition 4.2, Eq. (37) can be solved analytically, which is shown in the following propositon:

**Proposition A.1.** *Given $\boldsymbol{x}_s$ for any time $s > 0$, the solution $\boldsymbol{x}_t$ at time $t \in [0, s]$ of (37) is*

$$\boldsymbol{x}_t = \frac{\alpha_t}{\alpha_s}\boldsymbol{x}_s + \alpha_t \boldsymbol{F}_{\boldsymbol{\theta}}(s, t) + \alpha_t \boldsymbol{G}(s, t),$$

$$\boldsymbol{F}_{\boldsymbol{\theta}}(s, t) = \int_{\lambda_s}^{\lambda_t} e^{-\lambda}\left(1 + \tau^2(\lambda)\right)\boldsymbol{\epsilon}_{\boldsymbol{\theta}}(\boldsymbol{x}_\lambda, \lambda)\mathrm{d}\lambda \tag{38}$$

$$\boldsymbol{G}(s, t) = \int_s^t e^{-\lambda_u}\tau(u)\sqrt{-2\frac{\mathrm{d}\lambda_u}{\mathrm{d}u}}\mathrm{d}\bar{\boldsymbol{w}}_u,$$

*where $\boldsymbol{G}(s, t)$ is an Itô integral [38] with the special property*

$$\alpha_t \boldsymbol{G}(s, t) \sim \mathcal{N}\left(\mathbf{0}, \alpha_t^2 \int_{\lambda_s}^{\lambda_t} 2e^{-2\lambda}\tau^2(\lambda)\mathrm{d}\lambda\right). \tag{39}$$

*Proof.* Define $\boldsymbol{y}_t = e^{-\int_{t_0}^t \frac{\mathrm{d}\log\alpha_v}{\mathrm{d}v}\mathrm{d}v}\boldsymbol{x}_t$, where $t_0 \in [0, T]$ is a constant. Differentiate $\boldsymbol{y}_t$ with respect to $t$, we get

$$
\begin{aligned}
\mathrm{d}\boldsymbol{y}_t &= -\frac{\mathrm{d}\log\alpha_t}{\mathrm{d}t}e^{-\int_{t_0}^t \frac{\mathrm{d}\log\alpha_v}{\mathrm{d}v}\mathrm{d}v}\boldsymbol{x}_t\mathrm{d}t + e^{-\int_{t_0}^t \frac{\mathrm{d}\log\alpha_v}{\mathrm{d}v}\mathrm{d}v}\mathrm{d}\boldsymbol{x}_t \\
&= -\frac{\mathrm{d}\log\alpha_t}{\mathrm{d}t}e^{-\int_{t_0}^t \frac{\mathrm{d}\log\alpha_v}{\mathrm{d}v}\mathrm{d}v}\boldsymbol{x}_t\mathrm{d}t + e^{-\int_{t_0}^t \frac{\mathrm{d}\log\alpha_v}{\mathrm{d}v}\mathrm{d}v}\frac{\mathrm{d}\log\alpha_t}{\mathrm{d}t}\boldsymbol{x}_t\mathrm{d}t \\
&\quad - e^{-\int_{t_0}^t \frac{\mathrm{d}\log\alpha_v}{\mathrm{d}v}\mathrm{d}v}\left(1 + \tau^2(t)\right)\sigma_t\boldsymbol{\epsilon}_{\boldsymbol{\theta}}(\boldsymbol{x}_t, t)\frac{\mathrm{d}\lambda_t}{\mathrm{d}t}\mathrm{d}t + e^{-\int_{t_0}^t \frac{\mathrm{d}\log\alpha_v}{\mathrm{d}v}\mathrm{d}v}\tau(t)\sigma_t\sqrt{-2\frac{\mathrm{d}\lambda_t}{\mathrm{d}t}}\mathrm{d}\bar{\boldsymbol{w}}_t \\
&= -e^{-\int_{t_0}^t \frac{\mathrm{d}\log\alpha_v}{\mathrm{d}v}\mathrm{d}v}\left(1 + \tau^2(t)\right)\sigma_t\boldsymbol{\epsilon}_{\boldsymbol{\theta}}(\boldsymbol{x}_t, t)\frac{\mathrm{d}\lambda_t}{\mathrm{d}t}\mathrm{d}t + e^{-\int_{t_0}^t \frac{\mathrm{d}\log\alpha_v}{\mathrm{d}v}\mathrm{d}v}\tau(t)\sigma_t\sqrt{-2\frac{\mathrm{d}\lambda_t}{\mathrm{d}t}}\mathrm{d}\bar{\boldsymbol{w}}_t.
\end{aligned}
\tag{40}
$$

Integrating both sides from $s$ to $t$

$$
\begin{aligned}
\boldsymbol{y}_t = \boldsymbol{y}_s &- \int_s^t e^{-\int_{t_0}^u \frac{\mathrm{d}\log\alpha_v}{\mathrm{d}v}\mathrm{d}v}\left(1 + \tau^2(u)\right)\sigma_u\boldsymbol{\epsilon}_{\boldsymbol{\theta}}(\boldsymbol{x}_u, u)\frac{\mathrm{d}\lambda_u}{\mathrm{d}u}\mathrm{d}u \\
&+ \int_s^t e^{-\int_{t_0}^u \frac{\mathrm{d}\log\alpha_v}{\mathrm{d}v}\mathrm{d}v}\tau(u)\sigma_u\sqrt{-2\frac{\mathrm{d}\lambda_u}{\mathrm{d}u}}\mathrm{d}\bar{\boldsymbol{w}}_u.
\end{aligned}
\tag{41}
$$

Substituting the definition of $\boldsymbol{y}_t$, $\boldsymbol{y}_s$ into Eq. (41), we obtain

$$
\begin{aligned}
\boldsymbol{x}_t &= e^{\int_s^t \frac{\mathrm{d}\log\alpha_u}{\mathrm{d}u}\mathrm{d}u}\boldsymbol{x}_s + \int_s^t e^{-\int_t^u \frac{\mathrm{d}\log\alpha_v}{\mathrm{d}v}\mathrm{d}v}\left(1 + \tau^2(u)\right)\sigma_u\boldsymbol{\epsilon}_{\boldsymbol{\theta}}(\boldsymbol{x}_u, u)\frac{\mathrm{d}\lambda_u}{\mathrm{d}u}\mathrm{d}u \\
&\quad + \int_s^t e^{-\int_t^u \frac{\mathrm{d}\log\alpha_v}{\mathrm{d}v}\mathrm{d}v}\tau(u)\sigma_u\sqrt{-2\frac{\mathrm{d}\lambda_u}{\mathrm{d}u}}\mathrm{d}\bar{\boldsymbol{w}}_u \\
&= \frac{\alpha_t}{\alpha_s}\boldsymbol{x}_s + \int_s^t \frac{\alpha_t}{\alpha_u}\left(1 + \tau^2(u)\right)\sigma_u\boldsymbol{\epsilon}_{\boldsymbol{\theta}}(\boldsymbol{x}_u, u)\frac{\mathrm{d}\lambda_u}{\mathrm{d}u}\mathrm{d}u + \int_s^t \frac{\alpha_t}{\alpha_u}\tau(u)\sigma_u\sqrt{-2\frac{\mathrm{d}\lambda_u}{\mathrm{d}u}}\mathrm{d}\bar{\boldsymbol{w}}_u \\
&= \frac{\alpha_t}{\alpha_s}\boldsymbol{x}_s + \alpha_t\int_{\lambda_s}^{\lambda_t} e^{-\lambda}\left(1 + \tau^2(\lambda)\right)\boldsymbol{\epsilon}_{\boldsymbol{\theta}}(\boldsymbol{x}_\lambda, \lambda)\mathrm{d}\lambda + \alpha_t\int_s^t e^{-\lambda_u}\tau(u)\sqrt{-2\frac{\mathrm{d}\lambda_u}{\mathrm{d}u}}\mathrm{d}\bar{\boldsymbol{w}}_u.
\end{aligned}
\tag{42}
$$

The Itô integral term $\alpha_t \int_s^t e^{-\lambda_u}\tau(u)\sqrt{-2\frac{\mathrm{d}\lambda_u}{\mathrm{d}u}}\mathrm{d}\bar{\boldsymbol{w}}_u$ follows a Gaussian distribution. Following the derivation in Proposition 4.2, the mean of the *Itô* integral term is:

$$\mathbb{E}\left[\alpha_t\int_s^t e^{-\lambda_u}\tau(u)\sqrt{-2\frac{\mathrm{d}\lambda_u}{\mathrm{d}u}}\mathrm{d}\bar{\boldsymbol{w}}_u\right] = 0 \tag{43}$$

and the expectation is

$$
\begin{aligned}
&\mathrm{Var}\left[\alpha_t \int_s^t e^{-\lambda_u}\tau(u)\sqrt{-2\frac{\mathrm{d}\lambda_u}{\mathrm{d}u}}\mathrm{d}\bar{\boldsymbol{w}}_u\right]\\
=&\mathbb{E}\left[\left(\alpha_t \int_s^t e^{-\lambda_u}\tau(u)\sqrt{-2\frac{\mathrm{d}\lambda_u}{\mathrm{d}u}}\mathrm{d}\bar{\boldsymbol{w}}_u\right)^2\right]-\left(\mathbb{E}\left[\alpha_t \int_s^t e^{-\lambda_u}\tau(u)\sqrt{-2\frac{\mathrm{d}\lambda_u}{\mathrm{d}u}}\mathrm{d}\bar{\boldsymbol{w}}_u\right]\right)^2\\
=&\mathbb{E}\left[\left(\alpha_t \int_t^s e^{-\lambda_u}\tau(u)\sqrt{-2\frac{\mathrm{d}\lambda_u}{\mathrm{d}u}}\mathrm{d}\boldsymbol{w}_u\right)^2\right]-0\\
=&\alpha_t^2\mathbb{E}\left[\int_t^s \left(e^{-\lambda_u}\tau(u)\sqrt{-2\frac{\mathrm{d}\lambda_u}{\mathrm{d}u}}\right)^2\mathrm{d}u\right]\\
=&\alpha_t^2\int_t^s e^{-2\lambda_u}\tau^2(u)\mathrm{d}u\left(-2\frac{\mathrm{d}\lambda_u}{\mathrm{d}u}\right)\mathrm{d}u\\
=&\alpha_t^2\int_{\lambda_s}^{\lambda_t} 2e^{-2\lambda}\tau^2(\lambda)\mathrm{d}\lambda
\end{aligned}
\tag{44}
$$

$\square$

### A.2.4 Comparison between Data and Noise Reparameterizations

In Table 1 we perform an ablation study on data and noise reparameterizations, the experiment results show that under the same magnitude of stochasticity, the proposed *SA-Solver* in data reparameterization has a better convergence which leads to better FID results under the same NFEs. In this subsection, we provide a theoretical view of this phenomenon.

**Corollary A.2.** *For any bounded measurable function $\tau(t)$, the following inequality holds*

$$
\sigma_t^2\left(1-e^{-2\int_{\lambda_s}^{\lambda_t}\tau^2(\tilde{\lambda})\mathrm{d}\tilde{\lambda}}\right)\leq \alpha_t^2\int_{\lambda_s}^{\lambda_t} 2e^{-2\lambda}\tau^2(\lambda)\mathrm{d}\lambda.
\tag{45}
$$

*Proof.* It's equivalent to show that

$$
1-e^{-2\int_{\lambda_s}^{\lambda_t}\tau^2(\tilde{\lambda})\mathrm{d}\tilde{\lambda}}\leq e^{2\lambda_t}\int_{\lambda_s}^{\lambda_t} 2e^{-2\lambda}\tau^2(\lambda)\mathrm{d}\lambda.
\tag{46}
$$

From the basic inequality $1-e^{-x}\leq x$, we have

$$
1-e^{-2\int_{\lambda_s}^{\lambda_t}\tau^2(\tilde{\lambda})\mathrm{d}\tilde{\lambda}}\leq 2\int_{\lambda_s}^{\lambda_t}\tau^2(\lambda)\mathrm{d}\lambda.
\tag{47}
$$

Thus it's sufficient to show that

$$
e^{2\lambda_t}\int_{\lambda_s}^{\lambda_t} 2e^{-2\lambda}\tau^2(\lambda)\mathrm{d}\lambda\geq 2\int_{\lambda_s}^{\lambda_t}\tau^2(\lambda)\mathrm{d}\lambda,
\tag{48}
$$

which is true because

$$
\int_{\lambda_s}^{\lambda_t} 2\left(e^{2(\lambda_t-\lambda)}-1\right)\tau^2(\lambda)\mathrm{d}\lambda\geq 0,
\tag{49}
$$

$\square$

This corollary indicates that the same SDE under two different reparameterizations has different properties under the effect of the exponential integrator. Specifically, in the numerical scheme, the data reparameterization will inject smaller noise in each step's updation. We speculate that this is the reason that the data reparameterization has a better convergence, shown as in Table 1.

# B Derivations and Proofs for *SA-Solver*

## B.1 Preliminary

We will first review some basic concepts and formulas in the numerical solutions of SDEs [39]. Suppose we have an *Itô* SDE $\mathrm{d}\boldsymbol{x}_t = f(\boldsymbol{x}_t, t)\mathrm{d}t + g(\boldsymbol{x}_t, t)\mathrm{d}\boldsymbol{w}_t$ and time steps $\{t_i\}_{i=0}^M$, $t_i \in [0, T]$ to numerically solve the SDE. For a random variable $Z$, we define the $L_1$ norm $\|Z\|_{L_1} = \mathbb{E}[|Z|]$, the $L_2$ norm $\|Z\|_{L_2} = \mathbb{E}[|Z|^2]^{\frac{1}{2}}$, where $|\cdot|$ is the Euclidean norm. Denote $h = \max\limits_{1 \leq i \leq M}(t_i - t_{i-1})$.

**Definition B.1.** *We shall say that a time-discrete approximation $\boldsymbol{x}_0, \cdots, \boldsymbol{x}_M$, where $\boldsymbol{x}_i$ is a numerical approximation of $\boldsymbol{x}_{t_i}$, converges strongly with order $\gamma > 0$, if there exists a positive constant $C$, which does not depend on $h$ and a $h_0 > 0$ such that*

$$\max_{0 \leq i \leq M} \|\boldsymbol{x}_{t_i} - \boldsymbol{x}_i\|_{L_1} \leq Ch^\gamma, \quad \forall h \leq h_0. \tag{50}$$

**Definition B.2.** *We say it is mean-square convergent with order $\gamma > 0$, if there exists a positive constant $C$, which does not depend on $h$ and a $h_0 > 0$ such that*

$$\max_{0 \leq i \leq M} \|\boldsymbol{x}_{t_i} - \boldsymbol{x}_i\|_{L_2} \leq Ch^\gamma, \quad \forall h \leq h_0. \tag{51}$$

**Remark 2.** *To prove the strong convergence order $\gamma$ of a numerical scheme, it's sufficient to show the mean-square convergence order $\gamma$. This is from Hölder Inequality $\mathbb{E}[|Z|] \leq \mathbb{E}[|Z|^2]^{\frac{1}{2}} \mathbb{E}[|1|^2]^{\frac{1}{2}} \leq \mathbb{E}[|Z|^2]^{\frac{1}{2}}$. Thus $\max\limits_{0 \leq i \leq M} \|\boldsymbol{x}_{t_i} - \boldsymbol{x}_i\|_{L_1} \leq \max\limits_{0 \leq i \leq M} \|\boldsymbol{x}_{t_i} - \boldsymbol{x}_i\|_{L_2}$.*

We also need the following definition and assumptions, which usually holds in practical diffusion models.

**Definition B.3.** *A function $h : \mathbb{R}^d \times [0, T] \to \mathbb{R}^d$ satisfies a linear growth condition if there exists a constant $K$ such that*

$$|h(\boldsymbol{x}, t)| \leq K(1 + |\boldsymbol{x}|^2)^{\frac{1}{2}} \tag{52}$$

**Assumption B.4.** *The data prediction model $\boldsymbol{x_\theta}$ and its derivatives such as $\partial_t \boldsymbol{x_\theta}$, $\nabla_{\boldsymbol{x}}\boldsymbol{x_\theta}$ and $\Delta\boldsymbol{x_\theta}$ satisfy the linear growth condition.*

**Assumption B.5.** *The data prediction model $\boldsymbol{x_\theta}$ satisfies a uniform Lipschitz condition with respect to $\boldsymbol{x}$*

$$|\boldsymbol{x_\theta}(\boldsymbol{x}_1, t) - \boldsymbol{x_\theta}(\boldsymbol{x}_2, t)| \leq L|\boldsymbol{x}_1 - \boldsymbol{x}_2|, \quad \forall x, y \in \mathbb{R}^d, t \in [0, T] \tag{53}$$

## B.2 Outline of the Proof

In the remaining part of this section, we will focus on our variance controlled SDE

$$\mathrm{d}\boldsymbol{x}_t = \left[\left(\frac{\mathrm{d}\log\alpha_t}{\mathrm{d}t} - (1 + \tau^2(t))\frac{\mathrm{d}\lambda_t}{\mathrm{d}t}\right)\boldsymbol{x}_t + (1 + \tau^2(t))\alpha_t\boldsymbol{x_\theta}(\boldsymbol{x}_t, t)\frac{\mathrm{d}\lambda_t}{\mathrm{d}t}\right]\mathrm{d}t$$
$$+ \tau(t)\sigma_t\sqrt{-2\frac{\mathrm{d}\lambda_t}{\mathrm{d}t}}\mathrm{d}\bar{\boldsymbol{w}}_t. \tag{54}$$

Consider the general case of the numerical scheme as follows:

$$\boldsymbol{x}_{i+1} = \frac{\sigma_{t_{i+1}}}{\sigma_{t_i}}e^{-\int_{\lambda_{t_i}}^{\lambda_{t_{i+1}}} \tau^2(\lambda_u)\mathrm{d}\lambda_u}\boldsymbol{x}_i + \sum_{j=-1}^{s-1} b_{i-j}\boldsymbol{x_\theta}(\boldsymbol{x}_{i-j}, t_{i-j})$$
$$+ \sigma_{t_{i+1}}\int_{t_i}^{t_{i+1}} e^{-\int_u^{t_{i+1}} \tau^2(\lambda)\mathrm{d}\lambda}\tau(u)\sqrt{-2\frac{\mathrm{d}\lambda_u}{\mathrm{d}u}}\mathrm{d}\bar{\boldsymbol{w}}_u. \tag{55}$$

in which Eq. (17) and Eq (14) are the special case of this scheme. We will provide proof of the mean-square convergence order of the numerical scheme $\max\limits_{0 \leq i \leq M} \|\boldsymbol{x}_{t_i} - \boldsymbol{x}_i\|_{L_2}$. We define the local

error of the numerical scheme Eq. (55) for the approximation of the SDE Eq. (54) as

$$
L_{i+1} = \boldsymbol{x}_{t_{i+1}} - \boldsymbol{x}_{i+1} = \boldsymbol{x}_{t_{i+1}} - \frac{\sigma_{t_{i+1}}}{\sigma_{t_i}} e^{-\int_{\lambda_{t_i}}^{\lambda_{t_{i+1}}} \tau^2(\lambda)\mathrm{d}\lambda} \boldsymbol{x}_{t_i} - \sum_{j=-1}^{s-1} b_{i-j}\boldsymbol{x}_{\boldsymbol{\theta}}(\boldsymbol{x}_{t_{i-j}}, t_{i-j})
$$
$$
- \sigma_{t_{i+1}} \int_{t_i}^{t_{i+1}} e^{-\int_u^{t_{i+1}} \tau^2(\lambda)\mathrm{d}\lambda} \tau(u) \sqrt{-2\frac{\mathrm{d}\lambda_u}{\mathrm{d}u}} \mathrm{d}\bar{\boldsymbol{w}}_u.
\tag{56}
$$

$L_{i+1}$ can be decomposed into $R_{i+1}$ and $S_{i+1}$. Then the mean-square convergence can be derived, which is summarized in the following theorem proved by [29]:

**Theorem B.6** ([29], Theorem 1). *The mean-square convergent of $\boldsymbol{x}_i$ is bounded by*

$$
\max_{0 \le i \le M} \|\boldsymbol{x}_{t_i} - \boldsymbol{x}_i\|_{L_2} \le S \left\{ \max_{0 \le i \le s-1} \|D_i\|_{L_2} + \max_{s \le i \le M} \left( \frac{\|R_i\|_{L_2}}{h} + \frac{\|S_i\|_{L_2}}{h^{\frac{1}{2}}} \right) \right\}.
\tag{57}
$$

In Eq. (57), $D_i, i = 0, \cdots, s-1$ are the initial error which we do not consider. Given Theorem B.6, to show the convergence order $\mathcal{O}(\max_{0 \le t \le T} \tau(t)h + h^s)$ of our $s$-step *SA-Predictor* and the convergence order $\mathcal{O}(\max_{0 \le t \le T} \tau(t)h + h^{s+1})$ of our $s$-step *SA-Corrector*, we just need to prove the following lemmas.

**Lemma B.7** (Convergence rate of $s$-step *SA-Predictor*). *For*

$$
\boldsymbol{x}_{t_{i+1}} = \frac{\sigma_{t_{i+1}}}{\sigma_{t_i}} e^{-\int_{\lambda_{t_i}}^{\lambda_{t_{i+1}}} \tau^2(\lambda_u)\mathrm{d}\lambda_u} \boldsymbol{x}_{t_i} + \sum_{j=0}^{s-1} b_{i-j}\boldsymbol{x}_{\boldsymbol{\theta}}(\boldsymbol{x}_{t_{i-j}}, t_{i-j}) + \tilde{\sigma}_i \boldsymbol{\xi}, \quad \boldsymbol{\xi} \sim \mathcal{N}(\boldsymbol{0}, \boldsymbol{I}),
$$
$$
\tilde{\sigma}_i = \sigma_{t_{i+1}} \sqrt{1 - e^{-2\int_{\lambda_{t_i}}^{\lambda_{t_{i+1}}} \tau^2(\lambda)\mathrm{d}\lambda}}
$$
$$
b_{i-j} = \sigma_{t_{i+1}} \int_{\lambda_{t_i}}^{\lambda_{t_{i+1}}} e^{-\int_{\lambda_u}^{\lambda_{t_{i+1}}} \tau^2(\lambda_v)\mathrm{d}\lambda_v} \left(1 + \tau^2(\lambda_u)\right) e^{\lambda_u} l_{i-j}(\lambda_u)\mathrm{d}\lambda_u, \quad \forall 0 \le j \le s-1
\tag{58}
$$

*There exists an decomposition of local error $L_i$ such that $L_i = R_i + S_i$ and*

$$
\|R_i\|_{L_2} \le h^{s+1}, \|S_i\|_{L_2} \le \max_{0 \le t \le T} \tau(t)h^{\frac{3}{2}},
\tag{59}
$$

**Lemma B.8** (Convergence rate of $s$-step *SA-Corrector*). *For*

$$
\boldsymbol{x}_{t_{i+1}} = \frac{\sigma_{t_{i+1}}}{\sigma_{t_i}} e^{-\int_{\lambda_{t_i}}^{\lambda_{t_{i+1}}} \tau^2(\lambda_u)\mathrm{d}\lambda_u} \boldsymbol{x}_{t_i} + \hat{b}_{i+1}\boldsymbol{x}_{\boldsymbol{\theta}}(\boldsymbol{x}_{t_{i+1}}^p, t_{i+1}) + \sum_{j=0}^{\hat{s}-1} \hat{b}_{i-j}\boldsymbol{x}_{\boldsymbol{\theta}}(\boldsymbol{x}_{t_{i-j}}, t_{i-j}) + \tilde{\sigma}_i \boldsymbol{\xi},
$$
$$
\tilde{\sigma}_i = \sigma_{t_{i+1}} \sqrt{1 - e^{-2\int_{\lambda_{t_i}}^{\lambda_{t_{i+1}}} \tau^2(\lambda)\mathrm{d}\lambda}}
$$
$$
\hat{b}_{i-j} = \sigma_{t_{i+1}} \int_{\lambda_{t_i}}^{\lambda_{t_{i+1}}} e^{-\int_{\lambda_u}^{\lambda_{t_{i+1}}} \tau^2(\lambda_v)\mathrm{d}\lambda_v} \left(1 + \tau^2(\lambda_u)\right) e^{\lambda_u} \hat{l}_{i-j}(\lambda_u)\mathrm{d}\lambda_u, \quad \forall 0 \le j \le s-1
$$
$$
\hat{b}_{i+1} = \sigma_{t_{i+1}} \int_{\lambda_{t_i}}^{\lambda_{t_{i+1}}} e^{-\int_{\lambda_u}^{\lambda_{t_{i+1}}} \tau^2(\lambda_v)\mathrm{d}\lambda_v} \left(1 + \tau^2(\lambda_u)\right) e^{\lambda_u} \hat{l}_{i+1}(\lambda_u)\mathrm{d}\lambda_u
\tag{60}
$$

*There exists an decomposition of local error $L_i$ such that $L_i = R_i + S_i$ and*

$$
\|R_i\|_{L_2} \le h^{s+2}, \|S_i\|_{L_2} \le \max_{0 \le t \le T} \tau(t)h^{\frac{3}{2}},
\tag{61}
$$

Lemma B.7 and B.8 will be proved in Sec. B.4.

## B.3   Lemmas for the Proof

To better analyze the local error here, we state the following definitions and results from [45]. For a continuous function $y : \mathbb{R}^d \times [0, T] \to \mathbb{R}^d$, a general multiple Wiener integral over the subinterval $[t, t + h] \subset [0, T]$ is given by

$$
I_{r_1 r_2 \cdots r_j}^{t, t+h}(y) = \int_t^{t+h} \int_t^{s_1} \cdots \int_t^{s_{j-1}} y(\boldsymbol{x}_{s_j}, s_j)\mathrm{d}w_{r_1}(s_j) \cdots \mathrm{d}w_{r_j}(s_1),
\tag{62}
$$

where $r_i \in \{0, 1, \cdots, d\}$ and $\mathrm{d}w_0(s) = \mathrm{d}s$. Then we have the following lemma.

**Lemma B.9** (Bound of Wiener Integral). *For any function $y : \mathbb{R}^d \times [0, T] \to \mathbb{R}^d$ that satisfies a growth condition in the form $|y(\boldsymbol{x}, t)| \leq K(1 + |\boldsymbol{x}|^2)^{\frac{1}{2}}$, for any $\boldsymbol{x} \in \mathbb{R}^d$, and any $t \in [0, T]$, $h > 0$ such that $t + h \in [0, T]$, we have that*

$$\mathbb{E}\left[ I_{r_1 r_2 \cdots r_j}^{t, t+h}(y) | \mathcal{F}_t \right] = 0 \quad \text{if } r_i \neq 0 \text{ for some } i \in \{1, \cdots, j\}, \tag{63}$$

$$\left\| I_{r_1 r_2 \cdots r_j}^{t, t+h}(y) \right\|_{L_2} = \mathcal{O}\left( h^{l_1 + \frac{l_2}{2}} \right), \tag{64}$$

*where $l_1$ is the number of zero indices and $l_2$ is the number of non-zero indices $r_i$.*

**Lemma B.10** (Property of Lagrange interpolation polynomial). *For $s + 1$ points $(t_{i+1}, y_{i+1}), (t_i, y_i), \cdots, (t_{i-(s-1)}, y_{i-(s-1)})$, the Lagrange interpolation polynomial is*

$$L(t) = \sum_{k=i-(s-1)}^{i+1} l_k(t) y_k. \tag{65}$$

*Then the following s+1 equalities hold*

$$\sum_{k=i-(s-1)}^{i+1} l_k(u) = 1,$$

$$\sum_{k=i-(s-1)}^{i+1} l_k(u) \int_{t_{i-(s-1)}}^{t_k} \mathrm{d}u_2 = \int_{t_{i-(s-1)}}^{u} \mathrm{d}u_2,$$

$$\vdots$$

$$\sum_{k=i-(s-1)}^{i+1} l_k(u) \int_{t_{i-(s-1)}}^{t_k} \int_{t_{i-(s-1)}}^{u_2} \cdots \int_{t_{i-(s-1)}}^{u_s} \mathrm{d}u_{s+1} \cdots \mathrm{d}u_3 \mathrm{d}u_2 =$$

$$\int_{t_{i-(s-1)}}^{u} \int_{t_{i-(s-1)}}^{u_2} \cdots \int_{t_{i-(s-1)}}^{u_s} \mathrm{d}u_{s+1} \cdots \mathrm{d}u_3 \mathrm{d}u_2 \tag{66}$$

*Proof.* For the first equality, consider $y_k \equiv 1$ for $i - (s-1) \leq k \leq i+1$. The Lagrange interpolation polynomial for these $y_k$s is a constant function $L(t) \equiv 1$. We have $L(u) = \sum_{k=i-(s-1)}^{i+1} l_k(u) = 1$.

For the second equality, consider $y_k = \int_{t_{i-(s-1)}}^{t_k} \mathrm{d}u_2$. The Lagrange interpolation polynomial for these $y_k$s is a polynomial of degree 1 $L(t) = t - t_{i-(s-1)}$. We have $L(u) = \sum_{k=i-(s-1)}^{i+1} l_k(u) \int_{t_{i-(s-1)}}^{t_k} \mathrm{d}u_2 = u - t_{i-(s-1)} = \int_{t_{i-(s-1)}}^{u} \mathrm{d}u_2$.

For equalities from the third to the last, without loss of generality, we prove the $p - th$ equality, where $3 \leq p \leq s + 1$. Consider $y_k = \int_{t_{i-(s-1)}}^{t_k} \int_{t_{i-(s-1)}}^{u_2} \cdots \int_{t_{i-(s-1)}}^{u_{p-1}} \mathrm{d}u_p \cdots \mathrm{d}u_3 \mathrm{d}u_2$. The Lagrange interpolation polynomial for these $y_k$s is a polynomial of degree $p - 1$ $L(t) = \int_{t_{i-(s-1)}}^{t} \int_{t_{i-(s-1)}}^{u_2} \cdots \int_{t_{i-(s-1)}}^{u_{p-1}} \mathrm{d}u_p \cdots \mathrm{d}u_3 \mathrm{d}u_2$. We have $L(u) = \int_{t_{i-(s-1)}}^{u} \int_{t_{i-(s-1)}}^{u_2} \cdots \int_{t_{i-(s-1)}}^{u_{p-1}} \mathrm{d}u_p \cdots \mathrm{d}u_3 \mathrm{d}u_2 = \int_{t_{i-(s-1)}}^{u} \int_{t_{i-(s-1)}}^{u_2} \cdots \int_{t_{i-(s-1)}}^{u_{p-1}} \mathrm{d}u_p \cdots \mathrm{d}u_3 \mathrm{d}u_2$. $\square$

## B.4 Proof of Lemma B.7 (for Theorem. 5.1) and Lemma B.8 (for Theorem. 5.2)

To simplify the notation, we will introduce two operators which will appear in the Itô formula. Suppose we have an *Itô* SDE $\mathrm{d}\boldsymbol{x}_t = f(\boldsymbol{x}_t, t)\mathrm{d}t + g(\boldsymbol{x}_t, t)\mathrm{d}\boldsymbol{w}_t$ and $h(\boldsymbol{x}, t)$ is a twice continuously differentiable function. Let $\Gamma_0(\cdot) = \partial_t(\cdot) + \nabla_{\boldsymbol{x}}(\cdot)f$, $\Gamma_1(\cdot) = \frac{g^2}{2}\Delta(\cdot)$ and $\Gamma_2(\cdot) = \nabla_{\boldsymbol{x}}(\cdot)g$ in which

$\nabla_{\boldsymbol{x}}$ is the Jacobian matrix, and $\Delta$ is the Laplacian operator. With the notation here, we can express the Itô formula for $h(\boldsymbol{x}, t)$ as

$$h(\boldsymbol{x}_t, t) = h(\boldsymbol{x}_s, s) + \int_s^t \left( \Gamma_0(h) + \Gamma_1(h) \right) \mathrm{d}t + \int_s^t \Gamma_2(h) \mathrm{d}\bar{\boldsymbol{w}}_t. \tag{67}$$

Given the above lemmas, we will analyze the local error $L_{i+1}$ step by step. Inspired by Theorem B.6, for data-prediction reparameterization model, $L_{i+i}$ can be estimated by decomposing the terms step by step. The first step of decomposition is summarized as the following lemma:

**Lemma B.11** (First step of estimating local error $L_{i+1}$ in data-prediction reparameterization model). *Given the exact solution of data prediction model*

$$\boldsymbol{x}_t = \frac{\sigma_t}{\sigma_s} e^{-\int_{\lambda_s}^{\lambda_t} \tau^2(\tilde{\lambda})\mathrm{d}\tilde{\lambda}} \boldsymbol{x}_s + \sigma_t \boldsymbol{F}_{\boldsymbol{\theta}}(s, t) + \sigma_t \boldsymbol{G}(s, t),$$

$$\boldsymbol{F}_{\boldsymbol{\theta}}(s, t) = \int_{\lambda_s}^{\lambda_t} e^{-\int_{\lambda}^{\lambda_t} \tau^2(\tilde{\lambda})\mathrm{d}\tilde{\lambda}} \left(1 + \tau^2(\lambda)\right) e^{\lambda} \boldsymbol{x}_{\boldsymbol{\theta}}(\boldsymbol{x}_\lambda, \lambda) \, \mathrm{d}\lambda \tag{68}$$

$$\boldsymbol{G}(s, t) = \int_s^t e^{-\int_{\lambda_u}^{\lambda_t} \tau^2(\tilde{\lambda})\mathrm{d}\tilde{\lambda}} \tau(u) \sqrt{-2\frac{\mathrm{d}\lambda_u}{\mathrm{d}u}} \mathrm{d}\bar{\boldsymbol{w}}_u,$$

*With proper $b_k, k \in [i - (s-1), i+1]$, The local error $L_{i+1}$ in Eq. (56) is*

$$L_{i+1} = R_{i+1}^{(1)} + S_{i+1}^{(1)} \tag{69}$$

*where*

$$S_{i+1}^{(1)} = \mathcal{O}\left( \max_{0 \le t \le T} \tau(t) h^{\frac{3}{2}} \right)$$

$$\begin{aligned}
R_{i+1}^{(1)} = & \sum_{k=i-(s-1)}^{i-1} \sigma_{t_{i+1}} \left( \int_{t_i}^{t_{i+1}} e^{-\int_{\lambda_u}^{\lambda_{t_{i+1}}} \tau^2(\lambda)\mathrm{d}\lambda} \left(1 + \tau^2(u)\right) e^{\lambda_u} \frac{\mathrm{d}\lambda_u}{\mathrm{d}u} \mathrm{d}u \right) \times \\
& \left( \int_{t_k}^{t_{k+1}} \Gamma_0(\boldsymbol{x}_{\boldsymbol{\theta}})\mathrm{d}t \right) \\
& + \sigma_{t_{i+1}} \int_{t_i}^{t_{i+1}} e^{-\int_{\lambda_u}^{\lambda_{t_{i+1}}} \tau^2(\lambda)\mathrm{d}\lambda} \left(1 + \tau^2(u)\right) e^{\lambda_u} \left( \int_{t_i}^u \Gamma_0(\boldsymbol{x}_{\boldsymbol{\theta}})\mathrm{d}t \right) \frac{\mathrm{d}\lambda_u}{\mathrm{d}u} \mathrm{d}u \\
& - \sum_{j=-1}^{s-1} b_{i-j} \sum_{k=i-(s-1)}^{i-j-1} \int_{t_k}^{t_{k+1}} \Gamma_0(\boldsymbol{x}_{\boldsymbol{\theta}})\mathrm{d}t.
\end{aligned} \tag{70}$$

*Proof.* The difference between Eq. (55) and Eq. (56) is that $\boldsymbol{x}_j$ is our numerical approximation, while $\boldsymbol{x}_{t_j}$ is the exact solution of SDE Eq. (54) at time $t = t_j$. Substitute the exact solution Eq. (31) of $\boldsymbol{x}_{t_{i+1}}$, we have

$$\begin{aligned}
L_{i+1} = & \frac{\sigma_{t_{i+1}}}{\sigma_{t_i}} e^{-\int_{\lambda_{t_i}}^{\lambda_{t_{i+1}}} \tau^2(\lambda)\mathrm{d}\lambda} \boldsymbol{x}_{t_i} + \sigma_{t_{i+1}} \int_{t_i}^{t_{i+1}} e^{-\int_u^{t_{i+1}} \tau^2(\lambda)\mathrm{d}\lambda} \tau(u) \sqrt{-2\frac{\mathrm{d}\lambda_u}{\mathrm{d}u}} \mathrm{d}\bar{\boldsymbol{w}}_u \\
& + \sigma_{t_{i+1}} \int_{\lambda_{t_i}}^{\lambda_{t_{i+1}}} e^{-\int_{\lambda_u}^{\lambda_{t_{i+1}}} \tau^2(\lambda)\mathrm{d}\lambda} \left(1 + \tau^2(\lambda_u)\right) e^{\lambda_u} \boldsymbol{x}_{\boldsymbol{\theta}}(\boldsymbol{x}_{\lambda_u}, \lambda_u)\mathrm{d}\lambda_u \\
& - \sigma_{t_{i+1}} \int_{t_i}^{t_{i+1}} e^{-\int_u^{t_{i+1}} \tau^2(\lambda)\mathrm{d}\lambda} \tau(u) \sqrt{-2\frac{\mathrm{d}\lambda_u}{\mathrm{d}u}} \mathrm{d}\bar{\boldsymbol{w}}_u \\
& - \frac{\sigma_{t_{i+1}}}{\sigma_{t_i}} e^{-\int_{\lambda_{t_i}}^{\lambda_{t_{i+1}}} \tau^2(\lambda_u)\mathrm{d}\lambda_u} \boldsymbol{x}_{t_i} - \sum_{j=-1}^{s-1} b_{i-j} \boldsymbol{x}_{\boldsymbol{\theta}}(\boldsymbol{x}_{t_{i-j}}, t_{i-j}) \\
= & \sigma_{t_{i+1}} \int_{t_i}^{t_{i+1}} e^{-\int_{\lambda_u}^{\lambda_{t_{i+1}}} \tau^2(\lambda)\mathrm{d}\lambda} \left(1 + \tau^2(u)\right) e^{\lambda_u} \boldsymbol{x}_{\boldsymbol{\theta}}(\boldsymbol{x}_u, u) \frac{\mathrm{d}\lambda_u}{\mathrm{d}u} \mathrm{d}u \\
& - \sum_{j=-1}^{s-1} b_{i-j} \boldsymbol{x}_{\boldsymbol{\theta}}(\boldsymbol{x}_{t_{i-j}}, t_{i-j}).
\end{aligned} \tag{71}$$

Let $f(\boldsymbol{x}, t) = \left(\frac{\mathrm{d} \log \alpha_t}{\mathrm{d}t} - \left(1 + \tau^2(t)\right) \frac{\mathrm{d}\lambda_t}{\mathrm{d}t}\right) \boldsymbol{x} + \left(1 + \tau^2(t)\right) \alpha_t \boldsymbol{x_\theta}(\boldsymbol{x}, t) \frac{\mathrm{d}\lambda_t}{\mathrm{d}t}$, $g(t) = \tau(t)\sigma_t \sqrt{-2\frac{\mathrm{d}\lambda_t}{\mathrm{d}t}}$.
By Itô's formula [38], we have

$$
\begin{aligned}
\boldsymbol{x_\theta}(\boldsymbol{x}_u, u) = {}& \boldsymbol{x_\theta}(\boldsymbol{x}_{t_{i-(s-1)}}, t_{i-(s-1)}) + \sum_{k=i-(s-1)}^{i-1} \int_{t_k}^{t_{k+1}} \left(\Gamma_0(\boldsymbol{x_\theta}) + \Gamma_1(\boldsymbol{x_\theta})\right) \mathrm{d}t \\
& + \int_{t_i}^{u} \left(\Gamma_0(\boldsymbol{x_\theta}) + \Gamma_1(\boldsymbol{x_\theta})\right) \mathrm{d}t + \sum_{k=i-(s-1)}^{i-1} \int_{t_k}^{t_{k+1}} \Gamma_2(\boldsymbol{x_\theta}) \mathrm{d}\bar{\boldsymbol{w}}_t + \int_{t_i}^{u} \Gamma_2(\boldsymbol{x_\theta}) \mathrm{d}\bar{\boldsymbol{w}}_t,
\end{aligned}
\tag{72}
$$

$$
\begin{aligned}
\boldsymbol{x_\theta}(\boldsymbol{x}_{t_{i-j}}, t_{i-j}) = {}& \boldsymbol{x_\theta}(\boldsymbol{x}_{t_{i-(s-1)}}, t_{i-(s-1)}) + \sum_{k=i-(s-1)}^{i-j-1} \int_{t_k}^{t_{k+1}} \left(\Gamma_0(\boldsymbol{x_\theta}) + \Gamma_1(\boldsymbol{x_\theta})\right) \mathrm{d}t \\
& + \sum_{k=i-(s-1)}^{i-j-1} \int_{t_k}^{t_{k+1}} \Gamma_2(\boldsymbol{x_\theta}) \mathrm{d}\bar{\boldsymbol{w}}_t,
\end{aligned}
\tag{73}
$$

Substituting Eq. (72) and Eq. (73) into Eq. (71), we have

$$
\begin{aligned}
& L_{i+1} \\
& = \left(\sigma_{t_{i+1}} \int_{t_i}^{t_{i+1}} e^{-\int_{\lambda_u}^{\lambda_{t_{i+1}}} \tau^2(\lambda)\mathrm{d}\lambda} \left(1 + \tau^2(u)\right) e^{\lambda_u} \frac{\mathrm{d}\lambda_u}{\mathrm{d}u} \mathrm{d}u - \sum_{j=-1}^{s-1} b_{i-j}\right) \boldsymbol{x_\theta}(\boldsymbol{x}_{t_{i-(s-1)}}, t_{i-(s-1)}) \\
& + \sum_{k=i-(s-1)}^{i-1} \sigma_{t_{i+1}} \left(\int_{t_i}^{t_{i+1}} e^{-\int_{\lambda_u}^{\lambda_{t_{i+1}}} \tau^2(\lambda)\mathrm{d}\lambda} \left(1 + \tau^2(u)\right) e^{\lambda_u} \frac{\mathrm{d}\lambda_u}{\mathrm{d}u} \mathrm{d}u\right) \times \\
& \quad \left(\int_{t_k}^{t_{k+1}} \left(\Gamma_0(\boldsymbol{x_\theta}) + \Gamma_1(\boldsymbol{x_\theta})\right) \mathrm{d}t\right) \\
& + \sum_{k=i-(s-1)}^{i-1} \sigma_{t_{i+1}} \left(\int_{t_i}^{t_{i+1}} e^{-\int_{\lambda_u}^{\lambda_{t_{i+1}}} \tau^2(\lambda)\mathrm{d}\lambda} \left(1 + \tau^2(u)\right) e^{\lambda_u} \frac{\mathrm{d}\lambda_u}{\mathrm{d}u} \mathrm{d}u\right) \times \left(\int_{t_k}^{t_{k+1}} \Gamma_2(\boldsymbol{x_\theta}) \mathrm{d}\bar{\boldsymbol{w}}_t\right) \\
& + \sigma_{t_{i+1}} \int_{t_i}^{t_{i+1}} e^{-\int_{\lambda_u}^{\lambda_{t_{i+1}}} \tau^2(\lambda)\mathrm{d}\lambda} \left(1 + \tau^2(u)\right) e^{\lambda_u} \left(\int_{t_i}^{u} \left(\Gamma_0(\boldsymbol{x_\theta}) + \Gamma_1(\boldsymbol{x_\theta})\right) \mathrm{d}t\right) \frac{\mathrm{d}\lambda_u}{\mathrm{d}u} \mathrm{d}u \\
& + \sigma_{t_{i+1}} \int_{t_i}^{t_{i+1}} e^{-\int_{\lambda_u}^{\lambda_{t_{i+1}}} \tau^2(\lambda)\mathrm{d}\lambda} \left(1 + \tau^2(u)\right) e^{\lambda_u} \left(\int_{t_i}^{u} \Gamma_2(\boldsymbol{x_\theta}) \mathrm{d}\bar{\boldsymbol{w}}_t\right) \frac{\mathrm{d}\lambda_u}{\mathrm{d}u} \mathrm{d}u \\
& - \sum_{j=-1}^{s-1} b_{i-j} \left(\sum_{k=i-(s-1)}^{i-j-1} \int_{t_k}^{t_{k+1}} \left(\Gamma_0(\boldsymbol{x_\theta}) + \Gamma_1(\boldsymbol{x_\theta})\right) \mathrm{d}t + \sum_{k=i-(s-1)}^{i-j-1} \int_{t_k}^{t_{k+1}} \Gamma_2(\boldsymbol{x_\theta}) \mathrm{d}\bar{\boldsymbol{w}}_t\right)
\end{aligned}
\tag{74}
$$

We will divide the local error $L_{i+1}$ into distinct terms. The first term has a coefficient

$$
\sigma_{t_{i+1}} \int_{t_i}^{t_{i+1}} e^{-\int_{\lambda_u}^{\lambda_{t_{i+1}}} \tau^2(\lambda)\mathrm{d}\lambda} \left(1 + \tau^2(u)\right) e^{\lambda_u} \frac{\mathrm{d}\lambda_u}{\mathrm{d}u} \mathrm{d}u - \sum_{j=-1}^{s-1} b_{i-j}.
\tag{75}
$$

By Lemma B.10, $b_k$ constructed by the integral of Lagrange polynomial in Eq. (58) and Eq. (60) satisfies $b_k = \mathcal{O}(h)$ and the coefficient (75) is zero. Furthermore, we have $g(t) = \tau(t)\sigma_t \sqrt{-2\frac{\mathrm{d}\lambda_t}{\mathrm{d}t}} =$

$\mathcal{O}\left(\max\limits_{0\leq t\leq T}\tau(t)\right)$. By Lemma B.9, we have the following estimations

$$\sum_{k=i-(s-1)}^{i-1}\sigma_{t_{i+1}}\left(\int_{t_i}^{t_{i+1}}e^{-\int_{\lambda_u}^{\lambda_{t_{i+1}}}\tau^2(\lambda)\mathrm{d}\lambda}\left(1+\tau^2(u)\right)e^{\lambda_u}\frac{\mathrm{d}\lambda_u}{\mathrm{d}u}\mathrm{d}u\right)\times\left(\int_{t_k}^{t_{k+1}}\Gamma_1(\boldsymbol{x_\theta})\mathrm{d}t\right)$$

$$=\mathcal{O}\left(\max_{0\leq t\leq T}\tau^2(t)h^2\right),$$

$$\sum_{k=i-(s-1)}^{i-1}\sigma_{t_{i+1}}\left(\int_{t_i}^{t_{i+1}}e^{-\int_{\lambda_u}^{\lambda_{t_{i+1}}}\tau^2(\lambda)\mathrm{d}\lambda}\left(1+\tau^2(u)\right)e^{\lambda_u}\frac{\mathrm{d}\lambda_u}{\mathrm{d}u}\mathrm{d}u\right)\times\left(\int_{t_k}^{t_{k+1}}\Gamma_2(\boldsymbol{x_\theta})\mathrm{d}\bar{\boldsymbol{w}}_t\right)$$

$$=\mathcal{O}\left(\max_{0\leq t\leq T}\tau(t)h^{\frac{3}{2}}\right),$$

$$\sigma_{t_{i+1}}\int_{t_i}^{t_{i+1}}e^{-\int_{\lambda_u}^{\lambda_{t_{i+1}}}\tau^2(\lambda)\mathrm{d}\lambda}\left(1+\tau^2(u)\right)e^{\lambda_u}\left(\int_{t_i}^{u}\Gamma_1(\boldsymbol{x_\theta})\mathrm{d}t\right)\frac{\mathrm{d}\lambda_u}{\mathrm{d}u}\mathrm{d}u=\mathcal{O}\left(\max_{0\leq t\leq T}\tau^2(t)h^2\right),$$

$$\sigma_{t_{i+1}}\int_{t_i}^{t_{i+1}}e^{-\int_{\lambda_u}^{\lambda_{t_{i+1}}}\tau^2(\lambda)\mathrm{d}\lambda}\left(1+\tau^2(u)\right)e^{\lambda_u}\left(\int_{t_i}^{u}\Gamma_2(\boldsymbol{x_\theta})\mathrm{d}\bar{\boldsymbol{w}}_t\right)\frac{\mathrm{d}\lambda_u}{\mathrm{d}u}\mathrm{d}u=\mathcal{O}\left(\max_{0\leq t\leq T}\tau(t)h^{\frac{3}{2}}\right),$$

$$\sum_{j=-1}^{s-1}b_{i-j}\sum_{k=i-(s-1)}^{i-j-1}\int_{t_k}^{t_{k+1}}\Gamma_1(\boldsymbol{x_\theta})\mathrm{d}t=\mathcal{O}\left(\max_{0\leq t\leq T}\tau^2(t)h^2\right),$$

$$\sum_{j=-1}^{s-1}b_{i-j}\sum_{k=i-(s-1)}^{i-j-1}\int_{t_k}^{t_{k+1}}\Gamma_2(\boldsymbol{x_\theta})\mathrm{d}\bar{\boldsymbol{w}}_t=\mathcal{O}\left(\max_{0\leq t\leq T}\tau(t)h^{\frac{3}{2}}\right),$$

$$\tag{76}$$

and the summation of the above terms is $S_{i+1}^{(1)}=\mathcal{O}\left(\max\limits_{0\leq t\leq T}\tau(t)h^{\frac{3}{2}}\right)$.

The remaining terms of local error are

$$R_{i+1}^{(1)}=\sum_{k=i-(s-1)}^{i-1}\sigma_{t_{i+1}}\left(\int_{t_i}^{t_{i+1}}e^{-\int_{\lambda_u}^{\lambda_{t_{i+1}}}\tau^2(\lambda)\mathrm{d}\lambda}\left(1+\tau^2(u)\right)e^{\lambda_u}\frac{\mathrm{d}\lambda_u}{\mathrm{d}u}\mathrm{d}u\right)\times\left(\int_{t_k}^{t_{k+1}}\Gamma_0(\boldsymbol{x_\theta})\mathrm{d}t\right)$$

$$+\sigma_{t_{i+1}}\int_{t_i}^{t_{i+1}}e^{-\int_{\lambda_u}^{\lambda_{t_{i+1}}}\tau^2(\lambda)\mathrm{d}\lambda}\left(1+\tau^2(u)\right)e^{\lambda_u}\left(\int_{t_i}^{u}\Gamma_0(\boldsymbol{x_\theta})\mathrm{d}t\right)\frac{\mathrm{d}\lambda_u}{\mathrm{d}u}\mathrm{d}u$$

$$-\sum_{j=-1}^{s-1}b_{i-j}\sum_{k=i-(s-1)}^{i-j-1}\int_{t_k}^{t_{k+1}}\Gamma_0(\boldsymbol{x_\theta})\mathrm{d}t,$$

$$\tag{77}$$

which completes the proof. $\qquad\square$

The remaining problem is to estimate the $R_{i+1}^{(1)}$ in Eq. (70). We can further expand the term $\Gamma_0(\boldsymbol{x_\theta})$ as following

$$\Gamma_0(\boldsymbol{x_\theta})(\boldsymbol{x}_t,t)$$

$$=\Gamma_0(\boldsymbol{x_\theta})(\boldsymbol{x}_{t_{i-(s-1)}},t_{i-(s-1)})+\int_{t_{i-(s-1)}}^{t}\left(\Gamma_0\Gamma_0(\boldsymbol{x_\theta})+\Gamma_1\Gamma_0(\boldsymbol{x_\theta})\right)\mathrm{d}t+\int_{t_{i-(s-1)}}^{t}\Gamma_2\Gamma_0(\boldsymbol{x_\theta})\mathrm{d}\bar{\boldsymbol{w}}_t.$$

$$\tag{78}$$

Substituting the expansion of $\Gamma_0(\boldsymbol{x_\theta})$, we perform the approximation of $\tilde{L}_{i+1}$, which is summarized with the following lemma:

**Lemma B.12** (Second step of estimating $L_{i+1}$ in data-prediction reparameterization model). *$R_{i+1}^{(1)}$ in Eq.* (70) *can be decomposed as*

$$R_{i+1}^{(1)}=R_{i+1}^{(2)}+S_{i+1}^{(2)},\tag{79}$$

*where*

$$S_{i+1}^{(2)} = \mathcal{O}\left(\max_{0 \leq t \leq T} \tau(t) h^{\frac{5}{2}}\right)$$

$$R_{i+1}^{(2)}$$

$$= \sigma_{t_{i+1}} \int_{t_i}^{t_{i+1}} e^{-\int_{\lambda_u}^{\lambda_{t_{i+1}}} \tau^2(\lambda) \mathrm{d}\lambda} \left(1 + \tau^2(u)\right) e^{\lambda_u} \left(\int_{t_{i-(s-1)}}^{u} \mathrm{d}u_2\right) \frac{\mathrm{d}\lambda_u}{\mathrm{d}u} \mathrm{d}u \cdot \Gamma_0(\boldsymbol{x_\theta})(\boldsymbol{x}_{t_{i-(s-1)}}, t_{i-(s-1)})$$

$$+ \sigma_{t_{i+1}} \int_{t_i}^{t_{i+1}} e^{-\int_{\lambda_u}^{\lambda_{t_{i+1}}} \tau^2(\lambda) \mathrm{d}\lambda} \left(1 + \tau^2(u)\right) e^{\lambda_u} \left(\int_{t_{i-(s-1)}}^{u} \int_{t_{i-(s-1)}}^{u_2} \Gamma_0 \Gamma_0(\boldsymbol{x_\theta}) \left(\boldsymbol{x}_{u_3}, u_3\right) \mathrm{d}u_3 \mathrm{d}u_2\right) \frac{\mathrm{d}\lambda_u}{\mathrm{d}u} \mathrm{d}u$$

$$- \sum_{j=-1}^{s-1} b_{i-j} \int_{t_{i-(s-1)}}^{t_{i-j}} \mathrm{d}u_2 \times \Gamma_0(\boldsymbol{x_\theta})(\boldsymbol{x}_{t_{i-(s-1)}}, t_{i-(s-1)})$$

$$- \sum_{j=-1}^{s-1} b_{i-j} \int_{t_{i-(s-1)}}^{t_{i-j}} \int_{t_{i-(s-1)}}^{u_2} \Gamma_0 \Gamma_0(\boldsymbol{x_\theta}) \left(\boldsymbol{x}_{u_3}, u_3\right) \mathrm{d}u_3 \mathrm{d}u_2$$

$$\tag{80}$$

*Proof.* We start with decomposing the term $R_{i+1}^{(1)}$

$$R_{i+1}^{(1)}$$

$$= \sigma_{t_{i+1}} \int_{t_i}^{t_{i+1}} e^{-\int_{\lambda_u}^{\lambda_{t_{i+1}}} \tau^2(\lambda) \mathrm{d}\lambda} \left(1 + \tau^2(u)\right) e^{\lambda_u} \left(\int_{t_{i-(s-1)}}^{u} \Gamma_0(\boldsymbol{x_\theta})(\boldsymbol{x}_{u_2}, u_2) \mathrm{d}u_2\right) \frac{\mathrm{d}\lambda_u}{\mathrm{d}u} \mathrm{d}u$$

$$- \sum_{j=-1}^{s-1} b_{i-j} \int_{t_{i-(s-1)}}^{t_{i-j}} \Gamma_0(\boldsymbol{x_\theta})(\boldsymbol{x}_{u_2}, u_2) \mathrm{d}u_2$$

$$= \sigma_{t_{i+1}} \int_{t_i}^{t_{i+1}} e^{-\int_{\lambda_u}^{\lambda_{t_{i+1}}} \tau^2(\lambda) \mathrm{d}\lambda} \left(1 + \tau^2(u)\right) e^{\lambda_u} \left(\int_{t_{i-(s-1)}}^{u} \mathrm{d}u_2\right) \frac{\mathrm{d}\lambda_u}{\mathrm{d}u} \mathrm{d}u \cdot \Gamma_0(\boldsymbol{x_\theta})(\boldsymbol{x}_{t_{i-(s-1)}}, t_{i-(s-1)})$$

$$+ \sigma_{t_{i+1}} \int_{t_i}^{t_{i+1}} e^{-\int_{\lambda_u}^{\lambda_{t_{i+1}}} \tau^2(\lambda) \mathrm{d}\lambda} \left(1 + \tau^2(u)\right) e^{\lambda_u}$$

$$\left(\int_{t_{i-(s-1)}}^{u} \int_{t_{i-(s-1)}}^{u_2} \left(\Gamma_0 \Gamma_0(\boldsymbol{x_\theta}) + \Gamma_1 \Gamma_0(\boldsymbol{x_\theta})\right) \left(\boldsymbol{x}_{u_3}, u_3\right) \mathrm{d}u_3 \mathrm{d}u_2\right) \frac{\mathrm{d}\lambda_u}{\mathrm{d}u} \mathrm{d}u$$

$$+ \sigma_{t_{i+1}} \int_{t_i}^{t_{i+1}} e^{-\int_{\lambda_u}^{\lambda_{t_{i+1}}} \tau^2(\lambda) \mathrm{d}\lambda} \left(1 + \tau^2(u)\right) e^{\lambda_u}$$

$$\left(\int_{t_{i-(s-1)}}^{u} \int_{t_{i-(s-1)}}^{u_2} \Gamma_2 \Gamma_0(\boldsymbol{x_\theta}) \left(\boldsymbol{x}_{u_3}, u_3\right) \mathrm{d}\bar{\boldsymbol{w}}_{u_3} \mathrm{d}u_2\right) \frac{\mathrm{d}\lambda_u}{\mathrm{d}u} \mathrm{d}u$$

$$- \sum_{j=-1}^{s-1} b_{i-j} \int_{t_{i-(s-1)}}^{t_{i-j}} \mathrm{d}u_2 \times \Gamma_0(\boldsymbol{x_\theta})(\boldsymbol{x}_{t_{i-(s-1)}}, t_{i-(s-1)})$$

$$- \sum_{j=-1}^{s-1} b_{i-j} \int_{t_{i-(s-1)}}^{t_{i-j}} \int_{t_{i-(s-1)}}^{u_2} \left(\Gamma_0 \Gamma_0(\boldsymbol{x_\theta}) + \Gamma_1 \Gamma_0(\boldsymbol{x_\theta})\right) \left(\boldsymbol{x}_{u_3}, u_3\right) \mathrm{d}u_3 \mathrm{d}u_2$$

$$- \sum_{j=-1}^{s-1} b_{i-j} \int_{t_{i-(s-1)}}^{t_{i-j}} \int_{t_{i-(s-1)}}^{u_2} \Gamma_2 \Gamma_0(\boldsymbol{x_\theta}) \left(\boldsymbol{x}_{u_3}, u_3\right) \mathrm{d}\bar{\boldsymbol{w}}_{u_3} \mathrm{d}u_2.$$

$$\tag{81}$$

We further estimate the terms with $\Gamma_1$ and $\Gamma_2$.

$$\sigma_{t_{i+1}} \int_{t_i}^{t_{i+1}} e^{-\int_{\lambda_u}^{\lambda_{t_{i+1}}} \tau^2(\lambda)\mathrm{d}\lambda} \left(1 + \tau^2(u)\right) e^{\lambda_u}$$

$$\left(\int_{t_{i-(s-1)}}^{u} \int_{t_{i-(s-1)}}^{u_2} \Gamma_1 \Gamma_0(\boldsymbol{x_\theta})\left(\boldsymbol{x}_{u_3}, u_3\right) \mathrm{d}u_3 \mathrm{d}u_2\right) \frac{\mathrm{d}\lambda_u}{\mathrm{d}u}\mathrm{d}u$$

$$= \mathcal{O}\left(\max_{0\le t\le T} \tau^2(t)h^3\right),$$

$$\sigma_{t_{i+1}} \int_{t_i}^{t_{i+1}} e^{-\int_{\lambda_u}^{\lambda_{t_{i+1}}} \tau^2(\lambda)\mathrm{d}\lambda} \left(1 + \tau^2(u)\right) e^{\lambda_u} \qquad (82)$$

$$\left(\int_{t_{i-(s-1)}}^{u} \int_{t_{i-(s-1)}}^{u_2} \Gamma_2 \Gamma_0(\boldsymbol{x_\theta})\left(\boldsymbol{x}_{u_3}, u_3\right) \mathrm{d}\bar{\boldsymbol{w}}_{u_3} \mathrm{d}u_2\right) \frac{\mathrm{d}\lambda_u}{\mathrm{d}u}\mathrm{d}u$$

$$= \mathcal{O}\left(\max_{0\le t\le T} \tau(t)h^{\frac{5}{2}}\right),$$

$$\sum_{j=-1}^{s-1} b_{i-j} \int_{t_{i-(s-1)}}^{t_{i-j}} \int_{t_{i-(s-1)}}^{u_2} \Gamma_1 \Gamma_0(\boldsymbol{x_\theta})\left(\boldsymbol{x}_{u_3}, u_3\right) \mathrm{d}u_3 \mathrm{d}u_2 = \mathcal{O}\left(\max_{0\le t\le T}\tau^2(t)h^3\right),$$

$$\sum_{j=-1}^{s-1} b_{i-j} \int_{t_{i-(s-1)}}^{t_{i-j}} \int_{t_{i-(s-1)}}^{u_2} \Gamma_2 \Gamma_0(\boldsymbol{x_\theta})\left(\boldsymbol{x}_{u_3}, u_3\right) \mathrm{d}\bar{\boldsymbol{w}}_{u_3} \mathrm{d}u_2 = \mathcal{O}\left(\max_{0\le t\le T}\tau(t)h^{\frac{5}{2}}\right).$$

The summation of the above terms is $S^{(2)} = \mathcal{O}\left(\max_{0\le t\le T}\tau(t)h^{\frac{5}{2}}\right)$. Compared with $S^{(1)}$, this term can be omitted.

The remaining local error is

$$R_{i+1}^{(2)} = \sigma_{t_{i+1}} \int_{t_i}^{t_{i+1}} e^{-\int_{\lambda_u}^{\lambda_{t_{i+1}}} \tau^2(\lambda)\mathrm{d}\lambda} \left(1 + \tau^2(u)\right) e^{\lambda_u} \left(\int_{t_{i-(s-1)}}^{u} \mathrm{d}u_2\right) \frac{\mathrm{d}\lambda_u}{\mathrm{d}u}\mathrm{d}u$$

$$\times \Gamma_0(\boldsymbol{x_\theta})(\boldsymbol{x}_{t_{i-(s-1)}}, t_{i-(s-1)})$$

$$+ \sigma_{t_{i+1}} \int_{t_i}^{t_{i+1}} e^{-\int_{\lambda_u}^{\lambda_{t_{i+1}}} \tau^2(\lambda)\mathrm{d}\lambda} \left(1 + \tau^2(u)\right) e^{\lambda_u}$$

$$\left(\int_{t_{i-(s-1)}}^{u} \int_{t_{i-(s-1)}}^{u_2} \Gamma_0 \Gamma_0(\boldsymbol{x_\theta})\left(\boldsymbol{x}_{u_3}, u_3\right) \mathrm{d}u_3 \mathrm{d}u_2\right) \frac{\mathrm{d}\lambda_u}{\mathrm{d}u}\mathrm{d}u \qquad (83)$$

$$- \sum_{j=-1}^{s-1} b_{i-j} \int_{t_{i-(s-1)}}^{t_{i-j}} \mathrm{d}u_2 \times \Gamma_0(\boldsymbol{x_\theta})(\boldsymbol{x}_{t_{i-(s-1)}}, t_{i-(s-1)})$$

$$- \sum_{j=-1}^{s-1} b_{i-j} \int_{t_{i-(s-1)}}^{t_{i-j}} \int_{t_{i-(s-1)}}^{u_2} \Gamma_0 \Gamma_0(\boldsymbol{x_\theta})\left(\boldsymbol{x}_{u_3}, u_3\right) \mathrm{d}u_3 \mathrm{d}u_2$$

which completes the proof. $\qquad\square$

**Remark 3.** *With Lemma B.11 and B.12, the local error $L_{i+1}$ can be decomposed to the term $S_{i+1} = S_{i+1}^{(1)} + S_{i+1}^{(2)}$ and the term $R_{i+1}^{(2)}$. It is clear that $S_{i+1} = \mathcal{O}\left(\max_{0\le t\le T}\tau(t)h^{\frac{3}{2}}\right)$, and we will show that given $b_{i-j}$ constructed by integral of Lagrange polynomial in Eq. (58) and Eq. (60), $R_{i+1}^{(2)} = \mathcal{O}\left(h^3\right)$.*

By Lemma B.10, $b_k$ constructed by the integral of Lagrange polynomial in Eq. (58) and Eq. (60) satisfies that the coefficient for $\Gamma_0(\boldsymbol{x_\theta})(\boldsymbol{x}_{t_{i-(s-1)}}, t_{i-(s-1)})$

$$\sigma_{t_{i+1}} \int_{t_i}^{t_{i+1}} e^{-\int_{\lambda_u}^{\lambda_{t_{i+1}}} \tau^2(\lambda)\mathrm{d}\lambda} \left(1 + \tau^2(u)\right) e^{\lambda_u} \left(\int_{t_{i-(s-1)}}^{u} \mathrm{d}u_2\right) \frac{\mathrm{d}\lambda_u}{\mathrm{d}u}\mathrm{d}u - \sum_{j=-1}^{s-1} b_{i-j} \int_{t_{i-(s-1)}}^{t_{i-j}} \mathrm{d}u_2,$$

$$\tag{84}$$

equals zero. And the remaining term in $R_{i+1}^{(2)}$ is $\mathcal{O}(h^3)$.

**Remark 4.** *We will show that the local error can be further decomposed such that $L_{i+1} = R_{i+1}^{(s)} + \sum_{j=1}^{s} S_{i+1}^{(j)}$. In this case $S_{i+1} = \sum_{j=1}^{s} S_{i+1}^{(j)}$ is the term such that $S_{i+1} = \mathcal{O}\left(\max_{0 \le t \le T} \tau(t) h^{\frac{3}{2}}\right)$, and we will show that by our constructed $b_{i-j}$, $R_{i+1}^{(s)} = \mathcal{O}\left(h^{s+1}\right)$.*

**Lemma B.13** $(j-th$ *step of estimating $L_{i+1}$ in data-prediction reparameterization model).* *For $j \le s + 1$, $R_{i+1}^{(j-1)}$ in Eq. (70) can be decomposed as*

$$R_{i+1}^{(j-1)} = R_{i+1}^{(j)} + S_{i+1}^{(j)}, \tag{85}$$

*where*

$$S_{i+1}^{(j)} = \mathcal{O}\left(\max_{0 \le t \le T} \tau(t) h^{\frac{2j+1}{2}}\right)$$

$$R_{i+1}^{(j)}$$

$$= \sigma_{t_{i+1}} \int_{t_i}^{t_{i+1}} e^{-\int_{\lambda_u}^{\lambda_{t_{i+1}}} \tau^2(\lambda)\mathrm{d}\lambda} \left(1 + \tau^2(u)\right) e^{\lambda_u} \left(\int_{t_{i-(s-1)}}^{u} \int_{t_{i-(s-1)}}^{u_2} \cdots \int_{t_{i-(s-1)}}^{u_{j-1}} \mathrm{d}u_j \cdots \mathrm{d}u_3\mathrm{d}u_2\right) \frac{\mathrm{d}\lambda_u}{\mathrm{d}u}\mathrm{d}u$$

$$\cdot \overbrace{\Gamma_0 \cdots \Gamma_0}^{j-1}(\boldsymbol{x_\theta})(\boldsymbol{x}_{t_{i-(s-1)}}, t_{i-(s-1)})$$

$$+ \sigma_{t_{i+1}} \int_{t_i}^{t_{i+1}} e^{-\int_{\lambda_u}^{\lambda_{t_{i+1}}} \tau^2(\lambda)\mathrm{d}\lambda} \left(1 + \tau^2(u)\right) e^{\lambda_u}$$

$$\left(\int_{t_{i-(s-1)}}^{u} \int_{t_{i-(s-1)}}^{u_2} \cdots \int_{t_{i-(s-1)}}^{u_j} \overbrace{\Gamma_0 \cdots \Gamma_0}^{j}(\boldsymbol{x_\theta}) \left(\boldsymbol{x}_{u_{j+1}}, u_{j+1}\right) \mathrm{d}u_{j+1} \cdots \mathrm{d}u_3\mathrm{d}u_2\right) \frac{\mathrm{d}\lambda_u}{\mathrm{d}u}\mathrm{d}u$$

$$- \sum_{j=-1}^{s-1} b_{i-j} \int_{t_{i-(s-1)}}^{t_{i-j}} \int_{t_{i-(s-1)}}^{u_2} \cdots \int_{t_{i-(s-1)}}^{u_{j-1}} \mathrm{d}u_j \cdots \mathrm{d}u_3\mathrm{d}u_2 \cdot \overbrace{\Gamma_0 \cdots \Gamma_0}^{j-1}(\boldsymbol{x_\theta})(\boldsymbol{x}_{t_{i-(s-1)}}, t_{i-(s-1)})$$

$$- \sum_{j=-1}^{s-1} b_{i-j} \int_{t_{i-(s-1)}}^{t_{i-j}} \int_{t_{i-(s-1)}}^{u_2} \cdots \int_{t_{i-(s-1)}}^{u_j} \overbrace{\Gamma_0 \cdots \Gamma_0}^{j}(\boldsymbol{x_\theta}) \left(\boldsymbol{x}_{u_{j+1}}, u_{j+1}\right) \mathrm{d}u_{j+1} \cdots \mathrm{d}u_3\mathrm{d}u_2$$

$$\tag{86}$$

*Furthermore, given that $b_k$ is constructed by the integral of Lagrange polynomial in Eq. (58) and Eq. (60), $R_{i+1}^{(j)} = \mathcal{O}(h^{j+1})$*

**Sketch of the proof** (1) Use the Itô formula Eq. (67) to expand $\overbrace{\Gamma_0 \cdots \Gamma_0}^{j-1}(\boldsymbol{x_\theta})$. (2) Use Lemma B.9 to estimate the stochastic term $S^{(j)}$. For the remaining term $R^{(j)}$, by Lemma B.10, $b_k$ constructed by the integral of Lagrange polynomial in Eq. (58) and Eq. (60) satisfies that the coefficient before

$$\overbrace{\Gamma_0\cdots\Gamma_0}^{j-1}(\boldsymbol{x_\theta})(\boldsymbol{x}_{t_{i-(s-1)}},t_{i-(s-1)})$$

$$\sigma_{t_{i+1}}\int_{t_i}^{t_{i+1}}e^{-\int_{\lambda_u}^{\lambda_{t_{i+1}}}\tau^2(\lambda)\mathrm{d}\lambda}\left(1+\tau^2(u)\right)e^{\lambda_u}\left(\int_{t_{i-(s-1)}}^u\int_{t_{i-(s-1)}}^{u_2}\cdots\int_{t_{i-(s-1)}}^{u_{j-1}}\mathrm{d}u_j\cdots\mathrm{d}u_3\mathrm{d}u_2\right)\frac{\mathrm{d}\lambda_u}{\mathrm{d}u}\mathrm{d}u$$

$$-\sum_{j=-1}^{s-1}b_{i-j}\int_{t_{i-(s-1)}}^{t_{i-j}}\int_{t_{i-(s-1)}}^{u_2}\cdots\int_{t_{i-(s-1)}}^{u_{j-1}}\mathrm{d}u_j\cdots\mathrm{d}u_3\mathrm{d}u_2.$$

(87)

equals zero. And the remaining term in $R_{i+1}^{(j)}$ is $\mathcal{O}(h^{j+1})$.

The process can be repeated until the coefficient before $\overbrace{\Gamma_0\cdots\Gamma_0}^{s}(\boldsymbol{x_\theta})(\boldsymbol{x}_{t_{i-(s-1)}},t_{i-(s-1)})$ is

$$\sigma_{t_{i+1}}\int_{t_i}^{t_{i+1}}e^{-\int_{\lambda_u}^{\lambda_{t_{i+1}}}\tau^2(\lambda)\mathrm{d}\lambda}\left(1+\tau^2(u)\right)e^{\lambda_u}\left(\int_{t_{i-(s-1)}}^u\int_{t_{i-(s-1)}}^{u_2}\cdots\int_{t_{i-(s-1)}}^{u_s}\mathrm{d}u_{s+1}\cdots\mathrm{d}u_3\mathrm{d}u_2\right)\frac{\mathrm{d}\lambda_u}{\mathrm{d}u}\mathrm{d}u$$

$$-\sum_{j=-1}^{s-1}b_{i-j}\int_{t_{i-(s-1)}}^{t_{i-j}}\int_{t_{i-(s-1)}}^{u_2}\cdots\int_{t_{i-(s-1)}}^{u_s}\mathrm{d}u_{s+1}\cdots\mathrm{d}u_3\mathrm{d}u_2.$$

(88)

which equals zero. And the remaining term $R_{i+1}^{s+1}$ is $\mathcal{O}(h^{s+2})$.

We conclude with the proof of Lemma B.7 and B.8.

**Proof for Lemma B.8 (Convergence for $s$-step *SA-Corrector*)**   The stochastic term $S_{i+1}$ can be estimated as $\mathcal{O}\left(\max_{0\leq t\leq T}\tau(t)h^{\frac{3}{2}}\right)$. Lemma B.10 prove that with $b_{i-j}$ defined in Theorem 5.2, the coefficients of Eq. (75), Eq. (84), Eq. (87) and Eq. (88) equal zero. Thus the deterministic term $R_{i+1}$ can be estimated as $\mathcal{O}(h^{s+2})$. The proof is completed.

**Proof for Lemma B.7 (Convergence for $s$-step *SA-Predictor*)**   The stochastic term $S_{i+1}$ can be estimated as $\mathcal{O}\left(\max_{0\leq t\leq T}\tau(t)h^{\frac{3}{2}}\right)$ from Eq. (76) and Eq. (82). Lemma B.10 prove that with $b_{i-j}$ defined in Theorem 5.1, the coefficients of Eq. (75), Eq. (84), Eq. (87) and Eq. (88) equal zero except for the last term. This is because in $s$-step *SA-Predictor* we only have $s$ points in contrast to $s+1$ points in $s$-step *SA-Corrector*, for which we can only obtain the first s equalities in Lemma B.10. Thus the deterministic term $R_{i+1}$ can be estimated as $\mathcal{O}(h^{s+1})$. The proof is completed.

## B.5   Relationship with Existing Samplers

### B.5.1   Relationship with DDIM

DDIM [22] generates samples through the following process:

$$\boldsymbol{x}_{t_{i+1}}=\alpha_{t_{i+1}}\left(\frac{\boldsymbol{x}_{t_i}-\sigma_{t_i}\boldsymbol{\epsilon_\theta}(\boldsymbol{x}_{t_i},t_i)}{\alpha_{t_i}}\right)+\sqrt{1-\alpha_{t_{i+1}}^2-\hat{\sigma}_{t_i}^2}\boldsymbol{\epsilon_\theta}(\boldsymbol{x}_{t_i},t_i)+\hat{\sigma}_{t_i}\boldsymbol{\xi},\qquad(89)$$

where $\boldsymbol{\xi}\sim\mathcal{N}(\boldsymbol{0},\boldsymbol{I})$, $\hat{\sigma}_{t_i}$ is a variable parameter. In practice, DDIM introduces a parameter $\eta$ such that when $\eta=0$, the sampling process becomes deterministic and when $\eta=1$, the sampling process coincides with original DDPM [2]. Specifically, $\hat{\sigma}_{t_i}=\eta\sqrt{\frac{1-\alpha_{t_{i+1}}^2}{1-\alpha_{t_i}^2}\left(1-\frac{\alpha_{t_i}^2}{\alpha_{t_{i+1}}^2}\right)}$.

**Corollary 5.3**   For any $\eta$ in DDIM, there exists a $\tau_\eta(t):\mathbb{R}\to\mathbb{R}$ which is a piecewise constant function such that DDIM-$\eta$ coincides with our 1-step *SA-Predictor* when $\tau(t)=\tau_\eta(t)$ with data parameterization of our variance-controlled diffusion SDE.

*Proof.* Our 1-step *SA-Predictor* when $\tau(t) = \tau, t \in [t_i, t_{i+1}]$ with data parameterization of our variance-controlled diffusion SDE is

$$
\begin{aligned}
\boldsymbol{x}_{t_{i+1}} =& \frac{\sigma_{t_{i+1}}}{\sigma_{t_i}} e^{-\tau^2\left(\lambda_{t_{i+1}} - \lambda_{t_i}\right)} \boldsymbol{x}_{t_i} + \alpha_{t_{i+1}} \left(1 - e^{-\left(1+\tau^2\right)\left(\lambda_{t_{i+1}} - \lambda_{t_i}\right)}\right) \boldsymbol{x}_{\boldsymbol{\theta}}(\boldsymbol{x}_{t_i}, t_i) \\
& + \sigma_{t_{i+1}} \sqrt{1 - e^{-2\tau^2\left(\lambda_{t_{i+1}} - \lambda_{t_i}\right)}} \xi.
\end{aligned}
\tag{90}
$$

DDIM-$\eta$ generates samples through the following process

$$
\boldsymbol{x}_{t_{i+1}} = \alpha_{t_{i+1}} \boldsymbol{x}_{\boldsymbol{\theta}}\left(\boldsymbol{x}_{t_i}, t_i\right) + \sqrt{1 - \alpha_{t_{i+1}}^2 - \hat{\sigma}_{t_i}^2} \boldsymbol{\epsilon}_{\boldsymbol{\theta}}(\boldsymbol{x}_{t_i}, t_i) + \hat{\sigma}_{t_i} \boldsymbol{\xi}, \hat{\sigma}_{t_i} = \eta \sqrt{\frac{1 - \alpha_{t_{i+1}}^2}{1 - \alpha_{t_i}^2} \left(1 - \frac{\alpha_{t_i}^2}{\alpha_{t_{i+1}}^2}\right)}.
\tag{91}
$$

If we substitute $\hat{\sigma}_{t_i}$ with $\sigma_{t_{i+1}} \sqrt{1 - e^{-2\tau^2\left(\lambda_{t_{i+1}} - \lambda_{t_i}\right)}}$, we can verify that $\sqrt{1 - \alpha_{t_{i+1}}^2 - \hat{\sigma}_{t_i}^2} = \sigma_{t_{i+1}} e^{-\tau^2\left(\lambda_{t_{i+1}} - \lambda_{t_i}\right)}$. The DDIM-$\eta$ then becomes

$$
\begin{aligned}
\boldsymbol{x}_{t_{i+1}} =& \alpha_{t_{i+1}} \boldsymbol{x}_{\boldsymbol{\theta}}\left(\boldsymbol{x}_{t_i}, t_i\right) + \sigma_{t_{i+1}} e^{-\tau^2\left(\lambda_{t_{i+1}} - \lambda_{t_i}\right)} \left(\frac{\boldsymbol{x}_{t_i} - \alpha_{t_i} \boldsymbol{x}_{\boldsymbol{\theta}}(\boldsymbol{x}_{t_i}, t_i)}{\sigma_{t_i}}\right) \\
& + \sigma_{t_{i+1}} \sqrt{1 - e^{-2\tau^2\left(\lambda_{t_{i+1}} - \lambda_{t_i}\right)}} \boldsymbol{\xi} \\
=& \frac{\sigma_{t_{i+1}}}{\sigma_{t_i}} e^{-\tau^2\left(\lambda_{t_{i+1}} - \lambda_{t_i}\right)} \boldsymbol{x}_{t_i} + \left(\alpha_{t_{i+1}} - \frac{\alpha_{t_i}}{\sigma_{t_i}} \sigma_{t_{i+1}} e^{-\tau^2\left(\lambda_{t_{i+1}} - \lambda_{t_i}\right)}\right) \boldsymbol{x}_{\boldsymbol{\theta}}\left(\boldsymbol{x}_{t_i}, t_i\right) \\
& + \sigma_{t_{i+1}} \sqrt{1 - e^{-2\tau^2\left(\lambda_{t_{i+1}} - \lambda_{t_i}\right)}} \boldsymbol{\xi} \\
=& \frac{\sigma_{t_{i+1}}}{\sigma_{t_i}} e^{-\tau^2\left(\lambda_{t_{i+1}} - \lambda_{t_i}\right)} \boldsymbol{x}_{t_i} + \alpha_{t_{i+1}} \left(1 - e^{-\left(1+\tau^2\right)\left(\lambda_{t_{i+1}} - \lambda_{t_i}\right)}\right) \boldsymbol{x}_{\boldsymbol{\theta}}(\boldsymbol{x}_{t_i}, t_i) \\
& + \sigma_{t_{i+1}} \sqrt{1 - e^{-2\tau^2\left(\lambda_{t_{i+1}} - \lambda_{t_i}\right)}} \boldsymbol{\xi},
\end{aligned}
\tag{92}
$$

which is exactly the same with our 1-step *SA-Predictor*. To find the $\tau_\eta$, we solve the relationship

$$
\eta \sqrt{\frac{1 - \alpha_{t_{i+1}}^2}{1 - \alpha_{t_i}^2} \left(1 - \frac{\alpha_{t_i}^2}{\alpha_{t_{i+1}}^2}\right)} = \sigma_{t_{i+1}} \sqrt{1 - e^{-2\tau_\eta^2\left(\lambda_{t_{i+1}} - \lambda_{t_i}\right)}}.
\tag{93}
$$

The relationship between $\tau$ and $\eta$ is

$$
\eta = \sigma_{t_i} \sqrt{\frac{1 - e^{-2\tau_\eta^2\left(\lambda_{t_{i+1}} - \lambda_{t_i}\right)}}{1 - \frac{\alpha_{t_i}^2}{\alpha_{t_{i+1}}^2}}}, \tau_\eta = \sqrt{\frac{\log\left(1 - \frac{\eta^2}{\sigma_{t_i}^2} \left(1 - \frac{\alpha_{t_i}^2}{\alpha_{t_{i+1}}^2}\right)\right)}{-2\left(\lambda_{t_{i+1}} - \lambda_{t_i}\right)}}.
\tag{94}
$$

$\square$

In a concurrent paper [31], Lu *et al.* prove the result that their SDE-DPM-Solver++1 coincides with DDIM with a special $\eta$. Their result is a special case of Corollary 5.3 when $\tau_\eta \equiv 1$ and $\eta$ take a special value, while our result holds for arbitrary $\eta$.

### B.5.2   Relationship with DPM-Solver++(2M)

DPM-Solver++ [31] is a high-order solver which solves diffusion ODEs for guided sampling. DPM-Solver++(2M) is equivalent to the 2-step Adams-Bashforth scheme combined with the exponential integrator. While our 2-step *SA-Predictor* is also equivalent to the 2-step Adams-Bashforth scheme combined with the exponential integrator when $\tau(t) \equiv 0$. Thus DPM-Solver++(2M) is a special case of our 2-step *SA-Predictor* when $\tau(t) \equiv 0$.

### B.5.3  Relationship with UniPC

UniPC [24] is a unified predictor-corrector framework for solving diffusion ODEs. Specifically, UniPC-p uses a p-step Adams-Bashforth scheme combined with the exponential integrator as a predictor and a p-step Adams-Moulton scheme combined with the exponential integrator as a corrector. While our p-step *SA-Predictor* is also equivalent to the p-step Adams-Bashforth scheme combined with the exponential integrator when $\tau(t) \equiv 0$ and our p-step *SA-Corrector* is also equivalent to the p-step Adams-Moulton scheme combined with the exponential integrator when $\tau(t) \equiv 0$. Thus UniPC-p is a special case of our *SA-Solver* when $\tau(t) \equiv 0$ with predictor step p, corrector step p in Algorithm 1.

## C  Selection on the Magnitude of Stochasticity

In this section, we will show that we choose $\tau(t) \equiv 1$ in a number of NFEs. We will show that under certain conditions, the upper bound of KL divergence between the marginal distribution and the true distribution can be minimized when $\tau(t) \equiv 1$.

Let $p_t(\boldsymbol{x})$ denotes the marginal distribution of $\boldsymbol{x}_t$, by Proposition 4.1, we know that for any bounded measurable function $\tau(t) : [0, T] \to \mathbb{R}$, the following Reverse SDEs

$$\mathrm{d}\boldsymbol{x}_t = \left[ f(t)\boldsymbol{x}_t - \left( \frac{1 + \tau^2(t)}{2} \right) g^2(t)\nabla_{\boldsymbol{x}}\log p_t(\boldsymbol{x}_t) \right]\mathrm{d}t + \tau(t)g(t)\mathrm{d}\bar{\boldsymbol{w}}_t, \quad \boldsymbol{x}_T \sim p_T(\boldsymbol{x}_T), \quad (95)$$

have the same marginal probability distributions. In practice, we substitute $\nabla_{\boldsymbol{x}}\log p_t(\boldsymbol{x}_t)$ with $\boldsymbol{s}_{\boldsymbol{\theta}}(\boldsymbol{x}_t, t)$ and substitute $p_T(\boldsymbol{x}_T)$ wiht $\pi$ to sample the reverse SDE.

$$\mathrm{d}\boldsymbol{x}_t^{\boldsymbol{\theta}} = \left[ f(t)\boldsymbol{x}_t^{\boldsymbol{\theta}} - \left( \frac{1 + \tau^2(t)}{2} \right) g^2(t)\boldsymbol{s}_{\boldsymbol{\theta}}(\boldsymbol{x}_t^{\boldsymbol{\theta}}, t) \right]\mathrm{d}t + \tau(t)g(t)\mathrm{d}\bar{\boldsymbol{w}}_t, \quad \boldsymbol{x}_T^{\boldsymbol{\theta}} \sim \pi, \quad (96)$$

where $\pi$ is a known distribution, specifically here a Gaussian. We have the following theorem under the Assumption in Appendix A in [3].

**Theorem C.1.** *Let $p = p_0$ be the data distribution, which is the distribution if we sample from the ground truth reverse SDE (54) at time $0$. Let $p_{\boldsymbol{\theta}}^{\tau(t)}$ be the distribution if we sample from the practical reverse SDE (96) at time $0$. Under the assumptions above, we have*

$$D_{KL}\left( p \| p_{\boldsymbol{\theta}}^{\tau(t)} \right)$$

$$\leq D_{KL}\left( p_T \| \pi \right) + \frac{1}{8}\int_0^T \mathbb{E}_{p_t(\boldsymbol{x})}\left[ \left( \tau(t) + \frac{1}{\tau(t)} \right)^2 g^2(t)\|\nabla_{\boldsymbol{x}}\log p_t(\boldsymbol{x}) - \boldsymbol{s}_{\boldsymbol{\theta}}(\boldsymbol{x}, t)\|^2 \right]\mathrm{d}t. \quad (97)$$

*This evidence lower bound (ELBO) is minimized when $\tau(t) \equiv 1$.*

*Proof.* Denote the path measure of Eq. (95) and Eq. (96) as $\boldsymbol{\mu}$ and $\boldsymbol{\nu}$ respectively. Both $\boldsymbol{\mu}$ and $\boldsymbol{\nu}$ are uniquely determined by the corresponding SDEs due to assumptions. Consider a Markov kernel $K\left( \{\boldsymbol{z}_t\}_{t\in[0,T]}, \boldsymbol{y} \right) = \delta(\boldsymbol{z}_0 = \boldsymbol{y})$. Thus we have the following result

$$\int K\left( \{\boldsymbol{x}_t\}_{t\in[0,T]}, \boldsymbol{x} \right)\mathrm{d}\boldsymbol{\mu}\left( \{\boldsymbol{x}_t\}_{t\in[0,T]} \right) = p_0(\boldsymbol{x}), \quad (98)$$

$$\int K\left( \{\boldsymbol{x}_t^{\boldsymbol{\theta}}\}_{t\in[0,T]}, \boldsymbol{x} \right)\mathrm{d}\boldsymbol{\nu}\left( \{\boldsymbol{x}_t^{\boldsymbol{\theta}}\}_{t\in[0,T]} \right) = p_{\boldsymbol{\theta}}^{\tau(t)}(\boldsymbol{x}). \quad (99)$$

By data processing inequality for KL divergence

$$D_{KL}\left( p \| p_{\boldsymbol{\theta}}^{\tau(t)} \right) = D_{KL}\left( p_0 \| p_{\boldsymbol{\theta}}^{\tau(t)} \right)$$

$$= D_{KL}\left( \int K\left( \{\boldsymbol{x}_t\}_{t\in[0,T]}, \boldsymbol{x} \right)\mathrm{d}\mu\left( \{\boldsymbol{x}_t\}_{t\in[0,T]} \right) \Big\| \int K\left( \{\boldsymbol{x}_t^{\boldsymbol{\theta}}\}_{t\in[0,T]}, \boldsymbol{x} \right)\mathrm{d}\nu\left( \{\boldsymbol{x}_t^{\boldsymbol{\theta}}\}_{t\in[0,T]} \right) \right)$$

$$\leq D_{KL}\left( \boldsymbol{\mu} \| \boldsymbol{\nu} \right). \quad (100)$$

By the chain rule of KL divergence, we have

$$D_{KL}\left(\boldsymbol{\mu}\|\boldsymbol{\nu}\right) = D_{KL}\left(p_T\|\pi\right) + \mathbb{E}_{\boldsymbol{z}\sim p_T}\left[D_{KL}\left(\boldsymbol{\mu}(\cdot|\boldsymbol{x}_T = \boldsymbol{z})\|\boldsymbol{\nu}(\cdot|\boldsymbol{x}_T^{\boldsymbol{\theta}} = \boldsymbol{z}))\right]. \qquad (101)$$

By Girsanov Thoerem, $D_{KL}\left(\boldsymbol{\mu}(\cdot|\boldsymbol{x}_T = \boldsymbol{z})\|\boldsymbol{\nu}(\cdot|\boldsymbol{x}_T^{\boldsymbol{\theta}} = \boldsymbol{z})\right)$ can be computed as

$$\begin{aligned}
&D_{KL}\left(\boldsymbol{\mu}(\cdot|\boldsymbol{x}_T = \boldsymbol{z})\|\boldsymbol{\nu}(\cdot|\boldsymbol{x}_T^{\boldsymbol{\theta}} = \boldsymbol{z})\right) \\
=&\mathbb{E}_{\mu}\left[\int_0^T \frac{1}{2}\left(\tau(t) + \frac{1}{\tau(t)}\right) g(t)\left(\nabla_{\boldsymbol{x}}\log p_t(\boldsymbol{x}) - \boldsymbol{s}_{\boldsymbol{\theta}}(\boldsymbol{x}, t)\right)\mathrm{d}\bar{\boldsymbol{w}}_t\right] \\
&+ \mathbb{E}_{\mu}\left[\frac{1}{2}\int_0^T \frac{1}{4}\left(\tau(t) + \frac{1}{\tau(t)}\right)^2 g^2(t)\|\nabla_{\boldsymbol{x}}\log p_t(\boldsymbol{x}) - \boldsymbol{s}_{\boldsymbol{\theta}}(\boldsymbol{x}, t)\|^2\mathrm{d}t\right] \\
=&\frac{1}{8}\int_0^T \mathbb{E}_{p_t(\boldsymbol{x})}\left[\left(\tau(t) + \frac{1}{\tau(t)}\right)^2 g^2(t)\|\nabla_{\boldsymbol{x}}\log p_t(\boldsymbol{x}) - \boldsymbol{s}_{\boldsymbol{\theta}}(\boldsymbol{x}, t)\|^2\right]\mathrm{d}t
\end{aligned} \qquad (102)$$

$\square$

## D    Implementation Details

For our 2-step SA-Predictor and 1-step SA-Corrector, we find that the coefficient will degenerate to a simple case.

For 2-step SA-Predictor, assume on $[t_i, t_{i+1}]$, $\tau(t) = \tau$ is a constant,

$$b_i = e^{-\lambda_{t_{i+1}}\tau^2}\sigma_{t_{i+1}}(1 + \tau^2)\int_{\lambda_{t_i}}^{\lambda_{t_{i+1}}} e^{(1+\tau^2)\lambda}\frac{\lambda - \lambda_{t_{i-1}}}{\lambda_{t_i} - \lambda_{t_{i-1}}}\mathrm{d}\lambda, \qquad (103)$$

$$b_{i-1} = e^{-\lambda_{t_{i+1}}\tau^2}\sigma_{t_{i+1}}(1 + \tau^2)\int_{\lambda_{t_i}}^{\lambda_{t_{i+1}}} e^{(1+\tau^2)\lambda}\frac{\lambda - \lambda_{t_i}}{\lambda_{t_{i-1}} - \lambda_{t_i}}\mathrm{d}\lambda, \qquad (104)$$

we have

$$\begin{aligned}
\boldsymbol{x}_{t_{i+1}} =&\frac{\sigma_{t_{i+1}}}{\sigma_{t_i}}e^{-\int_{\lambda_{t_i}}^{\lambda_{t_{i+1}}}\tau^2(\tilde{\lambda})\mathrm{d}\tilde{\lambda}}\boldsymbol{x}_{t_i} + b_i\boldsymbol{x}_{\boldsymbol{\theta}}(\boldsymbol{x}_{t_i}, t_i) + b_{i-1}\boldsymbol{x}_{\boldsymbol{\theta}}(\boldsymbol{x}_{t_{i-1}}, t_{i-1}) + \tilde{\sigma}_i\boldsymbol{\xi} \\
=&\frac{\sigma_{t_{i+1}}}{\sigma_{t_i}}e^{-\int_{\lambda_{t_i}}^{\lambda_{t_{i+1}}}\tau^2(\tilde{\lambda})\mathrm{d}\tilde{\lambda}}\boldsymbol{x}_{t_i} + (b_i + b_{i-1})\boldsymbol{x}_{\boldsymbol{\theta}}(\boldsymbol{x}_{t_i}, t_i) \\
&- b_{i-1}(\boldsymbol{x}_{\boldsymbol{\theta}}(\boldsymbol{x}_{t_i}, t_i) - \boldsymbol{x}_{\boldsymbol{\theta}}(\boldsymbol{x}_{t_{i-1}}, t_{i-1})) + \tilde{\sigma}_i\boldsymbol{\xi}.
\end{aligned} \qquad (105)$$

Let $h = \lambda_{t_{i+1}} - \lambda_{t_i}$, we have

$$\begin{aligned}
b_i + b_{i-1} &= e^{-\lambda_{t_{i+1}}\tau^2}\sigma_{t_{i+1}}(1 + \tau^2)\int_{\lambda_{t_i}}^{\lambda_{t_{i+1}}} e^{(1+\tau^2)\lambda}\mathrm{d}\lambda \\
&= \alpha_{t_{i+1}}(1 - e^{-h(1+\tau^2)}) \\
b_{i-1} &= \alpha_{t_{i+1}}\frac{e^{-(1+\tau^2)h} + (1+\tau^2)h - 1}{(1+\tau^2)(\lambda_{t_i} - \lambda_{t_{i-1}})} \\
&= \frac{\alpha_{t_{i+1}}}{\lambda_{t_i} - \lambda_{t_{i-1}}}\frac{1 - (1+\tau^2)h + \frac{1}{2}(1+\tau^2)^2h^2 + \mathcal{O}(h^3) + (1+\tau^2)h - 1}{1+\tau^2} \\
&= \frac{\alpha_{t_{i+1}}}{\lambda_{t_i} - \lambda_{t_{i-1}}}\frac{1}{2}(1+\tau^2)h^2 + \mathcal{O}(h^3)
\end{aligned} \qquad (106)$$

Thus we implement $\hat{b}_{i-1}$ as $\frac{\alpha_{t_{i+1}}}{\lambda_{t_i} - \lambda_{t_{i-1}}}\frac{1}{2}(1+\tau^2)h^2$ and $\hat{b}_i$ as $\alpha_{t_{i+1}}(1 - e^{-h(1+\tau^2)}) - \hat{b}_{i-1}$. Note that substituting $b_i, b_{i-1}$ as $\hat{b}_i, \hat{b}_{i-1}$ will maintain the convergence order result of 2-step SA-Predictor since the modified term is $\mathcal{O}(h^3)$. The implementation detail for 1-step SA-Corrector is technically the same.

# E  Experiment Details

## E.1  Details on $\tau(t)$, Predictor Steps and Corrector Steps

**CIFAR10 32x32**  For the CIFAR10 experiment in Section 6.4, we use the pretrained baseline-unconditional-VE model[4] from [27].  It's an unconditional model with VE noise schedule. To fairly compare with results in [27], we use a piecewise constant function $\tau(t)$ inspired by [27]. Concretely, denoting $\sigma_t^{EDM} = \frac{\sigma_t}{\alpha_t}$, our $\tau(t)$ is set to be a constant $\tau$ in the interval $[(\sigma^{EDM})^{-1}(0.05), (\sigma^{EDM})^{-1}(1)]$ and to be zero outside the interval. We find empirically that this piecewise constant function setting makes our *SA-Solver* converge better, especially in large noise scale cases. We use a 3-step SA-Predictor and a 3-step SA-Corrector. For the CIFAR10 experiment in Section 6.2 and 6.5, we also use the piecewise constant function $\tau(t)$ as above. The predictor steps and corrector steps vary to verify the effectiveness of our proposed method in Section 6.2, while they are both set to be 3-steps in Section 6.5.

**ImageNet 64x64**  For the ImageNet 64x64 experiment in Section 6.4, we use the pretrained model[5] from [4]. It's a conditional model with VP cosine noise schedule. To fairly compare with results in [27], we use a piecewise constant function $\tau(t)$ inspired by [27]. Concretely, denoting $\sigma_t^{EDM} = \frac{\sigma_t}{\alpha_t}$, our $\tau(t)$ is set to be a constant $\tau$ in the interval $[(\sigma^{EDM})^{-1}(0.05), (\sigma^{EDM})^{-1}(50)]$ and to be zero outside the interval. We find empirically that this piecewise constant function setting makes our *SA-Solver* converge better, especially in large noise scale cases. We use a 3-step SA-Predictor and a 3-step SA-Corrector.

**Other experiments**  For other experiments, we use a constant function $\tau(t) \equiv \tau$. It's generally not the optimal choice for each individual task, thus further fine-grained tuning has the potential to improve the results. We aim to report the result of our *SA-Solver* without extra hyperparameter tuning. We use a 3-step SA-Predictor and a 3-step SA-Corrector under 20 NFEs and 2-step SA-Predictor and a 1-step SA-Corrector beyond 20 NFEs.

## E.2  Details on Pretrained Models and Settings

**CIFAR10 32x32**  For the CIFAR10 experiment, we use the pretrained baseline-unconditional-VE model[6] from [27]. It's an unconditional model with VE noise schedule. To fairly compare with results in [27], we follow the time step schedule in it. Specifically, we set $\sigma_{min} = 0.02$ and $\sigma_{max} = 80$ and select the step by $\sigma_i = (\sigma_{max}^{\frac{1}{7}} + \frac{i}{N-1}(\sigma_{min}^{\frac{1}{7}} - \sigma_{max}^{\frac{1}{7}}))^7$ for *SA-Solver* and UniPC. We directly report the results of the deterministic sampler and stochastic sampler of EDM. To make it a strong baseline, we report the results of the optimal setting for 4 hyper-parameters $\{S_{churn}, S_{tmin}, S_{tmax}, S_{noise}\}$ and report its lowest observed FID. While for *SA-Solver* and UniPC, we report the averaged observed FID.

**ImageNet 64x64**  For the ImageNet 64x64 experiment, we use the pretrained model[7] from [4]. It's a conditional model with VP cosine noise schedule. To fairly compare with results in [27], we follow the time step schedule in it and use conditional sampling. Specifically, we set $\sigma_{min} = 0.0064$ and $\sigma_{max} = 80$ and select the step by $\sigma_i = (\sigma_{max}^{\frac{1}{7}} + \frac{i}{N-1}(\sigma_{min}^{\frac{1}{7}} - \sigma_{max}^{\frac{1}{7}}))^7$ for *SA-Solver*, UniPC, DPM-Solver and DDIM. We directly report the results of the deterministic sampler and stochastic sampler of EDM. To make it a strong baseline, we report the results of the optimal setting for 4 hyper-parameters $\{S_{churn}, S_{tmin}, S_{tmax}, S_{noise}\}$ and report its lowest observed FID. While for *SA-Solver*, UniPC, DPM-Solver and DDIM, we report the averaged observed FID.

---

[4] `https://nvlabs-fi-cdn.nvidia.com/edm/pretrained/baseline/` `baseline-cifar10-32x32-uncond-ve.pkl`

[5] `https://openaipublic.blob.core.windows.net/diffusion/jul-2021/64x64_diffusion.pt`

[6] `https://nvlabs-fi-cdn.nvidia.com/edm/pretrained/baseline/` `baseline-cifar10-32x32-uncond-ve.pkl`

[7] `https://openaipublic.blob.core.windows.net/diffusion/jul-2021/64x64_diffusion.pt`

**ImageNet 256x256**   For the ImageNet 256x256 experiment, we use three different pretrained models:LDM[8](VP, handcrafted noise schedule) from [5], DiT-XL/2[9](VP, linear noise schedule) from [41], Min-SNR[10](VP, cosine noise schedule) from [42]. We use classifier-free guidance of scale $s = 1.5$ and a uniform time step schedule because it's the most common setting for guided sampling for ImageNet 256x256.

**ImageNet 512x512**   For the ImageNet 256x256 experiment, we use the pre-trained model: DiT-XL/2[11] from [41]. We use classifier-free guidance of scale $s = 1.5$ and a uniform time step schedule following the settings of DiT [41].

**LSUN Bedroom 256x256**   For the LSUN Bedroom 256x256 experiment, we use the pretrained model[12] from [4]. We use unconditional sampling and a uniform lambda step schedule from [23].

# F   Additional Results

We include the detailed FID results in Figure 1, Figure 2 and Figure 4 in the tables 4 to 14. The ablation study shows that stochasticity indeed helps improve sample quality. We find that for small NFEs, the magnitude of stochasticity should be small while for large NFEs, large magnitude of stochasticity helps improve sample quality. It can also be observed that in latent space, SDE converges faster as in Table 13. With only 10 NFEs, $\tau = 0.6$ is better than $\tau = 0$. With 20 NFEs, our *SA-Solver* can achieve 3.87 FID, which outperforms all ODE samplers even with far more steps.

Table 4: Sample quality measured by FID ↓ on CIFAR10 32x32 dataset (VE-baseline model from [27]) varying the number of function evaluations (NFE). For the results from EDM[†], we reported its lowest observed FID.

| Method \ NFE | 11 | 15 | 23 | 31 | 47 | 63 | 95 |
|---|---|---|---|---|---|---|---|
| DDIM($\eta = 0$) | 18.28 | 12.23 | 7.93 | 6.45 | 5.27 | 4.83 | 4.42 |
| DPM-Solver | 9.26 | 5.13 | 4.52 | 4.30 | 4.02 | 3.97 | 3.94 |
| UniPC | **6.42** | 5.02 | 4.19 | 4.00 | 3.91 | 3.90 | 3.89 |
| EDM(ODE)[†] | 13.46 | 5.62 | 4.04 | 3.82 | 3.79 | 3.80 | 3.79 |
| EDM(SDE)[†] | 23.94 | 8.94 | 4.73 | 3.95 | 3.59 | 3.36 | 3.06 |
| SA-Solver | 6.46 | **4.91** | **3.77** | **3.40** | **2.92** | **2.74** | **2.63** |

Table 5: Sample quality measured by FID ↓ on CIFAR10 32x32 dataset (VE-baseline model from [27]) varying the number of function evaluations (NFE) and the magnitude of stochasticity ($\tau$).

| SA-Solver \ NFE | 11 | 15 | 23 | 31 | 47 | 63 | 95 |
|---|---|---|---|---|---|---|---|
| $\tau = 0.0$ | **6.46** | 5.06 | 4.22 | 4.02 | 3.93 | 3.92 | 3.91 |
| $\tau = 0.2$ | 6.54 | 5.01 | 4.14 | 3.95 | 3.89 | 3.84 | 3.83 |
| $\tau = 0.4$ | 6.79 | **4.91** | 4.03 | 3.81 | 3.76 | 3.74 | 3.67 |
| $\tau = 0.6$ | 7.34 | 4.91 | 3.85 | 3.65 | 3.60 | 3.56 | 3.57 |
| $\tau = 0.8$ | 8.61 | 5.28 | **3.77** | 3.48 | 3.45 | 3.43 | 3.50 |
| $\tau = 1.0$ | 10.89 | 6.52 | 3.98 | **3.40** | 3.21 | 3.25 | 3.29 |
| $\tau = 1.2$ | 14.49 | 9.33 | 5.19 | 3.69 | 3.00 | 3.03 | 3.07 |
| $\tau = 1.4$ | 20.19 | 13.76 | 7.60 | 4.91 | **2.92** | 2.86 | 2.93 |
| $\tau = 1.6$ | 27.90 | 20.51 | 11.89 | 8.07 | 3.25 | **2.74** | 2.80 |
| $\tau = 1.8$ | 36.26 | 29.43 | 18.13 | 14.00 | 4.60 | 2.83 | **2.63** |

---

[8]https://ommer-lab.com/files/latent-diffusion/nitro/cin/model.ckpt

[9]https://dl.fbaipublicfiles.com/DiT/models/DiT-XL-2-256x256.pt

[10]https://github.com/TiankaiHang/Min-SNR-Diffusion-Training/releases/download/v0.0.0/ema_0.9999_xl.pt

[11]https://dl.fbaipublicfiles.com/DiT/models/DiT-XL-2-512x512.pt

[12]https://openaipublic.blob.core.windows.net/diffusion/jul-2021/lsun_bedroom.pt

Table 6: Sample quality measured by FID ↓ on ImageNet 64x64 dataset (model from [4]) varying the number of function evaluations (NFE). For the results from EDM[†], we reported its lowest observed FID.

| Method \ NFE | 15 | 23 | 31 | 47 | 63 | 95 |
|---|---|---|---|---|---|---|
| DDIM($\eta = 0$) | 8.48 | 5.39 | 4.27 | 3.46 | 3.17 | 2.95 |
| DPM-Solver | 3.49 | 3.04 | 2.88 | 2.80 | 2.76 | 2.74 |
| UniPC | 3.51 | 2.84 | 2.75 | 2.72 | 2.71 | 2.72 |
| EDM(ODE)[†] | 4.78 | 3.12 | 2.84 | 2.73 | 2.73 | 2.67 |
| EDM(SDE)[†] | 8.94 | 4.30 | 3.40 | 2.72 | 2.44 | 2.22 |
| SA-Solver | **3.41** | **2.61** | **2.23** | **1.95** | **1.88** | **1.81** |

Table 7: Sample quality measured by FID ↓ on ImageNet 64x64 dataset (model from [4]) varying the number of function evaluations (NFE) and the magnitude of stochasticity ($\tau$).

| SA-Solver \ NFE | 15 | 23 | 31 | 47 | 63 | 95 |
|---|---|---|---|---|---|---|
| $\tau = 0.0$ | 3.48 | 2.72 | 2.72 | 2.66 | 2.64 | 2.71 |
| $\tau = 0.2$ | **3.41** | 2.80 | 2.63 | 2.63 | 2.64 | 2.60 |
| $\tau = 0.4$ | 3.52 | 2.70 | 2.51 | 2.51 | 2.49 | 2.49 |
| $\tau = 0.6$ | 3.98 | **2.61** | 2.44 | 2.39 | 2.34 | 2.35 |
| $\tau = 0.8$ | 5.80 | 2.68 | 2.32 | 2.24 | 2.19 | 2.21 |
| $\tau = 1.0$ | 10.06 | 3.38 | **2.23** | 2.09 | 2.08 | 2.08 |
| $\tau = 1.2$ | 18.39 | 5.52 | 2.52 | **1.95** | 1.97 | 2.00 |
| $\tau = 1.4$ | 32.42 | 10.37 | 3.83 | 2.05 | 1.89 | 1.89 |
| $\tau = 1.6$ | 52.31 | 19.64 | 7.10 | 2.60 | **1.88** | **1.81** |

## G Additional Samples

We include additional samples in this section. In Figure 5 and Figure 6 we compare samples of our proposed *SA-Solver* with other diffusion samplers. In Figure 7 and Figure 8, we compare samples of our proposed *SA-Solver* under different NFEs and $\tau$. In Figure 9 and Figure 10, we compare samples of our proposed *SA-Solver* with other diffusion samplers on text-to-image tasks. Our *SA-Solver* can generate more diverse samples with more details.

Table 8: Sample quality measured by FID ↓ on CIFAR10 32x32 dataset (model trained by ourselves; see Section 6.5) varying the sampling method and the training epoch.

| method (NFE = 31) \ epoch | 1250 | 1300 | 1350 | 1400 | 1450 | 1500 |
|---|---|---|---|---|---|---|
| DDIM | 39.32 | 29.79 | 19.59 | 12.98 | 8.63 | 6.64 |
| DPM-Solver | 30.57 | 22.11 | 13.85 | 8.85 | 5.68 | 4.55 |
| EDM(ODE) | 27.51 | 19.82 | 12.37 | 8.03 | 5.33 | 4.32 |
| SA-Solver($\tau = 0.6$) | 20.55 | 14.89 | 9.71 | 6.55 | 4.61 | 4.08 |
| SA-Solver($\tau = 1.0$) | **13.62** | **10.01** | **6.79** | **4.81** | **3.70** | **3.47** |

Table 9: Sample quality measured by FID ↓ on ImageNet 256x256 dataset (model trained by ourselves; see Section 6.5) varying the sampling method and the training epoch.

| method (NFE = 40) \ epoch | 50 | 100 | 150 | 200 | 250 |
|---|---|---|---|---|---|
| DDIM | 19.40 | 9.61 | 6.75 | 5.86 | 5.12 |
| DPM-Solver | 18.75 | 8.96 | 6.15 | 5.28 | 4.62 |
| SA-Solver($\tau = 0.4$) | 17.93 | 8.39 | 5.69 | 4.84 | 4.24 |
| SA-Solver($\tau = 0.8$) | **16.57** | **7.54** | **5.15** | **4.48** | **3.99** |

Table 10: Sample quality measured by FID ↓ on ImageNet 256x256 dataset(model from [5]) varying the number of function evaluations (NFE).

| Method \ NFE | 5 | 10 | 20 | 40 | 60 | 80 | 100 |
|---|---|---|---|---|---|---|---|
| DDIM($\eta = 0$) | 58.68 | 16.32 | 6.82 | 4.71 | 4.45 | 4.28 | 4.23 |
| DPM-Solver | 166.88 | 6.19 | 5.51 | 4.17 | 4.18 | 4.21 | 4.15 |
| UniPC | 12.79 | 4.96 | 4.21 | 4.14 | 4.12 | 4.09 | 4.10 |
| DDIM($\eta = 1$) | 138.91 | 50.05 | 14.60 | 6.09 | 4.56 | 4.12 | 3.87 |
| SA-Solver | **11.46** | **4.82** | **3.88** | **3.47** | **3.37** | **3.37** | **3.33** |

Table 11: Ablation study on the effect of the magnitude of stochasticity using *SA-Solver*. Sample quality measured by FID ↓ on CIFAR10 32x32 dataset(model from [27]) varying the number of function evaluations (NFE) and the magnitude of stochasticity($\tau$).

| $\tau$ \ NFE | 15 | 23 | 31 | 47 | 63 | 95 | 127 |
|---|---|---|---|---|---|---|---|
| 0 | **4.84** | 4.11 | 3.94 | 3.86 | 3.88 | 3.87 | 3.87 |
| 0.2 | 4.96 | 4.04 | 3.84 | 3.75 | 3.74 | 3.79 | 3.75 |
| 0.4 | 5.27 | 4.00 | 3.87 | 3.64 | 3.71 | 3.70 | 3.62 |
| 0.6 | 6.05 | **3.95** | 3.61 | 3.49 | 3.46 | 3.53 | 3.43 |
| 0.8 | 7.40 | 4.12 | 3.53 | 3.28 | 3.37 | 3.32 | 3.30 |
| 1.0 | 10.00 | 4.49 | **3.41** | 3.18 | 3.24 | 3.17 | 3.15 |
| 1.2 | 13.58 | 5.14 | 3.59 | 3.10 | 3.02 | 2.97 | 3.05 |
| 1.4 | 17.88 | 6.55 | 3.94 | **3.04** | **3.01** | **2.89** | 2.95 |
| 1.6 | 22.42 | 8.44 | 4.69 | 3.20 | 3.02 | 2.94 | **2.89** |

Table 12: Ablation study on the effect of the magnitude of stochasticity using *SA-Solver*. Sample quality measured by FID ↓ on ImageNet 64x64 dataset(model from [4]) varying the number of function evaluations (NFE) and the magnitude of stochasticity($\tau$).

| $\tau$ \ NFE | 20 | 40 | 60 | 80 | 100 |
|---|---|---|---|---|---|
| 0 | **3.30** | 2.83 | 2.78 | 2.79 | 2.82 |
| 0.2 | 3.32 | 2.77 | 2.72 | 2.74 | 2.79 |
| 0.4 | 3.37 | 2.68 | 2.63 | 2.62 | 2.59 |
| 0.6 | 3.61 | 2.57 | 2.49 | 2.49 | 2.47 |
| 0.8 | 4.19 | **2.51** | 2.40 | 2.34 | 2.30 |
| 1.0 | 5.55 | 2.54 | 2.32 | 2.21 | 2.20 |
| 1.2 | 7.93 | 2.77 | **2.29** | **2.14** | 2.14 |
| 1.4 | 11.55 | 3.20 | 2.40 | **2.14** | **2.08** |
| 1.6 | 16.15 | 3.97 | 2.60 | 2.20 | 2.09 |

Table 13: Ablation study on the effect of the magnitude of stochasticity using *SA-Solver*. Sample quality measured by FID ↓ on ImageNet 256x256 dataset(model from [5]) varying the number of function evaluations (NFE) and the magnitude of stochasticity($\tau$).

| $\tau$ \ NFE | 5 | 10 | 20 | 40 | 60 | 80 | 100 |
|---|---|---|---|---|---|---|---|
| 0 | **11.46** | 5.04 | 4.30 | 4.16 | 4.12 | 4.10 | 4.16 |
| 0.2 | 11.88 | 4.89 | 4.29 | 4.05 | 4.02 | 4.01 | 4.03 |
| 0.4 | 12.69 | 4.84 | 4.14 | 3.86 | 3.84 | 3.83 | 3.84 |
| 0.6 | 14.84 | **4.82** | 3.99 | 3.63 | 3.62 | 3.63 | 3.61 |
| 0.8 | 18.82 | 5.09 | **3.87** | 3.55 | 3.50 | 3.47 | 3.47 |
| 1.0 | 25.96 | 6.06 | 3.88 | **3.47** | 3.41 | 3.39 | 3.38 |
| 1.2 | 37.20 | 8.23 | 3.92 | **3.47** | **3.37** | **3.37** | **3.33** |
| 1.4 | 53.03 | 12.93 | 4.08 | 3.53 | 3.40 | 3.38 | 3.36 |
| 1.6 | 71.30 | 24.08 | 4.43 | 3.56 | 3.44 | 3.45 | 3.33 |

Table 14: Ablation study on the effect of the magnitude of stochasticity using *SA-Solver*. Sample quality measured by FID ↓ on LSUN Bedroom 256x256 dataset(model from [4]) varying the number of function evaluations (NFE) and the magnitude of stochasticity($\tau$).

| $\tau$ \ NFE | 20 | 40 | 60 | 80 | 100 |
|---|---|---|---|---|---|
| 0 | 3.60 | 3.14 | 3.06 | 3.09 | 3.07 |
| 0.2 | **3.51** | 3.12 | 3.00 | 2.99 | 2.99 |
| 0.4 | 3.70 | 3.09 | 2.97 | 3.03 | 3.16 |
| 0.6 | 4.10 | **3.08** | **2.95** | 2.99 | 3.03 |
| 0.8 | 4.75 | 3.11 | 2.97 | 2.89 | 2.99 |
| 1.0 | 6.18 | 3.28 | 2.98 | 2.90 | **2.91** |
| 1.2 | 8.54 | 3.53 | 3.12 | **2.86** | 3.00 |
| 1.4 | 12.14 | 4.25 | 3.24 | 2.98 | 2.93 |
| 1.6 | 16.63 | 5.50 | 3.75 | 3.18 | 3.10 |

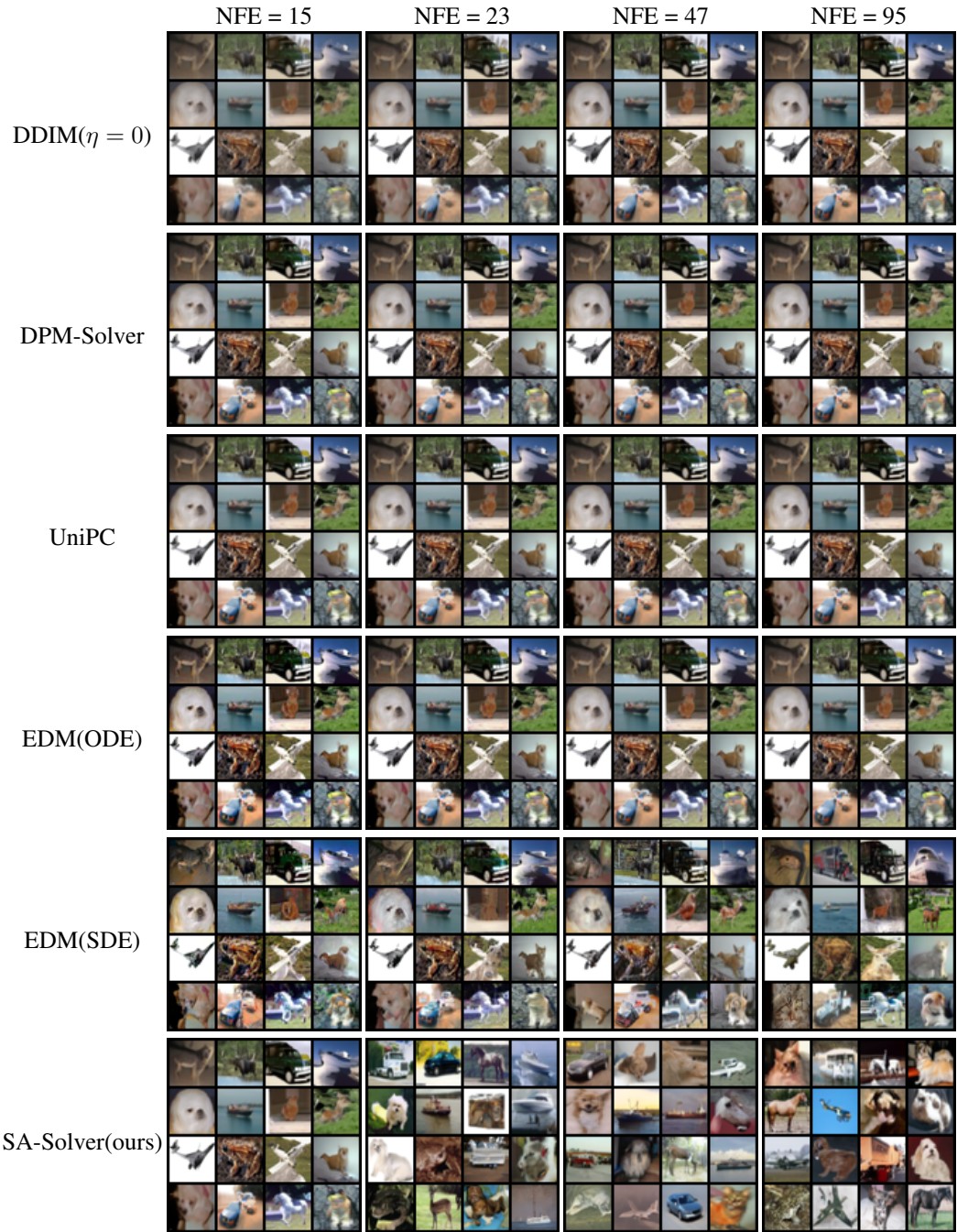

Figure 5: Samples by DDIM, DPM-Solver, UniPC, EDM(ODE), EDM(SDE) and our SA-Solver with 15, 23, 47, 95 NFEs with the same random seed from CIFAR10 32x32 VE baseline model [27]

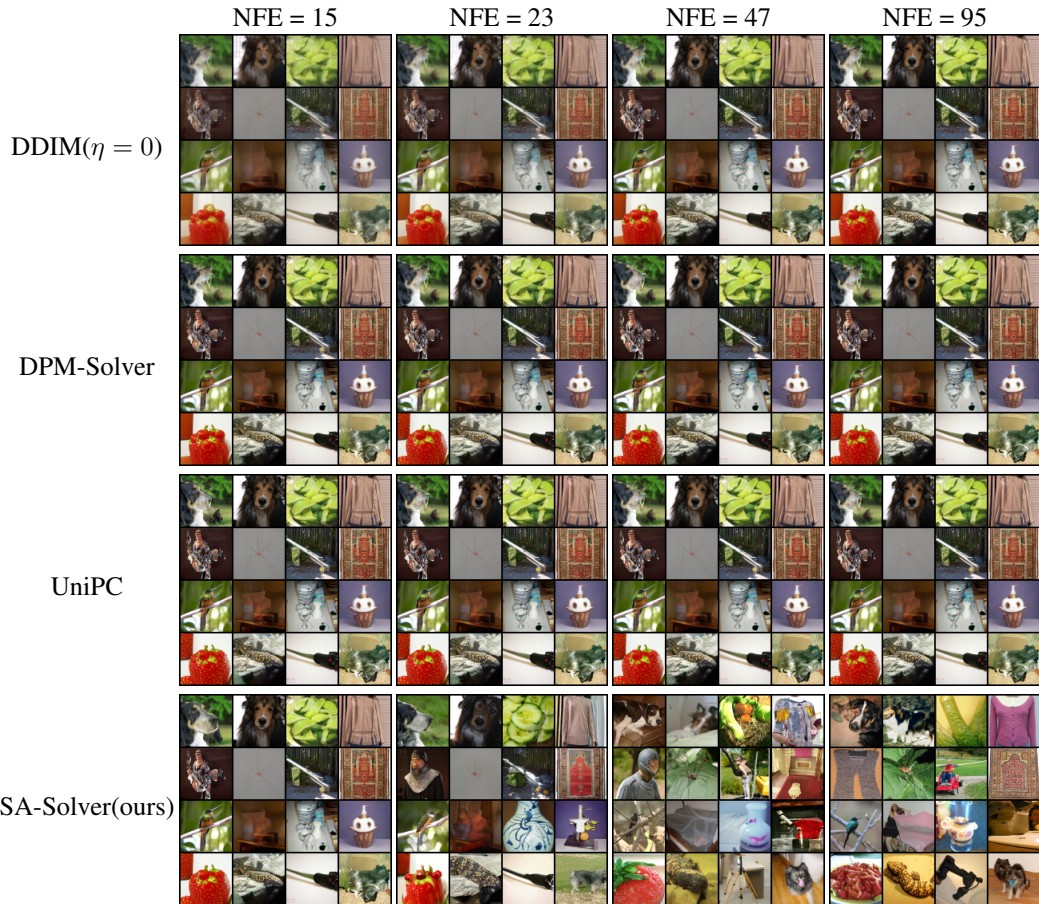

Figure 6: Samples by DDIM, DPM-Solver, UniPC, and our SA-Solver with 15, 23, 47, 95 NFEs with the same random seed from ImageNet 64x64 model [27](conditional sampling)

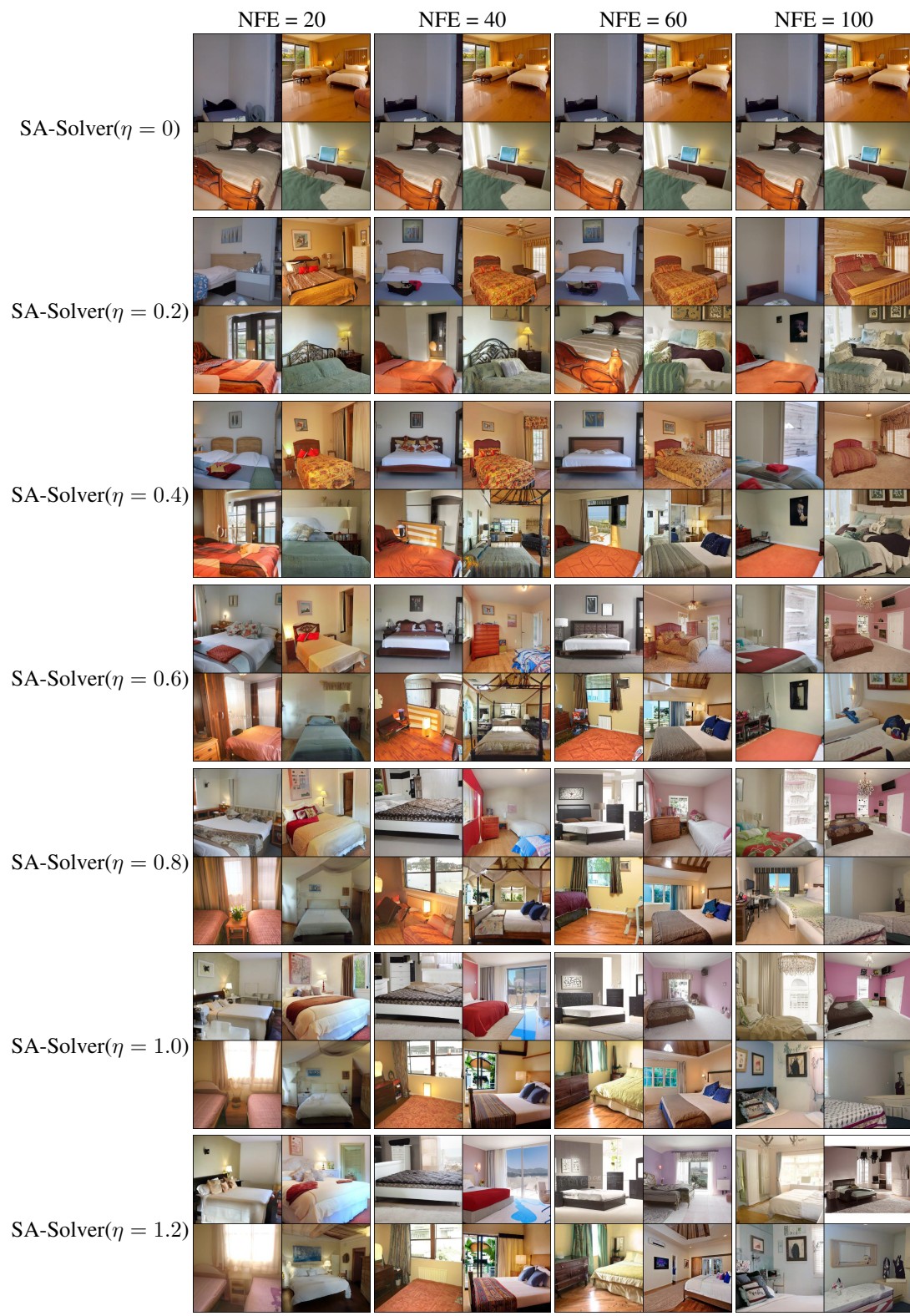

Figure 7: Samples by SA-Solver with 20, 40, 60, 100 NFEs varying stochasticity($\tau$) with the same random seed from LSUN-Bedroom 256x256 model [4](unconditional sampling).

NFE = 60

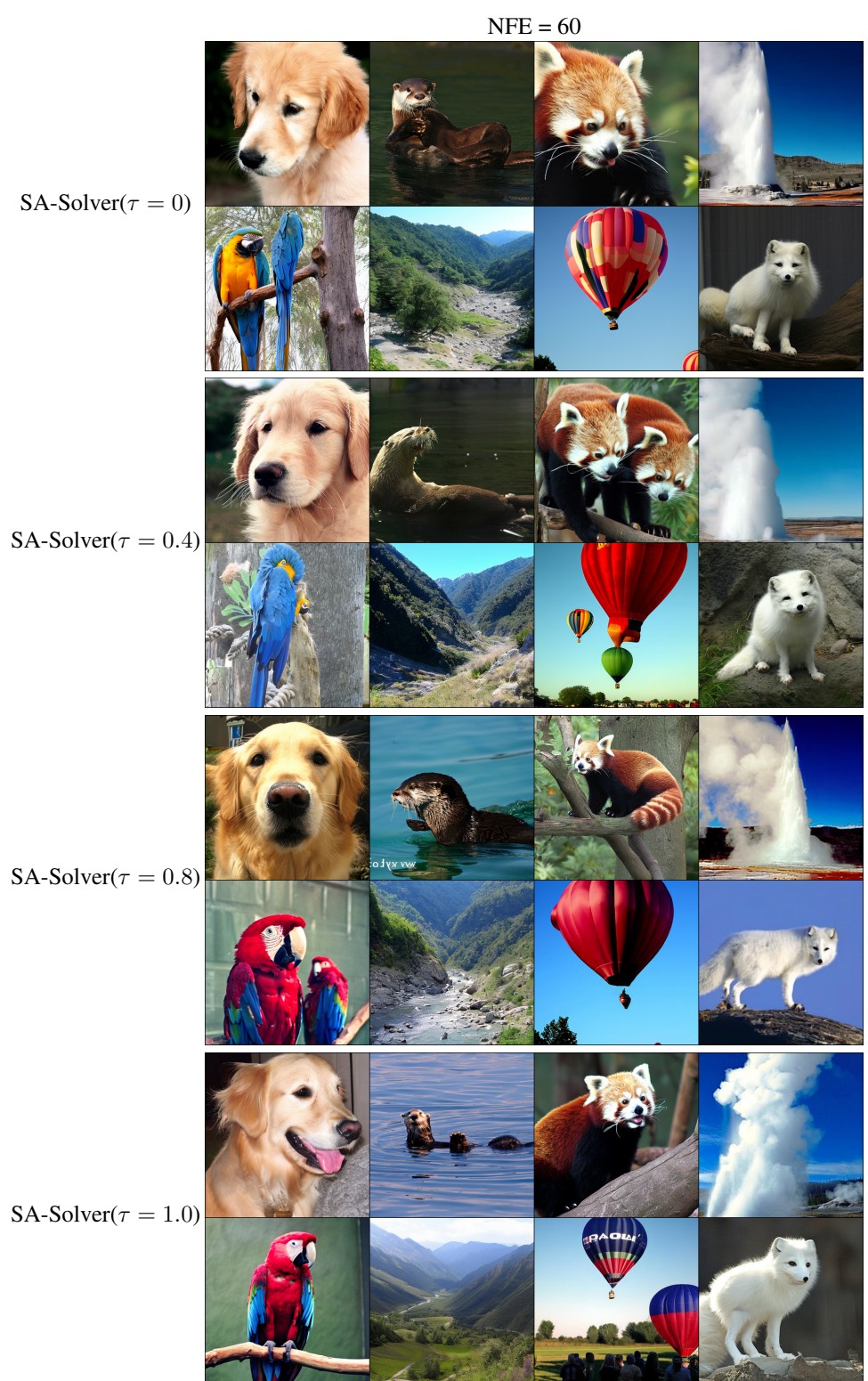

Figure 8: Samples by SA-Solver with 60 NFEs varying stochasticity($\tau$) with the same random seed from ImageNet 512x512 DiT model [41] with classifer-free guidance scale $s = 4.0$(default setting to show image).

|                | NFE = 20 | NFE = 50 | NFE = 100 |
|----------------|----------|----------|-----------|

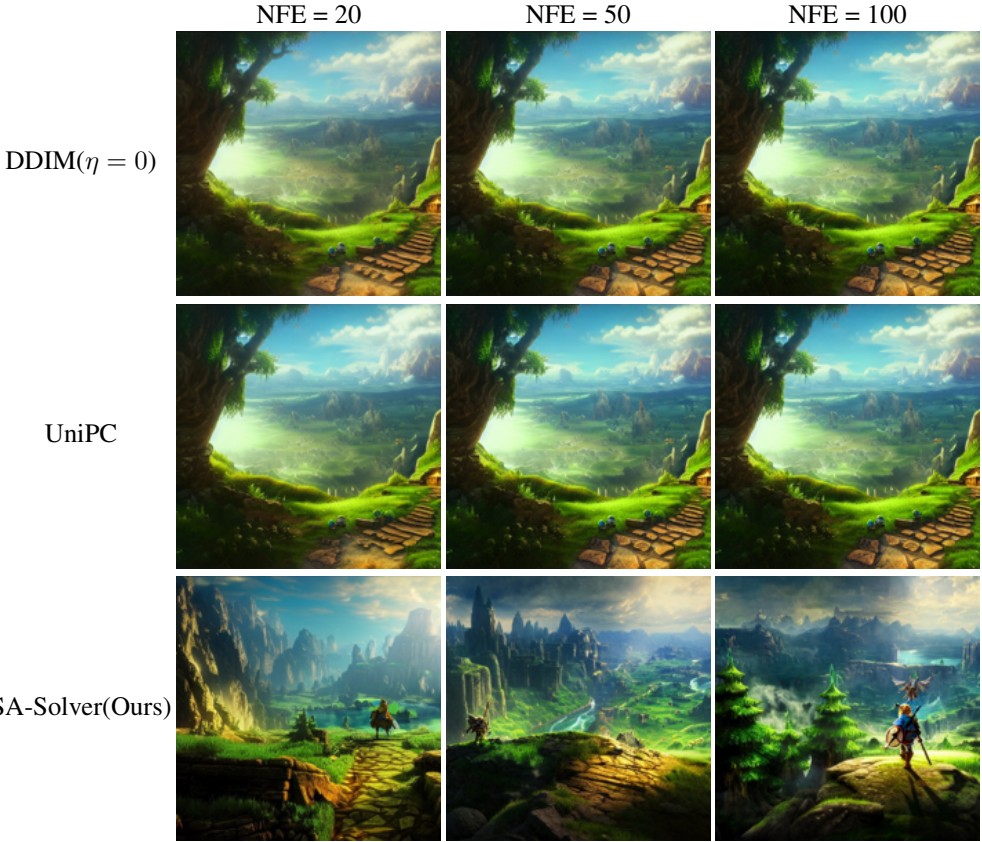

Figure 9: Samples using Stable-Diffusion v1.5 [5] with a classifier-free guidance scale 7.5 with different solvers and NFEs. Prompt:The Legend of Zelda landscape atmospheric, hyper realistic, 8k, epic composition, cinematic, octane render, artstation landscape vista photography by Carr Clifton Galen Rowell, 16K resolution, Landscape veduta photo by Dustin Lefevre tdraw, 8k resolution, detailed landscape painting by Ivan Shishkin, DeviantArt, Flickr, rendered in Enscape, Miyazaki, Nausicaa Ghibli, Breath of The Wild, 4k detailed post processing, artstation, rendering by octane, unreal engine.

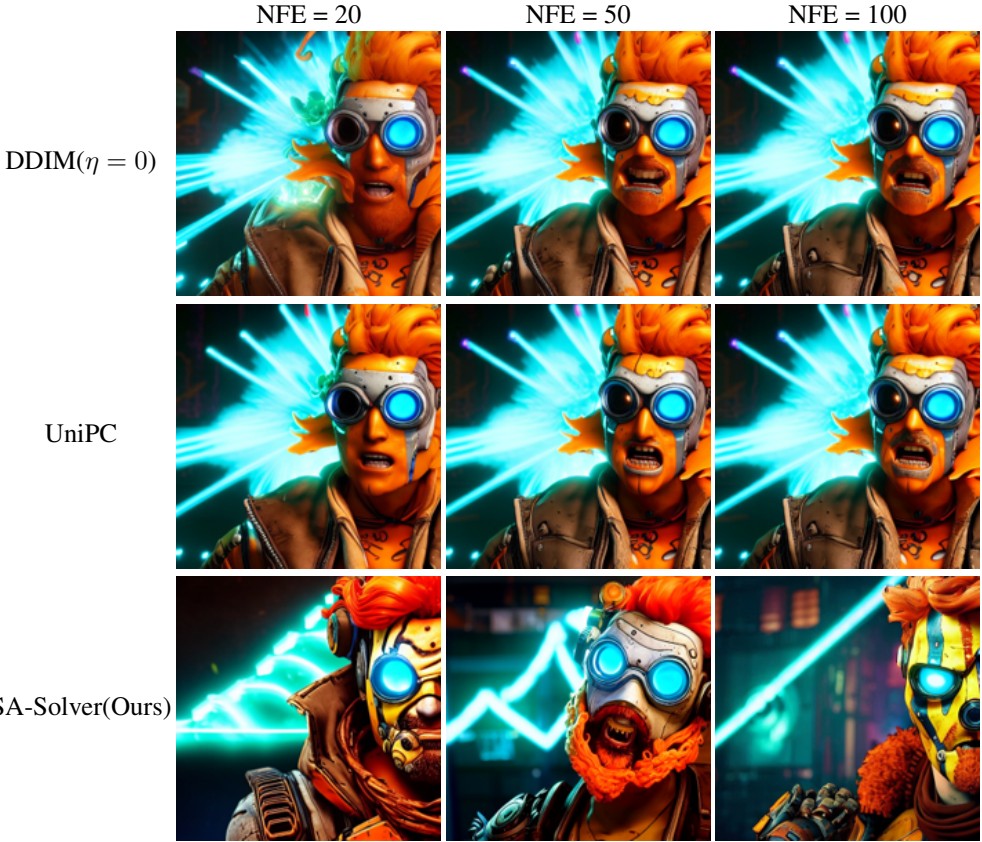

Figure 10: Samples using Stable-Diffusion v1.5 [5] with a classifier-free guidance scale 7.5 with different solvers and NFEs. Prompt:glowwave portrait of curly orange haired mad scientist man from borderlands 3, au naturel, hyper detailed, digital art, trending in artstation, cinematic lighting, studio quality, smooth render, unreal engine 5 rendered, octane rendered, art style by pixar dreamworks warner bros disney riot games and overwatch.

