# OpenReview forum: "SA-Solver: Stochastic Adams Solver for Fast Sampling of Diffusion Models"
_NeurIPS.cc/2023/Conference — NeurIPS 2023 poster_

### Official Review · Reviewer_vQfu · 2023-06-17

**Soundness:** 3 good
**Presentation:** 2 fair
**Contribution:** 3 good
**Rating:** 6
**Confidence:** 3

**Summary:**

The authors propose a sampling method for diffusion probabilistic models by solving an alternative SDE with the same marginal distribution. Approximation techniques are applied in constructing the computationally efficient solver and corrector . The results show a superior FID with less function evaluations.

**Strengths:**

1. The construction of the alternative SDE and the approximation techniques, including score function approximation, change of variable and Lagrange interpolation, provide a complete framework for efficient DFM generation.
2. The proposed method can achieve lower FID scores with less function evaluations which is promising.

**Weaknesses:**

1. The paper content seems not well organized yet.
2. In the appendix, for other methods, with different NFEs, the image content is consistent. But for the proposed method, the image content varies a lot. It might be caused by the inaccurate approximations.

**Questions:**

1. Could you elaborate on why the images are not consistent for different NFEs in your method?

**Limitations:**

The authors did not address the limitations.

There is no potential negative societal impact of their work.

---

> ### Author Rebuttal · Authors · 2023-08-09
>
> First, we would like to thank you for taking the time to carefully review our paper, acknowledgment of our novel contributions, and the insightful questions. Below we respond to the questions:
>
> Q1: The paper content seems not well organized yet.
>
> A1: Thank you for your sincere advice. The paper is organized in the following order. The introduction part is contained in Section 1. In Sections 2 and 3, we introduce the background and related work on the fast sampling of diffusion models. In Section 4, we propose a family of diffusion SDEs called variance controlled diffusion SDEs which shares the same marginal probability distribution. In Section 5, we introduce our SA-Solver, which utilizes the stochastic Adams method to solve the SDEs we proposed in Section 4. In Section 6, we conduct several ablation studies and experiments to demonstrate the effectiveness of our method. We will carefully finalize our draft to make it more readable.
>
> Q2: In the appendix, for other methods, with different NFEs, the image content is consistent. But for the proposed method, the image content varies a lot. It might be caused by inaccurate approximations.
>
> A2: Thank you for noting such an interesting phenomenon. In contrast to inaccurate approximations, we attribute the inconsistent content under different NFEs to the stochasticity of diffusion SDEs.
>
> Concretely, for diffusion ODEs, owing to its deterministic formulation, there is a one-to-one correspondence between starting noise and generated image, so that the same generated contents within the same starting noises under different NFEs (as shown in Figure 4). However, the one-to-one correspondence property does not hold for diffusion SDEs as it injected extra randomness during the sampling procedure. Under the same random seed, different NFEs in the setting of the stochastic sampler will sample different numbers of Gaussian noises, which is the reason the image content varies. This explains the observed inconsistent contents under different NFEs for stochastic sampling methods i.e., EDM (SDE) and our SA-Solver in Figure 4 in Appendix.

---

### Official Review · Reviewer_zg3j · 2023-07-04

**Soundness:** 3 good
**Presentation:** 3 good
**Contribution:** 3 good
**Rating:** 7
**Confidence:** 5

**Summary:**

This paper extends UniPC to a stochastic manner to obtain a faster sampler called SA-Solver for diffusion models. Specifically, the authors start from the formulation of diffusion SDE and derive the SA-Predictor and SA-Corrector. Extensive experiments on CIFAR10, ImageNet, LSUN, etc demonstrate the effectiveness of the proposed SA-Solver.

**Strengths:**

1. It is interesting to consider sampling from the SDE instead of ODE. This opens up a new direction for the fast sampling of DPMs. I am surprised that this simple practice can improve the sampling quality so much.
2. The writing is clear and the derivation of SA-Solver is easy to understand. I especially appreciate Section 5.3, where the relationship with UniPC helps me to understand the proposed method at a glance.
3. The experiments are thorough and convincing. Most required experiments are included, such as unconditional/conditional sampling with data/noise prediction with different image resolutions.

**Weaknesses:**

1. It is questionable whether the novelty of SA-Solver is enough. As said in L228, SA-Solver can be viewed as an extension of UniPC with a non-zero $\tau(t)$. However, I think the idea to introduce stochasticity back to the sampling of the diffusion models also contributes to the novelty. I just feel worried that this issue might be raised by other reviewers.
2. Some experiments are lacking, such as comparing the convergence of different methods on COCO using a pre-trained text-to-image diffusion model. For now, only some qualitative results are provided. It would be better to also demonstrate some quantitative comparisons.
3. Some minor format issues: too much space below Table 1. Maybe the authors can try to make it looks better.

**Questions:**

See the weaknesses.

**Limitations:**

The limitations have been fully discussed.

---

> ### Author Rebuttal · Authors · 2023-08-09
>
> First, we would like to thank you for taking the time to carefully review our paper, acknowledgment of our novel contributions, and the insightful questions. Below we respond to the questions:
>
> Q1: It is questionable whether the novelty of SA-Solver is enough.
>
> A1: As we have claimed in lines 38-39, "adding properly scaled noise in the diffusion SDE may facilitate the quality of generated data". Thus we focus on developing a solver for diffusion SDEs in this paper, which has not been well developed in the existing literature.
>
> Though our method originates from similar techniques as in existing methods of solving diffusion ODEs, i.e., multiple-steps and predictor-corrector. Generalizing the multiple-step method into diffusion SDEs is non-trivial. Concretely, the multi-step and predictor-corrector of diffusion SDEs are from numerical SDE, and the core idea behind them is quite different from their versions under diffusion ODEs (e.g., Ito-Taylor expansion v.s. Taylor expansion).
>
> We also clarify the difference between SA-Solver (ours) with the existing sampling methods in Sec. 5.3, while most of them aim to accelerate the sampling of diffusion ODEs. We show that these methods are the certain case of the proposed SA-Solver.
>
> Q2: Some experiments are lacking, such as comparing the convergence of different methods on COCO using a pre-trained text-to-image diffusion model.
>
> A2: We test 4 diffusion samplers: DDIM, DPM-Solver, UniPC, and our SA-Solver on Stable Diffusion v1.5. Following the standard evaluation procedure, we randomly draw 30k prompts from the MS-COCO validation set and report the FID results on the generated images. The results are provided in the table below. The results show that all solvers achieve similar FID results. We attribute this to the powerful pretrained decoder, which can map a non-converged latent code to a good image sample. This phenomenon has also been observed in section 7.2 in [1].
>
> | NFE\method | DDIM  | DPM-Solver | UniPC | SA-Solver |
> |-|-------|------------|-------|-----------|
> | 20| 10.48 | 10.40      | 10.46 | 10.22     |
> | 60 | 10.30 | 10.47      | 10.40 | 10.33     |
>
> In [1][2], the authors compare the convergence speed of different diffusion ODE samplers by reporting the l2-distance between the sampled latent and the ground truth latent (approximated by 999 steps DDIM) under the same random seed and initial noise. However, this examination method does not fit stochastic samplers since even given the same initial noise, the intrinsic stochastic property will guide the stochastic samplers to different sampled latent.
>
>
>
> Q3: Some minor format issues: too much space below Table 1. Maybe the authors can try to make it looks better.
>
> A3: Thank you for your sincere advice. We will revise it in the final version.
>
> [1] Lu, C., Zhou, Y., Bao, F., Chen, J., Li, C. and Zhu, J., 2022. Dpm-solver++: Fast solver for guided sampling of diffusion probabilistic models.
> [2] Zhao, W., Bai, L., Rao, Y., Zhou, J. and Lu, J., 2023. UniPC: A Unified Predictor-Corrector Framework for Fast Sampling of Diffusion Models.

---

> > ### Comment · Reviewer_zg3j · 2023-08-11
> >
> > Thanks for your response. I will keep my score.

---

### Official Review · Reviewer_VoWY · 2023-07-12

**Soundness:** 3 good
**Presentation:** 3 good
**Contribution:** 2 fair
**Rating:** 5
**Confidence:** 3

**Summary:**

The paper proposed a stochastic Adam solver for solving diffusion SDEs in an efficient way with a convergence guarantee. Authors adapt the stochastic Adam from numerical literature and use Lagrange interpolation to predict unknown terms. They show strong convergence for both predictor and corrector. Numerical experiments are done to verify their claims.

**Strengths:**

1. By deriving an explicit form of the solution of the SDE, authors are able to utilize stochastic Adam with Lagrange interpolation to find an efficient diffusion SDE solver,
2. Theoretical analysis is well-written.
3. Numerical experiments show improvements compared to other solvers, especially the diffusion ODE solvers.
4. It sort of unifies previous methods in some sense.

**Weaknesses:**

1. My main concern is, from Figure 2, it seems SA-solver only outperforms other methods within a certain range of NFE. This "optimal" range looks very different for different data sets. In reality, if I use SA-solver for the sake of doing fewer function evals, how do I know when it outperforms other methods?

2. I don't think I fully understand how to choose the parameter tau(t) in the experiments. Please comment on this.

**Questions:**

See weaknesses.

**Limitations:**

yes

---

> ### Author Rebuttal · Authors · 2023-08-09
>
> First, we would like to thank you for taking the time to carefully review our paper, acknowledgment of our novel contributions, and the insightful questions. Below we respond to the questions:
>
> Q1: I don't think I fully understand how to choose the parameter tau(t) in the experiments. Please comment on this.
>
> A1: In our experiment, we use a constant function $\tau(t) = \tau, \forall t \in [0, T]$. In section 6.1, we use $\tau = 1$ for the comparison of the data-prediction model and noise-prediction model. In section 6.2, we vary $\tau$ from $\{0.0, 0.2, \cdots, 1.6\}$ to explore the effect of different NFE and $\tau$ to the result. In section 6.3, in relatively small NFEs, we set a proper small $\tau$ value and use $\tau = 1$ over 20 NFEs.
>
> In fact, we want to clarify that $\tau = 1$ for SA-Solver is generally not the optimal setting. We uniformly use it for our method in Tables 1 and 2 over different datasets because we aim to make a fair comparison with other methods (no extra-tuned hyperparameters).
>
> To further address the concern about choosing the proper $\tau(t)$, we conduct an ablation study in section 6.1 (see Figure 1) by varying the value of $\tau$. Our empirical results suggest that increasing $\tau$ with NFEs improves the FID.
>
> Moreover, we empirically observe that using a non-constant $\tau(t)$ yields better results than the one presented in this paper. Thus we suggest using it in practice. The strategy of such improved $\tau(t)$ is taking $\tau(t)=\tau$ in the interval $[t_{min}, t_{max}]$, and $\tau(t)=0$ otherwise, where $t_{min}$ and $t_{max}$ are selected as in [1] Below are the results under improved $\tau(t)$.
>
> | method\NFE on CIFAR 10    | 11       | 15       | 23       | 31       | 47       | 63       | 95       |
> |---------------------------|----------|----------|----------|----------|----------|----------|----------|
> | SA-Solver(vanilla)        | 7.49     | **4.84** | 4.04     | 3.41     | 3.18     | 3.24     | 3.17     |
> | SA-Solver(improved)       | 6.46     | 4.91     | **3.77** | **3.40** | **2.92** | **2.74** | **2.63** |
> | Best over baseline method | **6.41** | 5.01     | 4.04     | 3.82     | 3.59     | 3.36     | 3.06     |
>
> | method\NFE on ImageNet64  | 15       | 23       | 31       | 47       | 63       | 95       |
> |---------------------------|----------|----------|----------|----------|----------|----------|
> | SA-Solver(vanilla)        | 3.65     | 3.08     | 2.77     | 2.40     | 2.30     | 2.22     |
> | SA-Solver(improved)       | **3.41** | **2.61** | **2.23** | **1.95** | **1.88** | **1.81** |
> | Best over baseline method | 3.49     | 2.83     | 2.75     | 2.72     | 2.44     | 2.22     |
>
> Q2: My main concern is, from Figure 2, it seems SA-solver only outperforms other methods within a certain range of NFE. This "optimal" range looks very different for different data sets. In reality, if I use SA-solver for the sake of doing fewer function evals, how do I know when it outperforms other methods?
>
> A2: Thank you for pointing out this. Under the improved $\tau(t)$ as in the A1, our method almost consistently beats the other baseline methods as presented in the above Tables.
>
> [1] Karras, T., Aittala, M., Aila, T. and Laine, S., 2022. Elucidating the design space of diffusion-based generative models.

---

### Official Review · Reviewer_txgZ · 2023-07-15

**Soundness:** 3 good
**Presentation:** 3 good
**Contribution:** 2 fair
**Rating:** 6
**Confidence:** 4

**Summary:**

The paper proposes a new solver for diffusion SDEs, termed SA-Solver, combining the ideas of a predictor-corrector scheme and the stochastic Adam solvers. The predictor and corrector utilize the Lagrange polynomials for extrapolation to lower the approximation error at future time stamps. Experimentally, SA -Solver improves over previous ODE and SDE across a wide range of image generation benchmarks.

**Strengths:**

- The considered problem of accelerating SDE solvers is of great practical value in the field, as SDE solver often delivers better sample quality than ODE solvers, but are hindered by their slow sampling speed.

- Adapting the idea in Stochastic Adams is interesting, offering higher-order convergence with Lagrange polynomial extrapolation.

- The fusion of the PC sampler and Lagrange polynomial yields superior empirical results on diffusion SDE, tested on a range of dataset resolutions from 32x32 to 256x256. It's nice to see the methods scale to different resolutions and architectures.

**Weaknesses:**

- Throughout the paper, the authors mention several times that empirically the quality of data generated by SDE has a better upper limit. I think some theoretical analysis in the main text to make the paper more self-contained. A concurrent [1] provides some theoretical arguments and comparisons between SDE and ODE, showing that the stochasticity in SDE can reduce overall sample errors. I think it would be beneficial to discuss them to better motivate the idea of the paper.

- The authors didn't provide detailed discussions of the benefit of introducing $\tau(t)$ (at the end the author set $\tau(t)=1$ in experiments). Since the authors observe the benefit of larger $\tau$ on larger NFE, why not use a varying $\tau$ in Fig.2 based on NFE? In addition, the benefit of a larger $\tau$ on larger NFE seems predictable based on the theory in [1].

 - The authors did some comparisons of epsilon-prediction and data prediction models. It seems that EDM [2] uses interpolation between these two by pre-conditioning. Does it offer better results?

- Unlike the deterministic path in ODE, the randomness in SDE can change the sample trajectories quite a bit. For example, the trajectories of generating a dog can stray into generating a cat. I wonder if this is an issue for using Lagrange polynomials in SDE, since the preceding predictions could be misleading. If not, could the author provide some intuitions? In addition, the authors mention that "predictor step 2 and corrector step 1 is the most stable setup". such small steps could be caused by the issue I raise.

- The paper separately gives the convergence order of predictor and corrector. Is it possible to give the theoretical order after combing the predictor and corrector?

- I think the author didn't use the "EDM VE" model for CIFAR-10. In Table 2 of EDM [2], the unconditional VE (in config F, EDM VE) can achieve an FID of 1.98 using only 35 NFE, as opposed to the >3 FID in Fig 2 in the current paper. So I guess the author use their VE model in config A (baseline VE). Could the author compare different methods using the VP model (config A), as in [1] and [2], because the VP model is more commonly used in practice and offers better results?

- The stochastic sampler in the concurrent work [1] obtains an FID ~ 1.8 using less than 100 NFE on ImageNet-64 with Pixel DPM, as opposed to the > 2.1 FID in Fig 2 of the current paper. I wonder if one could combine the multi-step idea in this paper with [1] to obtain further improvements.

- Could the author also provide quantitative results on Stable Diffusion experiments, to better showcase the advantage of SA-Solver? Current visualized images do not seem compelling.


### Minors

- line 225, DPM-solver++ and UniPC are *ODE* solvers which would not naturally be special cases of the proposed *SDE* solver.



[1] Restart Sampling for Improving Generative Processes, Xu et al, https://arxiv.org/abs/2306.14878

[2] Elucidating the Design Space of Diffusion-Based Generative Models, Karras et al, https://arxiv.org/abs/2206.00364


# Post rebuttal

Thanks the authors for the rebuttal. After another pass of the paper, it occurred to me that that paper is a direct application of the ideas (variation-of-parameter, or exponential integrator, and Langraian polynomial for approximation) in [1] to SDE samplers. The same formula of the exponential integrated version of SDE (Eq.5) has been proposed in prior works [1] (see their eq.17, arXiv version 1). In addition, [1] also uses a Lagrange polynomial to extrapolate. Due to it's straightforward extension of prior works, I will keep my score unchanged.

[1] Zhang, Qinsheng, and Yongxin Chen. "Fast sampling of diffusion models with exponential integrator." arXiv preprint arXiv:2204.13902 (2022).

**Questions:**

I copy-and-paste some questions from above:

- Since the authors observe the benefit of larger $\tau$ on larger NFE, why not use a varying $\tau$ in Fig.2 based on NFE?

- I wonder if this is an issue for using Lagrange polynomials in SDE, since the preceding predictions could be misleading. If not, could the author provide some intuitions? In addition, the authors mention that "predictor step 2 and corrector step 1 is the most stable setup". such small steps could be caused by the issue I raise.

- Is it possible to give the theoretical order after combing the predictor and corrector?

- Could the author compare different methods using the VP model (config A), as in [1] and [2], because the VP model is more commonly used in practice and offers better results?

- Could the author also provide quantitative results on Stable Diffusion experiments, to better showcase the advantage of SA-Solver?

**Limitations:**

Yes

---

> ### Author Rebuttal · Authors · 2023-08-09
>
> First, we would like to thank you for taking the time to carefully review our paper, acknowledgment of our novel contributions, and the insightful questions. Below we respond to the questions:
>
> Q1: I think some theoretical analysis in the main text to make the paper more self-contained. A concurrent [1] provides some theoretical arguments and comparisons between SDE and ODE.
>
> A1: Thank you for pointing out the interesting concurrent work [1], we will add a discussion to it in our revised version. In this work, the authors clarify the advantage of SDE over ODE by proving it has a lower upper bound on sampling error in Wasserstein-1 distance. An upper bound of sampling error in terms of KL divergence for variance-controlled SDE is also provided in our supplementary material, which indicates the advantage of SDE over ODE in KL divergence.
>
> Q2: The authors didn't provide detailed discussions of the benefit of introducing $\tau(t)$. Why not use a varying $\tau$ in Fig.2 based on NFE?
>
> A2: Thank you for pointing out it. In fact, $\tau = 1$ for SA-Solver is not the optimal setting, we use it because we aim to make a fair comparison with other methods (no extra-tuned hyperparameters).
>
> We evaluate our SA-Solver with non-constant $\tau(t)$ inspired by EDM [2] and tuned $\tau$ in $0, 0.2\cdots,1.6, 1.8$, and we observe an improved performance. Concretely, a $\tau(t)=\tau$ in the interval $[t_{min}, t_{max}]$, and $\tau(t)=0$ otherwise, where $t_{min}$ and $t_{max}$ are selected as in [2].
>
> |method\NFE on CIFAR10(EDM baseline-VE)|11|15|23|31|47|63|95|
> |-|-|-|-|-|-|-|-|
> |SA-Solver(vanilla)|7.49|**4.84**|4.04|3.41|3.18|3.24|3.17|
> |SA-Solver(improved)|**6.46**|4.91|**3.77**|**3.40**|**2.92**|**2.74**|**2.63**|
>
> |method\NFE on ImageNet64(ADM)|15|23|31|47|63|95|
> |-|-|-|-|-|-|-|
> |SA-Solver(vanilla)|3.65|3.08|2.77|2.40|2.30|2.22|
> |SA-Solver(improved)|**3.41**|**2.61**|**2.23**|**1.95**|**1.88**|**1.81**|
>
> The first row uses the original setting in the paper, which contains constant $\tau(t)$, not tuned $\tau$.
>
> The second row uses the improved setting, which contains the piecewise constant $\tau(t)$, tuned $\tau$.
>
> Q3: The authors did comparisons of epsilon and data prediction models. Does EDM's interpolation offer better results?
>
> A3: It is an interesting point! In Section A.2.4, we show that the data model injects a smaller variance compared with the noise model. Similarly, the injected noise of the interpolation model is larger than the one of the data model. Thus we speculate interpolation may not bring extra benefits.
>
> Q4: I wonder if this is an issue for using Lagrange polynomials in SDE since the sample trajectories change and the preceding predictions could be misleading.
>
> A4: The theoretical convergence result indicates at least the stochastic multistep method will converge in the distribution sense.
>
> Q5: Is it possible to give the theoretical order after combing the predictor and corrector?
>
> A5: The convergence result of the ODE predictor-corrector method is well established, e.g. see [5]. To the best of our knowledge, the theoretical convergence result of the stochastic multi-step method with p-c is not known yet.
>
> Q6: So I guess the author use their VE model in config A (baseline VE).
>
> A6: Yes. We will revise the expression to 'EDM baseline-VE' to avoid misunderstanding.
>
> Q7: Could the author compare different methods using the VP model (config A)?
>
> A7: We provide the results of the baseline-VP CIFAR10 model as below.
>
> |method\NFE |11|15|23|31|47|63|
> |-|-|-|-|-|-|-|
> |DDIM|17.07|11.57|7.33|5.68|4.41|3.89|
> |DPM-Solver|**6.31**|**4.72**|3.46|3.28|3.07|2.99|
> |UniPC|7.05|5.59|3.08|2.88|2.88|2.88|
> |EDM-ODE|18.41|6.52|3.52|3.10|2.99|2.95|
> |EDM-SDE|29.90|10.21|4.85|3.77|3.08|2.84|
> |SA-Solver|7.05|5.59|**3.03**|**2.70**|**2.50**|**2.39**|
>
> Q8: The stochastic sampler in the concurrent work [1] obtains an FID ~ 1.8 using less than 100 NFE on ImageNet-64, as opposed to the $>$ 2.1 FID of the current paper. I wonder if one could combine the two ideas to obtain further improvements.
>
> A8: Thank you for your advice! We think the idea in [1] is interesting and the result is promising. We first want to point out that we have a typo in line 271, and it causes a misleading. We actually use EDM baseline-VE for CIFAR10 while ADM[4] for ImageNet64. We will correct this typo in the revised version.
>
> Our results for ImageNet64 are based on ADM which is slightly weaker than EDM, i.e., 1.55 v.s. 1.36 for SOTA FID. To fairly compare, we conduct our method with the non-constant $\tau(t)$ in A2 on EDM ImageNet64 model and provide the results below. The idea that combines the two papers can be explored in the future.
>
> |method\NFE|39|67|99|165|
> |-|-|-|-|-|
> |Restart|2.38|1.95|1.71|1.51|
> |SA-Solver|**1.80**($\tau=1.0$)|**1.58**($\tau=1.4$)|**1.49**($\tau=1.8$)|**1.44**($\tau=2.2$)|
>
> Q9: Could the author also provide quantitative results on Stable Diffusion experiments, to better showcase the advantage of SA-Solver? Current visualized images do not seem compelling.
>
> A9: We test on Stable Diffusion v1.5. Following the standard evaluation procedure, we randomly draw 30k prompts from the MS-COCO validation set. The results show that all solvers achieve similar FID results. We attribute this to the powerful pretrained decoder, which can map a non-converged latent code to a good image sample. This phenomenon has also been observed in section 7.2 in [3].
>
> |NFE\method|DDIM|DPM-Solver|UniPC|SA-Solver|
> |-|-|-|--|-|
> |20|10.48|10.40|10.46|10.22|
> |60|10.30|10.47|10.40|10.33|
>
> [1]Xu, Yilun, et al. "Restart Sampling for Improving Generative Processes."
>
> [2]Karras, Tero, et al. "Elucidating the design space of diffusion-based generative models."
>
> [3]Lu, Cheng, et al. "Dpm-solver++: Fast solver for guided sampling of diffusion probabilistic models."
>
> [4]Dhariwal, Prafulla, and Alexander Nichol. "Diffusion models beat gans on image synthesis."
>
> [5]Gragg, William B., and Hans J. Stetter. "Generalized multistep predictor-corrector methods."

---

### Official Review · Reviewer_UcRr · 2023-07-26

**Soundness:** 3 good
**Presentation:** 3 good
**Contribution:** 4 excellent
**Rating:** 7
**Confidence:** 4

**Summary:**

This paper studies the problem of fast sampling of diffusion models. Since standard DDPM sampling is slow, this is a very important topic of late, with many competing methods. Many methods reformulate as a ODE solving problem, which makes it easier to do few-step sampling. However, it has been noted that sampling from the original SDE formulation (as opposed to ODE) can lead to better samples if there is budget for taking many diffusion steps. Therefore, this paper studies the problem of accelerating sampling of the SDE.

The method, named SA-Solver is a predictor corrector method. It incorporates some recent findings, such as semi-linearity, but mainly uses the method of Lagrange interpolations. More specifically, the predictor step involves (uniquely) fitting a polynomial of degree s-1 using s predictions and reading off the polynomial value at desired points. Finally, a correction step is incorporated that plug in the predicted value at the new timestep (t+1) back into the formula to choose a new predicted value (similar in spirit to heun's method). The paper draws connections to DDIM (deterministic version of 1-step SA-Predictor), DPM-Solver (deterministic version of 2-step SA-Predictor), and UniPC (deterministic version of SA_Solver)

The paper shows sampling results on cifar10, imagenet64, and imagenet256 (latent), demonstrating best FID scores (conditioned on the same number of NFEs) when using more than 30 NFEs.

**Strengths:**

The problem being studied is an important one, as sampling speed is arguably the biggest weakness of diffusion models (and diffusion models are being used in a huge number of applications right now).

The experimental results seem quite promising, showing better FID scores than DDIM/DPMSolver, which are very popular baselines.

**Weaknesses:**

Seems to be missing quantitative evaluation (e.g. CLIP score on stable diffusion) for text-to-image tasks. Also, evaluating on different domains besides image would improve the empirical results.

**Questions:**

How much of the performance is due to the predictor/corrector approach, versus the lagrange interpolation? An ablation study in this regard would improve the paper and our understanding of the method.

---

> ### Author Rebuttal · Authors · 2023-08-09
>
> First, we would like to thank you for taking the time to carefully review our paper, acknowledgment of our novel contributions, and the insightful questions. Below we respond to the questions:
>
> Q1: Seems to be missing quantitative evaluation (e.g. CLIP score on stable diffusion) for text-to-image tasks.
>
> A1: In fact, for text-to-image tasks, papers on fast samplers rarely report the FID and CLIP scores, as all solvers achieve similar FID results [1][2]. These works attribute this to the powerful pretrained decoder, which can map a non-converged latent code to a good image sample [1][2]. We test 4 diffusion samplers: DDIM, DPM-Solver, UniPC, and our SA-Solver on Stable Diffusion v1.5. Following the standard evaluation procedure, we randomly draw 30k prompts from the MS-COCO validation set and report the FID results on the generated images. The results are provided in the table below, which shows similar conclusions as the previous works.
>
> | NFE\method | DDIM  | DPM-Solver | UniPC | SA-Solver |
> |------------|-------|------------|-------|-----------|
> | 20         | 10.48 | 10.40      | 10.46 | 10.22     |
> | 60         | 10.30 | 10.47      | 10.40 | 10.33     |
>
> Q2: Also, evaluating on different domains besides image would improve the empirical results.
>
> A2: Thanks for your valuable advice! But we are not familiar with generation tasks in other domains. We will consider adding some experiments in other domains in the future.
>
> Q3: How much of the performance is due to the predictor/corrector approach, versus the Lagrange interpolation? An ablation study in this regard would improve the paper and our understanding of the method.
>
> A3: This is an interesting and valuable question. To explore this, we conduct an ablation study on the effect of Lagrange interpolation and predictor-corrector on the CIFAR10 dataset as follows. We use EDM baseline-VE pretrained checkpoint. Concretely, we vary the number of predictor steps and meanwhile conduct them with/without corrector to separately explore the effect of the two components. As can be seen, both Lagrange interpolation (Predictor 1-steps only v.s. Predictor 3-steps only) and predictor-corrector (Predictor 1-steps only v.s. Predictor 1-steps, Corrector 1-step, and Predictor 3-steps only v.s. Predictor 3-steps, Corrector 3-steps) improve the performance of our sampler.
>
> | method\NFE and $\tau$                    | 15 0.4   | 23 0.8   | 31 1.0   | 47 1.4   |
> |------------------------------------------|----------|----------|----------|----------|
> | Predictor 1-step only                    | 13.76    | 12.44    | 11.72    | 14.67    |
> | Predictor 1-step with Corrector 1-step   | 8.49     | 6.87     | 6.13     | 6.75     |
> | Predictor 3-steps only                   | 5.30     | 3.93     | 3.52     | 2.98     |
> | Predictor 3-steps with Corrector 3-steps | **4.91** | **3.77** | **3.40** | **2.92** |
>
> [1] Lu, C., Zhou, Y., Bao, F., Chen, J., Li, C. and Zhu, J., 2022. Dpm-solver++: Fast solver for guided sampling of diffusion probabilistic models.
>
> [2] Zhao, W., Bai, L., Rao, Y., Zhou, J. and Lu, J., 2023. UniPC: A Unified Predictor-Corrector Framework for Fast Sampling of Diffusion Models.

---

> > ### Comment · Reviewer_UcRr · 2023-08-10
> > **Response**
> >
> > Thank you for the response, the ablation study is excellent. I stand by my recommendation of acceptance.
> >
> >
> > A1: for text-to-image tasks, papers on fast samplers rarely report the FID and CLIP scores, as all solvers achieve similar FID results [1][2]. These works attribute this to the powerful pretrained decoder, which can map a non-converged latent code to a good image sample
> >
> > Having a powerful decoder is a property of latent diffusion (as opposed to pixel diffusion), and is not an issue of text-to-image vs unconditional generation right? In other words, this statement seems to also imply Imagenet 256x256 with latent DPM is easy?

---

> > > ### Author Response · Authors · 2023-08-13
> > >
> > > Thank you for your comment. We believe that two factors are present, influencing the FID of sd v1.5 on MS-COCO. One is the encoder/decoder architecture and the other is the text-to-image tasks. The rationale behind the latter aspect has not been definitively established yet. However, we hypothesize that improved sample quality doesn't necessarily guarantee that the sample will resemble the COCO sample under the same prompt. Please let us know if you have any other questions. Thanks again for your valuable review!

---

### Official Review · Reviewer_tWFq · 2023-08-01

**Soundness:** 3 good
**Presentation:** 2 fair
**Contribution:** 3 good
**Rating:** 5
**Confidence:** 4

**Summary:**

The paper presents a multistep SDE solver for diffusion models instead of ODE solvers. The main goal is to have diverse and high-quality samples while reducing the number of solver step required. To do this, the paper proposes a new SDE that includes an additional term, ensuring the marginal distribution unchanged. Based on this new SDE, the paper adopts stochastic Adam method, introducing two key component: SA-predictor, and SA-corrector. The convergences of these two are provided. Experiments are conducted in different scenario including varying stochastic noise scale, different models. The results support the claims on the reduction in NFEs.

**Strengths:**

The paper has the following strong points:

- The experiment shows state-of-the-art result in term of FID given a limited NFEs. This highlights the effectiveness of the proposed method in achieving impressive results with efficient resource utilization.
- It provides theoretical studies on the convergence of proposed method, offering valuable insights

**Weaknesses:**

The main weaknesses I find in this paper are:

- In term of the contribution to difussion model research, the proposed method may be considered incremental. This is because the key factor leading to good result, such as predictor-corrector, multi-step have been well-established in the existing literature.
- The paper presentation can be improved, in particular the exposure of stochastic Adam methods. I would be good to dedicate a small paragraph or section explicitly discussing ths method to provide readers with clear understand of how it is employed in the proposed approach.
- The paper lacks clarity in establishing the connection between the variance-control SDE and the use of stochastic Adam methods. The discussion should be more thorough and elaborate, especially in addressing why setting $\tau=1$ (falling back to DDPM) is the best choice in many cases. Then, stochastic Adam significantly contributes to the model's overall performance.

**Questions:**


How many of predictor step $s_p$ and corrector step $s_c$ is used?

Minor points,
- Line 199, please add the notation stepsize $h$
- Eq (13), please add the description for the Lagrange basis $l(t)$.
- Caption of Fig. 1, "scholastic" -> "stochastic"
- There are some typos in Appendix

**Limitations:**

The paper mentioned a current limitation is that the optimal $\tau(t)$ is unknown, which I agree. There many factors affecting this parameters including NFE, dataset and approximated score function.

---

> ### Author Rebuttal · Authors · 2023-08-09
>
> We thank you for your valuable comments and carefully reviews. Below are our responses to the raised questions:
>
> Q1: The proposed method may be considered incremental. This is because the key factor leading to good results, such as predictor-corrector, and multi-step have been well-established in the existing literature.
>
> A1: The multi-step solver and predictor-corrector method have been well-established and studied in ODE solvers of diffusion models, e.g. [1][2][3]. However, as we have claimed in lines 38-39, "adding properly scaled noise in the diffusion SDE may facilitate the quality of generated data". Thus we focus on developing a solver for diffusion SDEs in this paper, which has not been well developed in the existing literature.
>
> Back to your concern, although the mentioned techniques (e.g., multiple-step method) have been used in diffusion ODEs, generalizing it into diffusion SDEs is non-trivial. Concretely, the core technique of the multi-step method of diffusion SDEs is from numerical SDE (e.g., Ito-Taylor expansion), which is quite different from the one used in diffusion ODEs.
>
> Q2: The paper presentation can be improved, in particular the exposure of stochastic Adam methods. I would be good to dedicate a small paragraph or section explicitly discussing ths method to provide readers with a clear understanding of how it is employed in the proposed approach.
>
> A2: Thank you for your sincere advice. We will carefully revise the order of our content and make it more readable and make a discussion about the stochastic Adam methods right after line 168 as you suggested. And we consider adding a section to explicitly introduce the stochastic Adams method in the appendix to help readers better understand the method.
>
> Q3: The paper lacks clarity in establishing the connection between the variance-control SDE and the use of stochastic Adam methods.
>
> A3: Stochastic Adams method is a numerical method to solve SDEs and variance-controlled SDE is a specialized SDE. The connection has been clarified at the beginning of Section 5.
>
> Q4:  The discussion should be more thorough and elaborate, especially in addressing why setting $\tau = 1$
>  (falling back to DDPM) is the best choice in many cases.
>
> A4: Thank you for pointing out it. In fact, we want to clarify that $\tau = 1$ for SA-Solver is generally not the optimal setting,  as in line 259. We uniformly use it over different datasets because we aim to make a fair comparison with other methods (no extra-tuned hyperparameters). As can be seen in the ablation study to $\tau$ in Tables 6, 7, 8, and 9 in our supplementary material, SA-Solver with larger $\tau$ (even larger than 1) exhibits improved results, especially under larger NFEs.
>
> To further address your concern, we evaluate our SA-Solver with improved (compared with results in this paper) non-constant $\tau(t)$ inspired by EDM [4]. Concretely, a $\tau(t)=\tau$ in the interval $[t_{min}, t_{max}]$, and $\tau(t)=0$ otherwise, where $t_{min}$ and $t_{max}$ are selected as in [4]. The results are summarized below, and we will add them to the revised version.
>
> | method\NFE on CIFAR10(EDM baseline-VE)    | 11       | 15       | 23       | 31       | 47       | 63       | 95       |
> |---------------------------|----------|----------|----------|----------|----------|----------|----------|
> | SA-Solver(vanilla)        | 7.49     | **4.84** | 4.04     | 3.41     | 3.18     | 3.24     | 3.17     |
> | SA-Solver(improved)       |**6.46**     | 4.91     | **3.77** | **3.40** | **2.92** | **2.74** | **2.63** |
>
> | method\NFE on ImageNet64(ADM)  | 15       | 23       | 31       | 47       | 63       | 95       |
> |---------------------------|----------|----------|----------|----------|----------|----------|
> | SA-Solver(vanilla)        | 3.65     | 3.08     | 2.77     | 2.40     | 2.30     | 2.22     |
> | SA-Solver(improved)       | **3.41** | **2.61** | **2.23** | **1.95** | **1.88** | **1.81** |
>
> The first row uses the original setting in the paper, which contains constant $\tau(t)$, not tuned $\tau$.
>
> The second row uses the improved setting, which contains the piecewise constant $\tau(t)$, tuned $\tau$.
>
> Q5: How many of predictor step $s_p$ and corrector step $s_c$ is used?
>
>  A5: As clarified in line 239, we use $s_p = 2$ and $s_c = 1$ in the experiment part as they are observed stable in practice.
>
>
> [1] Liu, L., Ren, Y., Lin, Z. and Zhao, Z., 2022. Pseudo numerical methods for diffusion models on manifolds.
>
> [2] Zhao, W., Bai, L., Rao, Y., Zhou, J. and Lu, J., 2023. UniPC: A Unified Predictor-Corrector Framework for Fast Sampling of Diffusion Models.
>
> [3] Li, S., Liu, L., Chai, Z., Li, R. and Tan, X., 2023. ERA-Solver: Error-Robust Adams Solver for Fast Sampling of Diffusion Probabilistic Models.
>
> [4] Karras, T., Aittala, M., Aila, T. and Laine, S., 2022. Elucidating the design space of diffusion-based generative models.

---

### Decision · Program_Chairs · 2023-09-21

**Decision:**

Accept (poster)

**Comment:**

The reviews on this paper were overall positive. While some reviewers questioned the possibly incremental improvement upon existing SDE solvers (e.g., the predictor-corrector method for SDE has been known to be important for sample quality for a while now), I think experimental results presented in the paper make up for this. Also, this work contains some nice theoretical analysis. I encourage the authors to incorporate the feedback from the reviewers for the camera ready, in particular addressing the questions around the setting of $\tau$, and furthermore including the many ablation studies run in the rebuttal.